# TALES: Text Adventure Learning Environment Suite

## Abstract

Reasoning is an essential skill to enable Large Language Models (LLMs) to interact with the world. As tasks become more complex, they demand increasingly sophisticated and diverse reasoning capabilities for sequential decision-making, requiring structured reasoning over the context history to determine the next best action. We introduce TALES, a diverse collection of synthetic and human-written text-adventure games designed to challenge and evaluate diverse reasoning capabilities. We present results over a range of LLMs, open- and closed-weights, performing a qualitative analysis on the top performing models. Despite an impressive showing on synthetic games, even the top LLM-driven agents fail to achieve 20% on games designed for human enjoyment. Visualization of the experiments can be found at https://github.com/tale-suite/tale-suite-anonymized.

## 1 Introduction

Reasoning is crucial in sequential decision-making tasks where optimal actions depend on previous choices whose effects may only emerge later. In complex tasks, the agent often needs to leverage a variety of reasoning skills to make the best decision. This becomes more challenging in grounded environments, where the causal constraints between actions are fixed and cannot be violated. Therefore, the ability of a Large Language Model (LLM) to perform this structured thinking and follow these constraints across long contexts is critical for real-world application (Trivedi et al., 2024).

Through the lens of the Problem Space Hypothesis for complex problem solving (Newell, 1979), we identify four core reasoning skills vital to an LLM-driven agent's ability to interface with applications in real-world settings where there is limited human intervention: **Deductive reasoning**, to act upon general principles (Johnson-Laird, 1999); **Inductive reasoning**, to draw conclusions from interaction and observation (Heit, 2000); **Spatial reasoning**, to efficiently navigate and understand the spatial relationship between objects (Byrne & Johnson-Laird, 1989); and **Grounded reasoning**, to identify relevant information and perform admissible actions in a given context (Endsley et al., 2000). For an LLM agent to be successful in real-world applications, it must be able to continuously use and combine these core reasoning skills at every step.

Text-adventure games are a prime test-bed to evaluate an LLM agent's ability to reason in these modes due to the need to apply commonsense principles to perform directed exploration (Deductive), discover implicit dynamics through trial and error (Inductive), and operate in a situated environment (Spatial and Grounded). Figure 1 illustrates an agent navigating through a text-adventure game. At each step, one or more of the core reasoning skills may be required for optimal decision making, while a single failure in any reasoning skill can dramatically reduce overall performance. Success in these environments require the consistent and compositional use of the core reasoning skills, mirroring the challenges faced by LLM agents in real-world applications. While previous works have explored using text-adventure games as a metric of an LLM agent's capabilities, they either focus on one specific framework, provide significant scaffolds for the LLM, or drastically change the scope of the original task (Paglieri et al., 2024; Chang et al., 2024).

To evaluate an LLM agent's comprehensive reasoning capabilities, we introduce TALES, the first benchmark that unifies TextWorld (TW), TextWorldExpress (TWX), ALFWorld (AW), ScienceWorld (SW) and Jericho in their canonical forms. Unlike other benchmarks, we remove all environment-specific scaffolding in favor of a standardized evaluation. This creates a challenging and comprehensive evaluation suite of 122 tasks for better understanding the agent's

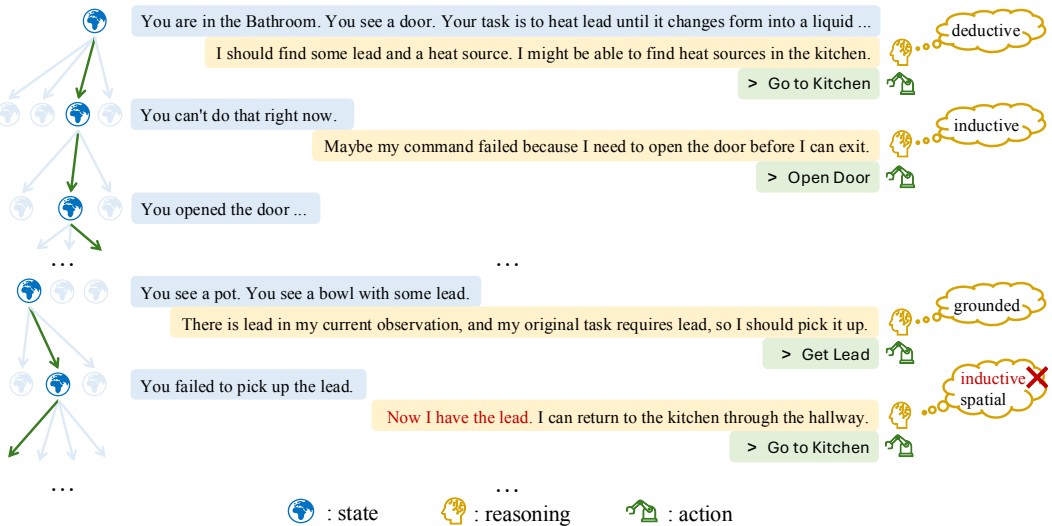

Figure 1: Example of a gameplay trajectory presenting the conversation between the game engine and an agent. We additionally fabricate the agent's reasoning to demonstrate the reasoning types this work concerns, detailed in Section 2. Here, the agent made a mistake in its inductive reasoning, which further caused the generation of a sub-optimal action.

baseline composite reasoning skills. We introduce TALES by following the ECBD framework (Liu et al., 2024) that outlines the key benchmark design decisions.

As an initial litmus test of an agent's compositional reasoning skills, we introduce the game SIMON SAYS to assess whether the agent has the baseline capabilities required to challenge TALES. In this classic children's game, players must follow instructions only when prefaced with "Simon says" - making it fundamentally an instruction-following task. The simplest formulation of our text-adventure implementation gives the player a direct walkthrough of required actions, similar to the iconic copy task (Graves et al., 2014) where models must reproduce given sequences. Despite this programmatic simplicity, we find that even advanced models struggle with this straightforward instruction-following challenge. We discover that success in this elementary task strongly predicts (Pearson r = 0.83) a model's ability to make meaningful progress in the more complex environments of TALES.

We show the performance of 42 models, open- and closed-weights, in a zero-shot setting on TALES **and perform extensive Reinforcement Learning (RL) fine-tuning experiments to quantify the impact of privileged or domain knowledge commonly used in prompts to scaffold agent performance.** We analyze game transcripts to identify common behaviors and failure modes from top models. While thinking LLMs tend to outperform non-thinking models overall, both model types struggle to reason across extremely long-horizon contexts where important information is sparsely scattered throughout. This limitation significantly hinders their ability to progress through the JERICHO framework (Hausknecht et al., 2020), a collection **of** games meant to be played by humans, slowly and iteratively over extended periods of time. Despite strong evidence of data contamination (See Appendix R) and success on easier frameworks, no agent is capable of completing the gauntlet of games in TALES in a zero-shot setting with minimal inductive bias.

In summary, our contributions are as follows:

- We introduce TALES, a unified framework **suite** for evaluating agents in text-adventure games.
- We outline the reasoning skills required for an agent to be able to successfully complete any text-adventure game **task** in the benchmark.
- We introduce the new SIMON SAYS game mode where the agent must echo a command sequence.
- We investigate the game transcripts of AW, SW and the iconic ZORK1 to find where even the top models fail to progress in games meant to be enjoyed and solved by human ingenuity.
- We provide zero-shot results averaged over 5 runs for 42 of the top LLMs as of the time of writing.

• We examine the effects of common prompt scaffolds through the RL fine-tuning of Qwen3-8B.

## 2 REASONING

The Problem-Space Hypothesis posits that humans solve problems through the creation and manipulation of problem spaces (Newell, 1979). A problem space is often represented by initial, intermediate, and goal states, possible operators, and environmental constraints. Complex tasks typically require a combination of reasoning skills to navigate these spaces effectively. Consider an LLM agent for computer use: it must identify its current state (screen content) and goal state (task completion), while discovering operators, both familiar and unclear interface affordances. The agent must apply existing knowledge to infer environmental constraints (e.g., what actions are permissible). As tasks grow complex, the problem space expands with both known and unknown operators and constraints. Success requires the agent to leverage a composite of different reasoning skills. Lacking a specific reasoning skill would lead to task failure as the error will cascade and be difficult to recover.

From these insights on the Problem-Space Hypothesis, we identified four reasoning skills critical for LLM agents to succeed in complex, grounded tasks. Those skills comprise the capability module (Liu et al., 2024) measured by TALES.

**Deductive reasoning:** The ability to derive valid actions through the logical application of general principles within a specific environmental context (Johnson-Laird, 1999). Deductive reasoning is particularly critical when environmental interactions are limited or when action has substantial costs and irreversible consequences. In such scenarios, the agent must leverage pre-existing knowledge to understand the affordance and constraints of the context and make correct actions towards the goal.

**Inductive reasoning:** The ability to draw conclusions through interactions and observations. This is a critical skill for agents that interface with complex, interactive systems. Given the diversity of tasks, the environment's affordance may be unknown or contradict with general principles (e.g., a software interface element behaving inconsistently across operating systems). The inductive reasoning skill allows the agent to discover operators (actions) and rules about the environment through exploration. This skill encompasses both adjustments to API calls to adhere to strict function signatures as well as learning from system feedback (Zhong et al., 2024).

**Spatial reasoning:** The ability to navigate effectively and understand the spatial relationship among game objects, including path finding, backtracking, and locating items (Momennejad et al., 2023). An agent with spatial reasoning skills can integrate information about locations to identify intermediary states between the initial and goal states.

**Grounded reasoning:** The ability to make decisions based on relevant information and current context. An agent with grounded reasoning skills can accurately perceive and understand the current state of the problem space. This reasoning skill is analogous to situational awareness in humans. Although an LLM may be pre-trained on a vast amount of world knowledge, it has to attend to task-specific information when making its decisions. As agents **may** have access to the full interaction history at every step, the ability to correctly identify what information is relevant to the current state and reason over said information becomes **critically** important as the length of the history grows.

The ability to leverage all of these skills is critical to the success of agents as the complexity of the task increases. Within longer contexts, these reasoning skills often become compositional with a failure in one skill leading to failures in the others later on.

We believe text-adventure games are an ideal environment to simulate the challenges of real-world tasks, i.e., evaluating an agent on all four core reasoning skills at the same time. A deductive failure may lead an agent to waste a large number of steps trying to accomplish a sub-goal that does not progress towards the objective. Inductive reasoning failures may cause an agent to repeatedly attempt the same action over and over without success. Figure 1 illustrates a simple task in a text-adventure game where multiple reasoning skills **may be** required at each step and a single failure leads to later errors. We discuss explicit failure modes that result from the lack of these core reasoning skills in Section 5 and design choices meant to allow for better evaluation in Section 6

Broadly speaking, interactive text-game environments are ideal test-beds for exploring highly complex, interactive and grounded scenarios where previous work has shown the potential for cross-domain transfer of knowledge and experiences (Ammanabrolu & Riedl, 2021).

| Properties | TEXTWORLD (TW) | TEXTWORLDEXPRESS (TWX) | ALFWORLD (AW) | SCIENCEWORLD (SW) | JERICHO (JERICHO) |
|---|---|---|---|---|---|
| # Games | 10 | 16 | 12 | 30 | 54 |
| Avg. walkthrough **steps** | 13.70 | 33.06 | 5.83 | 41.67 | 87.15 |
| Informative feedback | ✓ | ✓ | ✗ | ✓ | ✓ |
| Intermediate rewards | ✓ | ✓ | ✗ | ✓ | ✓ |
| Nearest-neighbor parser | ✓ | ✗ | ✗ | ✓ | ✓ |
| Dead States | ✓ | ✓ | ✗ | ✓ | ✓ |

Table 1: Key properties of each framework. Informative feedback: when an action fails, specific details about the nature of the failure are returned. Nearest-neighbor parser: environment can understand similar action phrases to mean the same thing. Dead States: actions can prevent completing the task until a reset. **All frameworks aside from** JERICHO **are synthetic: created specifically to train embodied AI agents.**

## 3 TALES

All frameworks included in TALES are *text-adventure game* environments where players are provided a textual observation, and sometimes an explicit goal, and are able to interact with the environment through short action phrases. If these action phrases are invalid, the parser will typically return an error message indicative of whether the action has been understood by the environment but cannot be done, or if the parser just does not understand the action. **Table 1 contrasts properties for each framework.** We provide a short description of each environment and any notable characteristics about the environment or rewards in the following section. We organize the following sections in the rough order of difficulty with the recommendation that users avoid testing on later frameworks without an agent that is able to succeed in earlier environments due to the scaling difficulty **and computational costs for evaluation.** We do not use all environments within each framework. See Appendix P for more details.

### 3.1 SIMON SAYS: YOU SHALL NOT PASS UNLESS YOU CAN SOLVE THIS TASK

For all frameworks included in TALES, there is a requirement for the agent to be at least minimally proficient in all reasoning skills to make any non-trivial progress. With the release of TALES, we also introduce a new TWX game in the form of "SIMON SAYS". The basic SIMON SAYS simply provides the agent an action to repeat at each turn while SIMON SAYS WITH MEMORY provides a list of actions to follow at the start of the game. Both versions award a point for every correct action. The game restarts if any action is performed out of order or is wrong. SIMON SAYS is unique compared to other games in TALES as it requires minimal reasoning to complete. However, we find it serves as a good heuristic to evaluate whether an agent is likely to succeed in TALES with a Pearson correlation coefficient of .83 when taken with respect to the average of all SIMON SAYS games against the entirety of TALES. A prerequisite to success in TALES is the ability to at least properly attend to information over a long horizons. SIMON SAYS is the simplest form of this, posed in a straightforward, instruction-following task. We include a graph visualizing the correlation between success in SIMON SAYS and success in TALES in Appendix K.

### 3.2 FRAMEWORKS

TEXTWORLD (TW) (Côté et al., 2018) is a framework originally designed for training agents with Reinforcement Learning (RL) on text-based games. It can generate synthetic text-adventure games of varying complexity. In TALES, we integrate the "CookingWorld" games that were used as part of the NeurIPS 2018 Competition[1]. The task involves following a recipe that requires finding ingredients and processing them according to said recipe. We selected one game per difficulty ranging from level 1 (with one location and a recipe of 1 ingredient) to level 10 (having 12 locations and a recipe with 3 ingredients). The player receives 1 point after completing sub-goals related to the task in the game.

---

[1]https://competitions.codalab.org/competitions/21557

TEXTWORLDEXPRESS (TWX) (Jansen & Côté, 2022) is a highly optimized re-implementation of many TW game scenarios that runs approximately three orders of magnitudes faster compared to the TW counterparts. We opt to use TWX over TW for the performance improvement where applicable. While significantly faster, an arguable drawback of using TWX over TW is also in its stricter parser. TWX simplifies its parser for speed and thus does not allow for nearest-neighbor action phrases. **Compared to other environments, the feedback from `help` only re-iterates the game objective.**

ALFWORLD (AW) (Shridhar et al., 2021) is a multi-modal framework, combining complementary visual and textual observations, where agents are asked to navigate and perform tasks in a household setting. All tasks provide only a terminal reward of 1 upon task completion. For TALES, we only use its textual modality as it has become the standard in the LLM literature when evaluated on AW (Yao et al., 2023; Shinn et al., 2023). The AW environments are unique in their lack of informative feedback. Where other environments have a predefined error message relating to the type of error, whether it is due to the parser not recognizing the command or the action not being possible, AW has only one error message: **`Nothing happens`**. In the original AW framework, the visual component compensates for the lack of detailed text feedback. However, this makes it significantly harder for agents relying solely on text-based interactions, compounded by the limitation that in ALFWORLD you can hold one object at a time.

SCIENCEWORLD (SW) (Wang et al., 2022) is a framework focused on the completion of elementary-level science curriculum tasks. Notably for many of its tasks, SW emulates an open-world setting where the player can complete the task in different ways that do not follow one expected trajectory. When it comes to heating objects, this part of the task can be completed by either the oven in the kitchen or the blast furnace in the workshop. Similarly, SW also allows the player the freedom to reset the game on command. This is especially important as a number of SW games have failure modes where it is no longer possible to complete the assigned task in that playthrough.

JERICHO (Hausknecht et al., 2020) is a suite of $54^2$ human-written, interactive fiction games. We consider JERICHO to be the most difficult framework due to the length and complexity of many of the games. Some can be completed within 17 steps while some others require over 500 steps. These games also cover an extremely wide range of genres and styles and lack the consistency of many other text-game environment suites designed for evaluating agents. For example, 9:05 follows the morning of an ordinary office worker **while** ANCHORHEAD is a Lovecraftian Horror Story.

## 4 EVALUATION

TALES enables evaluation by customizing models with specific prompts and agentic strategies. For our initial release, we adapt examinee models by considering a minimal agent scaffolding that uses the following prompt in a zero-shot settings, i.e., without any examples of playing text-based games.

> You are playing a text-based game and your goal is to finish it with the highest score. Upon reading the text observation, provide a *single* short phrase to interact with the game, e.g. `get lamp` (without backticks). When stuck, try using the `help` command to see what commands are available.

For the main results, we do not provide any other instructions to the LLMs on how to play the game. We aim to measure LLMs' raw capabilities exempt of inductive bias from **a** human expert with domain knowledge. In Section 6, we explore the effects of providing **different forms of privileged or domain knowledge directly** to the agent. When calling the LLMs, the observation and feedback are provided as the *user* inputs while the LLM actions are recorded as the *assistant* outputs.

For our results in the initial release of TALES, we cap the number of steps the agents can take in any environment to 100. In most frameworks, 100 steps serves as ample opportunity to make mistakes, self-correct, and eventually find success through directed exploration without allowing for a randomly acting agent to eventually happen upon success through chance. Even though only 57% of the total score is achievable within 100 steps in JERICHO, no agent approaches this score. This step limit thus serves an effective medium between the easier and the most challenging frameworks in TALES. As the step number is not explicitly referenced in the system prompt, this allows us to continue exploring longer horizon performances leveraging current game history in future work.

---

[2]We exclude HOLLYWOOD.Z3 because of segfault errors and THREATRE.Z5 due to game engine errors.

Table 2: Average scores per framework and total TALES score for the top 10 models. TALES **score is calculated by averaging success over all tasks.** Almost uniformly, reasoning LLMs outperform non-reasoning LLMs in all frameworks. However, a higher thinking budget does not always lead to better overall results.

| Model | TEXTWORLD | TEXTWORLDEXPRESS | ALFWORLD | SCIENCEWORLD | JERICHO | TALES Score |
|---|---|---|---|---|---|---|
| o3 (medium) | **100** | 91.9 | 88.3 | 93.0 | 15.7 | **58.7** |
| o3 (high) | 100 | 89.6 | 81.7 | **93.1** | 16.1 | 58.0 |
| gpt-5 (thinking) | 100 | 75.5 | **93.3** | 91.8 | **17.2** | 57.5 |
| o3 (low) | 99.1 | 89.8 | 70.0 | 88.3 | 14.2 | 54.8 |
| claude-3.7-sonnet (thinking) | 97.3 | 91.3 | 83.3 | 76.5 | 12.5 | 52.5 |
| claude-3.7-sonnet | 97.3 | **95.8** | 81.7 | 72.4 | 13.0 | 52.1 |
| claude-3.5-sonnet-latest | 95.5 | 81.6 | 75.0 | 82.3 | 9.6 | 50.4 |
| gpt-4.1 | 95.3 | 92.5 | 83.3 | 76.1 | 6.8 | 49.9 |
| gpt-5-mini (thinking) | 94.7 | 61.9 | 61.7 | 82.7 | 9.5 | 46.5 |
| o1 | 97.8 | 70.2 | 28.3 | 80.1 | 10.3 | 44.2 |

TALES captures the model's capability evidence by the score from each game environment, ranging from 0-100. Although each game environment has its own customized scoring rules, those rules mark significant milestones in solving the game. In Appendix B, we include a breakdown of the percentage of the max score from following the game walkthrough to a certain number of steps in JERICHO. Each game is played 5 times to account for the stochastic nature of LLMs, but **we** find minimal changes in performance.[3] When supported by the LLM, we set the temperature to 0, provide fixed random seeds, and limit the amount of reasoning tokens to 1024.

Table 2 shows the per-framework scores of the top 10 overall scoring models. While both thinking and non-thinking LLMs excel in synthetic environments, significant progress is still needed in AW, SW, and JERICHO, especially for non-thinking LLMs. However, more thinking is not always better. For both o3 and Claude-3.7-Sonnet, we see an increased thinking budget resulting in a lower overall score, with the non-thinking mode of Claude-3.7-Sonnet achieving the highest score in TWX.

## 5 ANALYSIS

In this section, we provide the results from a qualitative analysis of the game transcripts of the top LLMs for AW, SW, and ZORK1. This analysis is meant to supplement the automatic evaluation of the TALES score with empirical evidence of the importance of the core reasoning skills outlined in Section 2. We select AW and SW due to the lower overall performance in these environments compared to TW or TWX. We use ZORK1 as a representative sample of JERICHO due to its renown as one of the most famous and influential text-adventure games, and ability to challenge current state-of-the-art despite a high likelihood of data contamination.

Following our qualitative analysis of top LLM behaviors, we provide a short analysis of the strengths and weaknesses of the Claude-3.7-Sonnet thinking mode in contrast to the non-thinking mode. We focus on 3.7-Sonnet due to its performance in both modes as well as accessibility to the thinking traces. **In Appendix M we** take a representative subset of the game logs from all Claude models, explicitly labeling reasoning failures in a per-step granularity over 32 hand-annotated logs, **and investigate whether the 100 step limit for** TALES **poses an artificial limit through a trial of** ZORK1 **where the best performing models are allowed to explore for up to 1000 steps in Appendix N.**

### 5.1 REASONING FAILURE MODES IN ALFWORLD, SCIENCEWORLD, AND ZORK1

**LLM Agents Waste Steps With Weak Deductive Reasoning.** Agents with weaker deductive reasoning skills often waste many steps in undirected exploration. This manifests in interactions with distractor game elements and failures to have their commands understood by the environment action parser. In our system prompt, we explicitly tell the agent to use the 'help' command when stuck, which provides the agent with the set of action templates for the environment. We found that stronger models often immediately used the help action and leveraged the provided action templates to avoid wasting environment steps with invalid action phrases.

---

[3]We run 5 times for stability at 100 steps, though 3 runs may suffice if budget is limited.

**Most LLM Agents Can Inductively Reason On the Step-Level but Not the Trajectory-Level.** While the best agents could iteratively reason over step-level attempts, they failed to improve on past trajectories when resetting. On the step-level, we found agents displaying strong inductive reasoning skills were able to have their intent understood by using minor variations at each turn rather than repeating the exact same phrase. We found this to be a baseline competency required for agents to process through the environments, regardless of whether the help command was called. The top models displayed the ability to do this iteration methodology on the *step-level* but often failed to do so on the *trajectory-level*, repeating past mistakes instead of optimizing over previous attempts.

**Spatial and Grounded Reasoning Failures Often Result In and From Hallucination.** The strict causal-constraints of a situated environment allow for a unique lens to view what causes an LLM agent to hallucinate. Failures in these reasoning skills manifested through the LLM being unable to differentiate between what elements had appeared within the context and what elements were within the agent's current scope, or a failure to adjust to an error message from the environment, similar to Figure 1. While the top LLMs would not fabricate entirely new game elements, they would often attempt to directly interact with those mentioned previously in context but no longer in the agent's scope. This includes elements mentioned in the task description but not in observations.

## 5.2 To think or not to think? A Claude-case study

Due to overall performance and availability of the thinking traces, we analyze the game transcripts and thinking traces of Claude-3.7-Sonnet, comparing the model's performance with and without the ability to think. (White et al.) records a significant increase in performance for Claude-3.7-Sonnet with thinking enabled versus the base model. Despite this, we see an increase of less than 1% in TALES score for the thinking versus non-thinking modes of Claude-3.7-Sonnet.

**Thinking scaffolds different reasoning skills at different stages of progression.** We find Claude-3.7-Sonnet often exhibits one or more reasoning skills in its thinking traces throughout the progression of the game. Early, the primary reasoning skill explicitly exhibited is often deductive reasoning when identifying relevant game elements to interact with or areas to explore. Spatial reasoning appeared intermittently when a sub-goal was **completed and the agent had to navigate back to another location.** Later steps mainly focused on inductive reasoning to synthesize implicit knowledge gained from exploring the environment or grounded reasoning in evaluating the current state of the agent in the environment. In particular, this appeared to aid the reasoning agent in avoiding falling into the cyclical behavior patterns that appeared in even top models in more complex environments like SW or ZORK1.

**Thinking LLMs still fail when required to integrate multiple reasoning skills simultaneously.** Although Claude-3.7-Sonnet displayed all core reasoning skills individually at various points throughout the games, failures still **occurred** when multiple reasoning skills were required at key steps. Similar to the example shown in Figure 1, we see correct reasoning about one aspect of the current state only to neglect some other critical detail. In some cases when the missed detail leads to an immediate error, the LLM is capable of self-correcting. However, errors occurring later in the horizon often results in the agent being unable to identify and backtrack to the original point of failure.

# 6 Privileged or Domain Knowledge As Environment Modifications

(Liu et al., 2024) define an adaption module as "how test conditions are constructed", noting a need for fairness in evaluation condition across all objects of evaluation. **Providing excessive information during training and evaluation can lead to reported improvements of a method being unreliable when compared to baselines due to the scaffolding this information provides.** In TALES, we motivate a minimal adaption module to evaluate the **baseline** reasoning capabilities of LLM agents in complex, long-horizon, situated environments with text-adventure games. A number of other works explore individual frameworks within TALES, but all of these works, to our knowledge, introduce significant amounts of **privileged information** or domain knowledge through their implementation of the adaption module(Chang et al., 2024; Paglieri et al., 2024; Lu et al., 2025; Feng et al., 2025).

Table 3 illustrates common modifications to the adaption module. Admissible Actions are the actions the agent can take at any step, directly queried from the environment; Action Templates are those

Table 3: Zero-shot scores in TW for Qwen3-8B when different forms of **privileged or** domain knowledge are included in the prompt. Experiment settings are **kept the** same as main results outside of changes to **the** system prompt. We use Textworld as a zero-shot proxy, and perform more extensive RL fine-tuning experimentation below.

| In Prompt | Admissible Actions | Action Templates | Environment Dynamics | Textworld Score |
|---|---|---|---|---|
| BALROG | ✗ | ✓ | ✓ | 28% |
| AgentBoard | ✗ | ✓ | ✗ | 32% |
| IGE | ✗ | ✓ | ✓ | 28% |
| verl-agent | ✓ | ✗ | ✗ | 39% |
| TALES | ✗ | ✗ | ✗ | 18% |

otherwise provided by the `help` command; and Environment Dynamics refer to information such as **"The BBQ is for..., the key is for..."** that the agent would otherwise need to discover on its own. Each are problematic when taken as implicit knowledge the model should already have at the start rather than effective modifications to the environment via the adaption module.

## 6.1 PRIVILEGED AND DOMAIN INFORMATION

**We define "privileged information" as knowledge the agent has no access to within the scope of the environment, and "domain information" as knowledge the agent would otherwise need to discover through interaction and feedback. For example, using a ground-truth solution to perform reward assignment would be considered privileged information. If this solution were to be intentionally exposed in the environment for the agent to discover, it would be considered domain information. The per-step *Admissible Actions* are privileged information, given they are explicitly generated and verified by the framework. Within the scope of the environment itself, the model has no direct access to this set of commands.**

**While the action templates are accessible via the `help` command and environmental dynamics may be eventually discovered by the agent on its own, we argue this information should not be directly provided to the model at the start. The need to discover domain information through exploration represents one of the most challenging aspects of text-adventure games. Synthesizing this information requires strong inductive reasoning in parsing over game-play history and identifying the relationships between the agent, its actions, and feedback from the environment. While the argument could be made that this knowledge would be readily available in real-world applications, there are also many applications where the breadth of scope may make such pre-defined knowledge less useful. For example, trial and error debugging may require a coding agent to identify potential quirks about a code base purely through exploratory analysis. Similarly, we argue directly including domain knowledge in the prompt should be avoided, and explicitly acknowledged if done.**

## 6.2 IMPACT ON RL FINE-TUNING

**We evaluate the impact of directly providing privileged and domain information to the model during RL fine-tuning by ablating over the inclusion of the admissible actions and the contents of `help` in the prompt across the synthetic environments TW, TWX, AW, and SW. Given the model fails to find any success in SW, we exclude JERICHO which is even more challenging.**

**We use the verl-agent (Feng et al., 2025) framework due to their pre-existing results on AW. Integration of TALES is done standalone, rather than integrated into the existing verl-agent. Hyperparameters are primarily derived from their existing AW configuration and details from their appendix. We do not extensively sweep for ideal hyperparameters per environment as these results are meant to be a rudimentary baseline. See Appendix O for the exact hyperparameters used. Qwen3-8B(Team, 2025) is used due to its strong performance as a thinking-model. All hyperparameters are kept constant across experiments. Figure 2 shows the results, averaged over three seeds.**

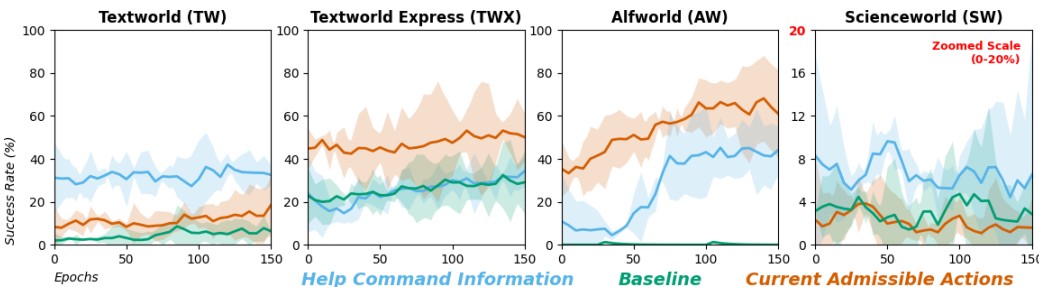

Figure 2: **Inclusion of admissible actions and the feedback from `help` in an agent's prompt during RL finetuning can result in a significant increase in performance over an unscaffolded baseline.** TWX shows no performance change due to `help` providing only the task description.

**Simpler environments can be made significantly easier by the inclusion of privileged information. The requirement to explore and iterate over what action commands are accepted by the game parser can be one of the most challenging aspects of an environment. The ability to do this iteration and keep track of the current state of the problem space tests a model's inductive, spatial, and grounded reasoning by forcing the choice of action to occur in the unbounded space of natural language. Bounding the space from which the agent must select its action to a subset of precomputed commands can drastically reduce the difficulty of the environment, as we can see by the extremely large differential in performance for TWX and AW when trained with and without admissible actions included in the prompt.**

**Training LLM Agents with privileged environment information should be avoided due to a failure to generalize. While capable of improving performance in simpler environments, leveraging privileged information as scaffolding can result in overfitting to the specified environment due to over-reliance. Taking admissible actions baseline as an example, we see significant improvements in TWX and AW. However, we fail to see a similar improvement in TW with performance even dropping in SW. These environment significantly feature actions that can leave the agent entirely unable to complete the given task. Without context, these actions can appear innocuous when embedded in the admissible actions. SW in particular has sufficient complexity such that the admissible actions can quickly grow combinatorially large, saturating the context with useless information. While not visible from a graph plotting success rate against epochs, computational cost on the environment can also become an issue[4].**

**Domain information can generalize as a strong deductive prior. With the exception of TWX, where `help` only repeats the task objective, we see improvements in all frameworks in the `help` baseline where the information included in the prompt is information the agent would be able to otherwise discover on its own. While this still decreases the difficulty of the environment, this otherwise accessible domain information acts as more natural guidance compared to the privileged information from the admissible actions.**

**LLM Agents face difficulty improving performance when exposed to a large number of diverse environments simultaneously. In the `help` and admissible actions baselines, we see an upward trend of success in AW, where all tasks involve relatively uniform task specifications. However, in TW, TWX, and SW, we see little to no learning occurring due to the extremely noisy reward signal from a wide range of task variety and difficulty.**

## 7 RELATED WORK

A large body of work exists in teaching agents to navigate and successfully complete text world games. We specifically divide this section into RL-Based agents, where the text-world is defined as a Partially Observable Markov Decision Process (POMDP) (Kaelbling et al., 1998) and LLM-based agents where information is fed to the LLM as an input with the output taken as an action.

---

[4]On a cluster of 8 H100s, the average wall-clock time for the training runs of SW with admissible actions was roughly 54 hours, with the baseline and `help`-baseline needing only an average of 22 hours.

**RL-Based:** Prior work explores text world games as benchmark for non-LLM-based agents (Narasimhan et al., 2015; Hausknecht et al., 2020). Due to the intractable action space of language, prior RL approaches used action templates to reduce the space **of** possible commands down to a subset learnable by an RL agent (Narasimhan et al., 2015; Ammanabrolu & Riedl, 2018; Yuan et al., 2018; Hausknecht et al., 2019; Ammanabrolu & Hausknecht, 2020; Ammanabrolu et al., 2020; Murugesan et al., 2021; Ryu et al., 2023). These agents are often augmented with a knowledge graph for better state tracking representation or for directing the agent (Ammanabrolu & Riedl, 2018; Hausknecht et al., 2019; Ammanabrolu & Hausknecht, 2020; Murugesan et al., 2021; Peng et al., 2023; Cui et al., 2023; 2024). Other approaches still use a base RL agent but use an LLM to guide the RL agent or generate diverse environments for generalization (Yao et al., 2020; Basavatia et al., 2024; Golchha et al., 2024).

**LLM-Based:** Early results demonstrated that even state-of-the-art pre-trained LLMs face difficulty when playing text-adventure games meant for human players (Tsai et al., 2023). Previous work has explored leveraging an external buffer or knowledge base to guide the agent (Shinn et al., 2023; Zhu et al., 2024). Other approaches leverage task decomposition into simpler sub-goals (Lin et al., 2023; Prasad et al., 2024). (Wang et al., 2024; Zhao et al., 2024) introduce approaches that leverage LLMs to modify the provided action space. Leveraging contrasting trajectories with LLMs to improve performance has also been explored (Song et al., 2024; Yang et al., 2024; Qiao et al., 2025).

## 8 CONCLUSION

In this work, we introduce TALES, a unified benchmark for LLM agents in text-adventure game environments. We identify a set of reasoning skills essential to agents operating through APIs to interface with outside environments.

We use SIMON SAYS to evaluate an agent's capability of the most basic composite reasoning needed to succeed in TALES. The game transcripts from leading LLMs reveal that, despite their impressive language capabilities, these models still struggle with core reasoning challenges inherent to text-adventure games. The difficulty stems not only from long-horizon dependencies and implicit environmental cues but also from the need for sequential, exploratory, and commonsense reasoning—skills that remain a bottleneck for even state-of-the-art LLMs.

We introduce baseline LLMs in our framework as canonical benchmarks for text-adventure environments without additional prompting or training **and provide a set of baseline RL ablations that demonstrate the effects of including privileged or domain information in the prompt**. Future work can improve upon these through supervised fine-tuning, in-context learning, or reinforcement learning. Thinking LLMs effectively leverage thinking traces; future research could guide these traces toward specific reasoning types for improved contextual awareness.

TALES supports game history truncation; studying how limited context scope affects LLM behavior and developing methods to condense game history by eliminating redundant information could enhance performance. While we manually analyze game logs to identify behaviors and failure modes, future work could automate this process using judge LLMs.

**Limitations.** Qualitative analysis was primarily performed by the authors due to familiarity with the included frameworks. However this may have resulted in implicit bias in the analysis due to this prior. The development of an automated method of annotation and cross-validation with an analysis from participants less familiar with the included frameworks may help further validate observations. We also cap the number of environmental steps for our work to 100. This cap allows for examination of an LLM's reasoning skills through their ability to perform directed exploration, but future work could explore extending this horizon and investigating what behaviors extremely long context elicit.

We view JERICHO as the ultimate test for an LLM agent's reasoning capabilities, however this is complicated by data contamination issues. As shown in Appendix R, we are almost certain modern LLMs were trained on ZORK1 playthrough transcripts, and this likely extends to many, if not all, games in the JERICHO suite. This raises the question of how much of the LLMs performance is influenced by the trajectories in its pre-training.

Overall, while progress has been made on synthetic text-adventure games, LLM-driven agents are still far from being able to complete games meant to be played for simple, human enjoyment.

## 9 REPRODUCIBILITY STATEMENT.

For the sake of anonymity, we provide an anonymized repository of all code used to gather the results presented in this work. This code will be released and made available to the public. We note that some of the close-sourced models may no longer be available at the time of reviewing or publication. Trajectories for these models can be provided upon request.

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

# A    ALL AGENT AVERAGE SCORES PER FRAMEWORK

In Table 4, we include the average scores per framework and average-per-game score of all LLM agents.

| Model | TEXTWORLD | TEXTWORLDEXPRESS | ALFWORLD | SCIENCEWORLD | JERICHO | Average Score |
|---|---|---|---|---|---|---|
| o3 (medium) | 100 | 91.9 | 88.3 | 93.0 | 15.7 | 58.7 |
| o3 (high) | 100 | 89.6 | 81.7 | 93.1 | 16.1 | 58.0 |
| gpt-5 (thinking) | 100 | 75.5 | 93.3 | 91.8 | 17.2 | 57.5 |
| o3 (low) | 99.1 | 89.8 | 70.0 | 88.3 | 14.2 | 54.8 |
| claude-3.7-sonnet (thinking) | 97.3 | 91.3 | 83.3 | 76.5 | 12.5 | 52.5 |
| claude-3.7-sonnet | 97.3 | 95.8 | 81.7 | 72.4 | 13.0 | 52.1 |
| claude-3.5-sonnet-latest | 95.5 | 81.6 | 75.0 | 82.3 | 9.6 | 50.4 |
| gpt-4.1 | 95.3 | 92.5 | 83.3 | 76.1 | 6.8 | 49.9 |
| gpt-5-mini (thinking) | 94.7 | 61.9 | 61.7 | 82.7 | 9.5 | 46.5 |
| o1 | 97.8 | 70.2 | 28.3 | 80.1 | 10.3 | 44.2 |
| gpt-4o | 83.6 | 80.6 | 56.7 | 61.4 | 5.6 | 40.6 |
| claude-3.5-haiku | 94.9 | 79.8 | 26.7 | 67.3 | 5.0 | 39.6 |
| Llama-3.1-405B-Instruct | 90.9 | 79.2 | 31.7 | 51.8 | 6.1 | 36.4 |
| gemini-2.0-flash | 80.8 | 76.1 | 20.0 | 57.1 | 5.4 | 35.0 |
| Qwen3-32B | 79.5 | 68.9 | 48.3 | 49.8 | 4.0 | 34.3 |
| Llama-3.3-70B-Instruct | 69.6 | 77.2 | 15.0 | 55.1 | 4.5 | 32.8 |
| Llama-3.1-70B-Instruct | 65.6 | 81.9 | 8.3 | 51.9 | 5.3 | 32.0 |
| Qwen2.5-72B-Instruct | 76.5 | 83.8 | 36.7 | 35.0 | 2.9 | 30.7 |
| Mistral-Large-Instruct-2407 | 82.4 | 68.3 | 6.7 | 46.1 | 5.8 | 30.3 |
| gpt-4.1-mini | 62.1 | 74.5 | 5.0 | 41.9 | 3.4 | 27.1 |
| gpt-4o-mini | 56.5 | 73.6 | 0.0 | 27.2 | 1.8 | 21.8 |
| Llama-4-Scout-17B-16E-Instruct | 41.1 | 68.4 | 0.0 | 27.0 | 1.8 | 19.8 |
| gpt-5-nano | 50.1 | 41.3 | 1.7 | 32.0 | 1.7 | 18.3 |
| Llama-4-Maverick-17B-128E-Instruct-FP8 | 43.5 | 56.1 | 8.3 | 11.5 | 2.0 | 15.5 |
| Mistral-Small-Instruct-2409 | 56.1 | 27.3 | 0.0 | 24.4 | 1.4 | 14.8 |
| Llama-3.1-8B-Instruct | 29.7 | 50.3 | 0.0 | 15.7 | 2.3 | 13.9 |
| DeepSeek-R1 | 37.1 | 38.6 | 0.0 | 15.8 | 1.0 | 12.4 |
| Qwen2.5-7B-Instruct | 27.7 | 45.6 | 0.0 | 12.6 | 0.7 | 11.7 |
| Llama-3.2-3B-Instruct | 21.4 | 42.0 | 0.0 | 10.0 | 1.5 | 10.4 |
| phi-4 | 20.8 | 43.8 | 0.0 | 8.9 | 1.6 | 10.3 |
| gpt-4.1-nano | 12.8 | 38.7 | 0.0 | 9.4 | 3.6 | 10.0 |
| Mistral-Small-24B-Instruct-2501 | 15.8 | 23.0 | 0.0 | 15.8 | 1.4 | 8.8 |
| DeepSeek-R1-Distill-Llama-70B | 8.7 | 39.8 | 0.0 | 7.7 | 1.3 | 8.4 |
| Ministral-8B-Instruct-2410 | 10.9 | 22.8 | 0.0 | 2.3 | 0.4 | 4.6 |
| Mistral-Small-3.1-24B-Instruct-2503 | 2.5 | 10.3 | 0.0 | 10.5 | 0.8 | 4.5 |
| Mixtral-8x22B-Instruct-v0.1 | 17.1 | 8.4 | 0.0 | 4.0 | 0.4 | 3.7 |
| Llama-3.2-1B-Instruct | 0.0 | 19.0 | 0.0 | 2.4 | 0.6 | 3.3 |
| Phi-3-mini-128k-instruct | 2.7 | 9.4 | 0.0 | 2.4 | 0.3 | 2.2 |
| Phi-3.5-MoE-instruct | 0.0 | 7.0 | 0.0 | 2.3 | 0.4 | 1.7 |
| Phi-4-mini-instruct | 0.0 | 5.5 | 0.0 | 2.3 | 0.5 | 1.5 |
| Mixtral-8x7B-Instruct-v0.1 | 0.0 | 1.6 | 0.0 | 4.0 | 0.3 | 1.3 |
| Phi-3.5-mini-instruct | 0.0 | 2.0 | 0.0 | 2.4 | 0.5 | 1.0 |
| Phi-3-medium-128k-instruct | 0.0 | 0.0 | 0.0 | 2.3 | 0.3 | 0.7 |

Table 4: Average scores per framework and total TALES score.

# B  JERICHO WALKTHROUGH SCORES

Table 5 shows the percent of achievable score when using the walkthrough for all JERICHO for 50, 100, 200, 300, 400, 500 and 1000 steps.

| Game | 50 Steps | 100 Steps | 200 Steps | 300 Steps | 400 Steps | 500 Steps | 1000 Steps |
|---|---|---|---|---|---|---|---|
| JerichoEnv905 | 100.000 | 100.000 | 100.000 | 100.000 | 100.000 | 100.000 | 100.000 |
| JerichoEnvAcorncourt | 100.000 | 100.000 | 100.000 | 100.000 | 100.000 | 100.000 | 100.000 |
| JerichoEnvAdvent | 26.300 | 42.600 | 63.100 | 100.000 | 100.000 | 100.000 | 100.000 |
| JerichoEnvAdventureland | 21.000 | 42.000 | 100.000 | 100.000 | 100.000 | 100.000 | 100.000 |
| JerichoEnvAfflicted | 46.700 | 100.000 | 100.000 | 100.000 | 100.000 | 100.000 | 100.000 |
| JerichoEnvAnchor | 5.000 | 11.000 | 29.000 | 41.000 | 52.000 | 64.000 | 99.000 |
| JerichoEnvAwaken | 60.000 | 100.000 | 100.000 | 100.000 | 100.000 | 100.000 | 100.000 |
| JerichoEnvBalances | 58.800 | 58.800 | 98.000 | 98.000 | 98.000 | 98.000 | 98.000 |
| JerichoEnvBallyhoo | 15.000 | 30.000 | 50.000 | 75.000 | 95.000 | 100.000 | 100.000 |
| JerichoEnvCurses | 3.800 | 5.600 | 12.700 | 28.200 | 38.200 | 47.500 | 81.800 |
| JerichoEnvCutthroat | 12.000 | 28.000 | 36.000 | 44.000 | 100.000 | 100.000 | 100.000 |
| JerichoEnvDeephome | 20.700 | 28.000 | 60.000 | 76.000 | 100.000 | 100.000 | 100.000 |
| JerichoEnvDetective | 100.000 | 100.000 | 100.000 | 100.000 | 100.000 | 100.000 | 100.000 |
| JerichoEnvDragon | 24.000 | 100.000 | 100.000 | 100.000 | 100.000 | 100.000 | 100.000 |
| JerichoEnvEnchanter | 11.300 | 31.200 | 70.000 | 100.000 | 100.000 | 100.000 | 100.000 |
| JerichoEnvEnter | 35.000 | 100.000 | 100.000 | 100.000 | 100.000 | 100.000 | 100.000 |
| JerichoEnvGold | 12.000 | 30.000 | 51.000 | 75.000 | 100.000 | 100.000 | 100.000 |
| JerichoEnvHhgg | 8.300 | 21.200 | 40.000 | 50.000 | 100.000 | 100.000 | 100.000 |
| JerichoEnvHuntdark | 0.000 | 100.000 | 100.000 | 100.000 | 100.000 | 100.000 | 100.000 |
| JerichoEnvInfidel | 12.500 | 20.000 | 70.000 | 100.000 | 100.000 | 100.000 | 100.000 |
| JerichoEnvInhumane | 33.300 | 77.800 | 100.000 | 100.000 | 100.000 | 100.000 | 100.000 |
| JerichoEnvJewel | 15.600 | 26.700 | 77.800 | 100.000 | 100.000 | 100.000 | 100.000 |
| JerichoEnvKarn | 5.900 | 23.500 | 38.200 | 67.600 | 100.000 | 100.000 | 100.000 |
| JerichoEnvLibrary | 100.000 | 100.000 | 100.000 | 100.000 | 100.000 | 100.000 | 100.000 |
| JerichoEnvLoose | 100.000 | 100.000 | 100.000 | 100.000 | 100.000 | 100.000 | 100.000 |
| JerichoEnvLostpig | 28.600 | 42.900 | 85.700 | 85.700 | 85.700 | 85.700 | 85.700 |
| JerichoEnvLudicorp | 13.300 | 25.300 | 58.700 | 92.700 | 100.000 | 100.000 | 100.000 |
| JerichoEnvLurking | 10.000 | 25.000 | 55.000 | 100.000 | 100.000 | 100.000 | 100.000 |
| JerichoEnvMoonlit | 0.000 | 100.000 | 100.000 | 100.000 | 100.000 | 100.000 | 100.000 |
| JerichoEnvMurdac | 6.800 | 18.000 | 18.000 | 48.000 | 99.600 | 99.600 | 99.600 |
| JerichoEnvNight | 60.000 | 100.000 | 100.000 | 100.000 | 100.000 | 100.000 | 100.000 |
| JerichoEnvOmniquest | 40.000 | 100.000 | 100.000 | 100.000 | 100.000 | 100.000 | 100.000 |
| JerichoEnvPartyfoul | 0.000 | 100.000 | 100.000 | 100.000 | 100.000 | 100.000 | 100.000 |
| JerichoEnvPentari | 100.000 | 100.000 | 100.000 | 100.000 | 100.000 | 100.000 | 100.000 |
| JerichoEnvPlanetfall | 7.500 | 26.300 | 35.000 | 60.000 | 100.000 | 100.000 | 100.000 |
| JerichoEnvPlundered | 16.000 | 44.000 | 100.000 | 100.000 | 100.000 | 100.000 | 100.000 |
| JerichoEnvReverb | 60.000 | 100.000 | 100.000 | 100.000 | 100.000 | 100.000 | 100.000 |
| JerichoEnvSeastalker | 28.000 | 44.000 | 90.000 | 100.000 | 100.000 | 100.000 | 100.000 |
| JerichoEnvSherlock | 23.000 | 37.000 | 55.000 | 84.000 | 100.000 | 100.000 | 100.000 |
| JerichoEnvSnacktime | 100.000 | 100.000 | 100.000 | 100.000 | 100.000 | 100.000 | 100.000 |
| JerichoEnvSorcerer | 23.700 | 37.500 | 53.700 | 100.000 | 100.000 | 100.000 | 100.000 |
| JerichoEnvSpellbrkr | 13.300 | 26.700 | 42.500 | 65.000 | 91.700 | 100.000 | 100.000 |
| JerichoEnvSpirit | 2.400 | 3.200 | 9.600 | 14.400 | 18.800 | 27.200 | 71.200 |
| JerichoEnvTemple | 28.600 | 57.100 | 100.000 | 100.000 | 100.000 | 100.000 | 100.000 |
| JerichoEnvTrinity | 15.000 | 22.000 | 32.000 | 47.000 | 58.000 | 78.000 | 100.000 |
| JerichoEnvTryst205 | 2.900 | 14.300 | 24.300 | 41.400 | 58.600 | 74.300 | 100.000 |
| JerichoEnvWeapon | 0.000 | 100.000 | 100.000 | 100.000 | 100.000 | 100.000 | 100.000 |
| JerichoEnvWishbringer | 24.000 | 50.000 | 100.000 | 100.000 | 100.000 | 100.000 | 100.000 |
| JerichoEnvYomomma | 25.700 | 97.100 | 97.100 | 97.100 | 97.100 | 97.100 | 97.100 |
| JerichoEnvZenon | 40.000 | 100.000 | 100.000 | 100.000 | 100.000 | 100.000 | 100.000 |
| JerichoEnvZork1 | 18.000 | 29.100 | 41.700 | 77.400 | 100.000 | 100.000 | 100.000 |
| JerichoEnvZork2 | 6.200 | 22.500 | 47.500 | 100.000 | 100.000 | 100.000 | 100.000 |
| JerichoEnvZork3 | 28.600 | 42.900 | 100.000 | 100.000 | 100.000 | 100.000 | 100.000 |
| JerichoEnvZtuu | 47.000 | 100.000 | 100.000 | 100.000 | 100.000 | 100.000 | 100.000 |

Table 5: Max score percentage reached by following the provided walkthrough for each JERICHO game.

# C  ALL AGENT AVERAGE FINAL TOKENS USED PER FRAMEWORK

In Table 6, we include the average final tokens used per game for each framework of all agents.

| Model | TEXTWORLD | TEXTWORLDEXPRESS | ALFWORLD | SCIENCEWORLD | JERICHO |
|---|---|---|---|---|---|
| o3 (medium) | 41342.0 | 107026.3 | 128740.7 | 92422.7 | 378168.5 |
| o3 (high) | 32148.6 | 51361.7 | 68275.0 | 51323.4 | 251760.8 |
| gpt-5 (thinking) | 38774.8 | 377563.2 | 145110.7 | 197999.9 | 649059.0 |
| o3 (low) | 51609.6 | 84218.5 | 119657.4 | 88557.9 | 318262.7 |
| claude-3.7-sonnet (thinking) | 69138.9 | 63072.8 | 74516.3 | 128752.7 | 311684.2 |
| claude-3.7-sonnet | 72072.9 | 46948.4 | 65131.1 | 152130.6 | 298533.1 |
| claude-3.5-sonnet-latest | 60766.1 | 68812.7 | 78765.4 | 106749.2 | 291950.3 |
| gpt-4.1 | 53378.9 | 46454.8 | 54107.6 | 86547.5 | 218123.2 |
| gpt-5-mini (thinking) | 151984.2 | 391444.6 | 508585.4 | 318636.6 | 878673.2 |
| o1 | 47765.6 | 113492.9 | 127460.2 | 74300.1 | 211958.7 |
| gpt-4o | 106863.0 | 49536.2 | 77946.0 | 107121.9 | 209712.8 |
| claude-3.5-haiku | 119839.9 | 85136.1 | 267643.6 | 205751.0 | 269458.9 |
| Llama-3.1-405B-Instruct | 66476.2 | 52624.6 | 106290.0 | 137657.2 | 226078.6 |
| gemini-2.0-flash | 142937.1 | 66075.5 | 138048.3 | 142883.3 | 230182.0 |
| Qwen3-32B | 198390.4 | 188065.3 | 190900.1 | 229708.1 | 374514.2 |
| Llama-3.3-70B-Instruct | 166373.8 | 70165.0 | 127348.7 | 128860.9 | 205362.8 |
| Llama-3.1-70B-Instruct | 133253.4 | 51885.9 | 106925.7 | 144615.6 | 210914.8 |
| Qwen2.5-72B-Instruct | 112658.0 | 52096.1 | 97211.2 | 168057.3 | 197628.3 |
| Mistral-Large-Instruct-2407 | 107788.5 | 110228.6 | 118395.4 | 163232.0 | 243256.0 |
| gpt-4.1-mini | 184516.3 | 92775.4 | 130758.2 | 125310.5 | 188824.3 |
| gpt-4o-mini | 159840.9 | 60210.7 | 145236.3 | 172875.7 | 182620.7 |
| Llama-4-Scout-17B-16E-Instruct | 289709.8 | 120173.9 | 172633.9 | 222464.1 | 229947.2 |
| gpt-5-nano | 770352.7 | 623055.1 | 821024.1 | 730904.7 | 825754.3 |
| Llama-4-Maverick-17B-128E-Instruct-FP8 | 287547.2 | 213139.5 | 354183.9 | 394875.5 | 372902.6 |
| Mistral-Small-Instruct-2409 | 163334.9 | 304510.9 | 107549.3 | 150730.7 | 208261.8 |
| Llama-3.1-8B-Instruct | 222239.7 | 358837.4 | 96582.5 | 152293.0 | 165505.8 |
| DeepSeek-R1 | 393654.5 | 398322.7 | 496328.4 | 431997.9 | 439399.3 |
| Qwen2.5-7B-Instruct | 143127.1 | 214926.3 | 91334.4 | 163021.2 | 171107.7 |
| Llama-3.2-3B-Instruct | 230950.5 | 79878.3 | 84620.4 | 195397.2 | 152544.7 |
| phi-4 | 189031.6 | 100363.9 | 126068.1 | 153395.2 | 178713.4 |
| gpt-4.1-nano | 545577.7 | 171767.5 | 277643.8 | 201505.9 | 182678.8 |
| Mistral-Small-24B-Instruct-2501 | 399093.8 | 500484.8 | 479125.0 | 418284.9 | 475649.7 |
| DeepSeek-R1-Distill-Llama-70B | 453695.7 | 637384.1 | 719404.2 | 482819.3 | 407401.8 |
| Ministral-8B-Instruct-2410 | 220157.9 | 337447.5 | 112710.5 | 108916.4 | 118104.9 |
| Mistral-Small-3.1-24B-Instruct-2503 | 448764.0 | 507986.4 | 477505.8 | 397054.6 | 514733.5 |
| Mixtral-8x22B-Instruct-v0.1 | 158782.2 | 137583.5 | 92832.7 | 134827.6 | 156515.8 |
| Llama-3.2-1B-Instruct | 567691.8 | 279214.8 | 457857.3 | 138285.6 | 201648.5 |
| Phi-3-mini-128k-instruct | 245215.0 | 429993.4 | 257852.2 | 253989.5 | 237881.5 |
| Phi-3.5-MoE-instruct | 274848.9 | 295190.9 | 240007.5 | 252055.6 | 271680.0 |
| Phi-4-mini-instruct | 231947.3 | 199299.1 | 195407.4 | 190887.4 | 212508.9 |
| Mixtral-8x7B-Instruct-v0.1 | 612791.9 | 555281.3 | 520434.6 | 560994.6 | 564967.6 |
| Phi-3.5-mini-instruct | 426125.5 | 476218.4 | 410459.6 | 327584.9 | 457434.4 |
| Phi-3-medium-128k-instruct | 620235.4 | 585925.5 | 581721.6 | 513787.5 | 595335.5 |

Table 6: Avg final tokens used per LLM per game for each framework. Ordering is based on the agent's cumulative average score shown in Table 4.

# D   AGENT SCORE STANDARD DEVIATIONS

In Table 7, we include the average standard deviation across seeds per framework of all LLM agents.

| Model | TEXTWORLD | TEXTWORLDEXPRESS | ALFWORLD | SCIENCEWORLD | JERICHO |
|---|---|---|---|---|---|
| o3 (medium) | 0.0 | 2.7 | 4.6 | 2.2 | 0.5 |
| o3 (high) | 0.0 | 3.2 | 9.1 | 1.1 | 1.3 |
| gpt-5 (thinking) | 0.0 | 5.5 | 7.0 | 2.5 | 1.1 |
| o3 (low) | 2.0 | 6.9 | 9.5 | 0.8 | 1.1 |
| claude-3.7-sonnet (thinking) | 2.8 | 4.7 | 10.2 | 2.9 | 0.9 |
| claude-3.7-sonnet | 0.0 | 1.4 | 3.7 | 3.7 | 1.1 |
| claude-3.5-sonnet-latest | 0.0 | 2.9 | 5.9 | 3.4 | 1.0 |
| gpt-4.1 | 2.6 | 1.9 | 11.8 | 2.3 | 0.8 |
| gpt-5-mini (thinking) | 3.6 | 2.8 | 9.5 | 6.1 | 1.4 |
| o1 | 1.2 | 4.4 | 4.6 | 5.0 | 1.7 |
| gpt-4o | 6.1 | 0.4 | 14.9 | 2.8 | 0.6 |
| claude-3.5-haiku | 5.3 | 0.0 | 3.7 | 2.6 | 0.6 |
| Llama-3.1-405B-Instruct | 5.0 | 4.9 | 10.9 | 4.5 | 0.5 |
| gemini-2.0-flash | 8.6 | 1.3 | 4.6 | 3.4 | 0.4 |
| Qwen3-32B | 6.8 | 1.9 | 10.9 | 3.2 | 0.4 |
| Llama-3.3-70B-Instruct | 2.8 | 3.4 | 3.7 | 2.3 | 0.1 |
| Llama-3.1-70B-Instruct | 3.5 | 1.9 | 5.9 | 4.5 | 0.2 |
| Qwen2.5-72B-Instruct | 2.0 | 2.5 | 4.6 | 3.8 | 0.7 |
| Mistral-Large-Instruct-2407 | 8.2 | 2.6 | 3.7 | 8.1 | 0.9 |
| gpt-4.1-mini | 6.1 | 1.7 | 7.5 | 3.6 | 0.3 |
| gpt-4o-mini | 5.4 | 1.7 | 0.0 | 1.5 | 0.2 |
| Llama-4-Scout-17B-16E-Instruct | 0.0 | 0.0 | 0.0 | 0.0 | 0.0 |
| gpt-5-nano | 7.7 | 5.1 | 3.7 | 4.1 | 0.3 |
| Llama-4-Maverick-17B-128E-Instruct-FP8 | 1.3 | 0.0 | 0.0 | 0.1 | 0.3 |
| Mistral-Small-Instruct-2409 | 5.1 | 0.0 | 0.0 | 2.2 | 0.0 |
| Llama-3.1-8B-Instruct | 4.7 | 2.9 | 0.0 | 0.9 | 0.1 |
| DeepSeek-R1 | 3.9 | 0.0 | 0.0 | 2.2 | 0.1 |
| Qwen2.5-7B-Instruct | 0.0 | 0.0 | 0.0 | 0.7 | 0.1 |
| Llama-3.2-3B-Instruct | 2.6 | 2.9 | 0.0 | 1.6 | 0.3 |
| phi-4 | 0.4 | 0.0 | 0.0 | 1.3 | 0.0 |
| gpt-4.1-nano | 2.1 | 4.5 | 0.0 | 1.0 | 2.4 |
| Mistral-Small-24B-Instruct-2501 | 3.1 | 1.0 | 0.0 | 1.1 | 0.3 |
| DeepSeek-R1-Distill-Llama-70B | 2.8 | 0.3 | 0.0 | 0.4 | 0.1 |
| Ministral-8B-Instruct-2410 | 4.2 | 0.0 | 0.0 | 0.0 | 0.0 |
| Mistral-Small-3.1-24B-Instruct-2503 | 0.0 | 0.0 | 0.0 | 0.3 | 0.0 |
| Mixtral-8x22B-Instruct-v0.1 | 3.0 | 2.3 | 0.0 | 1.7 | 0.1 |
| Llama-3.2-1B-Instruct | 0.0 | 0.0 | 0.0 | 0.0 | 0.0 |
| Phi-3-mini-128k-instruct | 2.0 | 0.0 | 0.0 | 0.3 | 0.0 |
| Phi-3.5-MoE-instruct | 0.0 | 2.7 | 0.0 | 0.0 | 0.1 |
| Phi-4-mini-instruct | 0.0 | 0.0 | 0.0 | 0.0 | 0.0 |
| Mixtral-8x7B-Instruct-v0.1 | 0.0 | 0.0 | 0.0 | 0.0 | 0.0 |
| Phi-3.5-mini-instruct | 0.0 | 1.1 | 0.0 | 0.1 | 0.0 |
| Phi-3-medium-128k-instruct | 0.0 | 0.0 | 0.0 | 0.0 | 0.0 |

Table 7: Standard deviation statistics for different LLMs Ordering is based on the agent's cumulative average score shown in Table 4.

# E  ALL GAMES

In Table 8 and Table 9 we list all tasks and games in their respective frameworks.

Table 8: Games Organized by Framework. Part 1.

**Jericho**

1. 905
2. Acorncourt
3. Advent
4. Adventureland
5. Afflicted
6. Anchor
7. Awaken
8. Balances
9. Ballyhoo
10. Curses
11. Cutthroat
12. Deephome
13. Detective
14. Dragon
15. Enchanter
16. Enter
17. Gold
18. Hhgg
19. Huntdark
20. Infidel
21. Inhumane
22. Jewel
23. Karn
24. Library
25. Loose
26. Lostpig
27. Ludicorp
28. Lurking
29. Moonlit
30. Murdac
31. Night
32. Omniquest
33. Partyfoul
34. Pentari
35. Planetfall
36. Plundered
37. Reverb
38. Seastalker
39. Sherlock
40. Snacktime
41. Sorcerer
42. Spellbrkr
43. Spirit
44. Temple
45. Theatre
46. Trinity
47. Tryst205
48. Weapon
49. Wishbringer
50. Yomomma
51. Zenon
52. Zork1
53. Zork2
54. Zork3
55. Ztuu

**ScienceWorld**

1. Boil
2. ChangeTheStateOfMatterOf
3. ChemistryMix
4. ChemistryMixPaintSecondaryColor
5. ChemistryMixPaintTertiaryColor
6. FindAnimal
7. FindLivingThing
8. FindNonLivingThing
9. FindPlant
10. Freeze
11. GrowFruit
12. GrowPlant
13. IdentifyLifeStages1
14. IdentifyLifeStages2
15. InclinedPlaneDetermineAngle
16. InclinedPlaneFrictionNamedSurfaces
17. InclinedPlaneFrictionUnnamedSurfaces
18. LifespanLongestLived
19. LifespanLongestLivedThenShortestLived
20. LifespanShortestLived
21. MeasureMeltingPointKnownSubstance
22. MeasureMeltingPointUnknownSubstance
23. Melt
24. MendelianGeneticsKnownPlant
25. MendelianGeneticsUnknownPlant
26. PowerComponent
27. PowerComponentRenewableVsNonrenewableEnergy
28. TestConductivity
29. TestConductivityOfUnknownSubstances
30. UseThermometer

Table 9: Games Organized by Framework. Part 2.

| **ALFWorld** | |
|---|---|
| 1. `LookAtObjInLightSeen` | 7. `PickCoolThenPlaceInRecepSeen` |
| 2. `LookAtObjInLightUnseen` | 8. `PickCoolThenPlaceInRecepUnseen` |
| 3. `PickAndPlaceSimpleSeen` | 9. `PickHeatThenPlaceInRecepSeen` |
| 4. `PickAndPlaceSimpleUnseen` | 10. `PickHeatThenPlaceInRecepUnseen` |
| 5. `PickCleanThenPlaceInRecepSeen` | 11. `PickTwoObjAndPlaceSeen` |
| 6. `PickCleanThenPlaceInRecepUnseen` | 12. `PickTwoObjAndPlaceUnseen` |

| **TextWorld** | |
|---|---|
| 1. `CookingLevel1` | 6. `CookingLevel6` |
| 2. `CookingLevel2` | 7. `CookingLevel7` |
| 3. `CookingLevel3` | 8. `CookingLevel8` |
| 4. `CookingLevel4` | 9. `CookingLevel9` |
| 5. `CookingLevel5` | 10. `CookingLevel10` |

| **TWX** | |
|---|---|
| 1. `Arithmetic` | 9. `SimonSaysWithMemory10` |
| 2. `CoinCollector` | 10. `SimonSaysWithMemory50` |
| 3. `CookingWorld` | 11. `SimonSaysWithMemory100` |
| 4. `MapReader` | 12. `SimonSaysWithMemory10Verbose` |
| 5. `PeckingOrder` | 13. `SimonSaysWithMemory50Verbose` |
| 6. `SimonSays10` | 14. `SimonSaysWithMemory100Verbose` |
| 7. `SimonSays50` | 15. `Sorting` |
| 8. `SimonSays100` | 16. `TextWorldCommonsense` |

# F  ALL SCORES PER GAME: TEXTWORLD

Table 10 shows the per-game scores of all models in TEXTWORLD across all seeds.

# G  ALL SCORES PER GAME: TEXTWORLDEXPRESS

Table 11 shows the average per-game scores of all models in TEXTWORLDEXPRESS across all seeds.

# H  ALL SCORES PER GAME: ALFWORLD

Table 12 shows the average per-game scores of all models in ALFWORLD across all seeds.

# I  ALL SCORES PER GAME: SCIENCEWORLD

Tables 13 and 14 shows the per-task average scores of all models in SCIENCEWORLD across all seeds.

# J  ALL SCORES PER GAME: JERICHO

Tables 15 and 16 shows the per-game scores of all models in JERICHO. * Indicates LLM has only been run on one seed. We will update the paper once all run seeds have been completed.

Table 10: Model Performance on TEXTWORLD Tasks.

| Models | CookingLevel1 | CookingLevel2 | CookingLevel3 | CookingLevel4 | CookingLevel5 | CookingLevel6 | CookingLevel7 | CookingLevel8 | CookingLevel9 | CookingLevel10 |
|---|---|---|---|---|---|---|---|---|---|---|
| o3 (medium) | 100.0 | 100.0 | 100.0 | 100.0 | 100.0 | 100.0 | 100.0 | 100.0 | 100.0 | 100.0 |
| o3 (high) | 100.0 | 100.0 | 100.0 | 100.0 | 100.0 | 100.0 | 100.0 | 100.0 | 100.0 | 100.0 |
| gpt-5 (thinking) | 100.0 | 100.0 | 100.0 | 100.0 | 100.0 | 100.0 | 100.0 | 100.0 | 100.0 | 100.0 |
| o3 (low) | 100.0 | 90.9 | 100.0 | 100.0 | 100.0 | 100.0 | 100.0 | 100.0 | 100.0 | 100.0 |
| claude-3.7-sonnet (thinking) | 100.0 | 72.7 | 100.0 | 100.0 | 100.0 | 100.0 | 100.0 | 100.0 | 100.0 | 100.0 |
| claude-3.7-sonnet | 100.0 | 72.7 | 100.0 | 100.0 | 100.0 | 100.0 | 100.0 | 100.0 | 100.0 | 100.0 |
| claude-3.5-sonnet-latest | 100.0 | 54.5 | 100.0 | 100.0 | 100.0 | 100.0 | 100.0 | 100.0 | 100.0 | 100.0 |
| gpt-4.1 | 100.0 | 61.8 | 100.0 | 100.0 | 100.0 | 100.0 | 100.0 | 100.0 | 100.0 | 90.9 |
| gpt-5-mini (thinking) | 100.0 | 47.3 | 100.0 | 100.0 | 100.0 | 100.0 | 100.0 | 100.0 | 100.0 | 100.0 |
| o1 | 100.0 | 78.2 | 100.0 | 100.0 | 100.0 | 100.0 | 100.0 | 100.0 | 100.0 | 100.0 |
| gpt-4o | 86.7 | 14.5 | 70.0 | 100.0 | 86.7 | 100.0 | 100.0 | 100.0 | 100.0 | 78.2 |
| claude-3.5-haiku | 100.0 | 58.2 | 100.0 | 100.0 | 100.0 | 100.0 | 100.0 | 100.0 | 100.0 | 90.9 |
| Llama-3.1-405B-Instruct | 100.0 | 54.5 | 85.0 | 100.0 | 100.0 | 100.0 | 100.0 | 100.0 | 100.0 | 69.1 |
| gemini-2.0-flash | 100.0 | 21.8 | 25.0 | 100.0 | 100.0 | 86.7 | 86.7 | 100.0 | 84.0 | 90.9 |
| Qwen3-32B | 100.0 | 27.3 | 85.0 | 100.0 | 100.0 | 46.7 | 100.0 | 88.0 | 68.0 | 81.8 |
| Llama-3.3-70B-Instruct | 100.0 | 0.0 | 25.0 | 100.0 | 100.0 | 0.0 | 100.0 | 100.0 | 92.0 | 90.9 |
| Llama-3.1-70B-Instruct | 100.0 | 0.0 | 25.0 | 100.0 | 100.0 | 0.0 | 100.0 | 100.0 | 84.0 | 47.3 |
| Qwen2.5-72B-Instruct | 100.0 | 0.0 | 25.0 | 100.0 | 86.7 | 100.0 | 100.0 | 100.0 | 100.0 | 40.0 |
| Mistral-Large-Instruct-2407 | 86.7 | 32.7 | 100.0 | 100.0 | 46.7 | 100.0 | 100.0 | 40.0 | 100.0 | 18.2 |
| gpt-4.1-mini | 60.0 | 5.5 | 70.0 | 100.0 | 46.7 | 86.7 | 100.0 | 52.0 | 76.0 | 36.4 |
| gpt-4o-mini | 100.0 | 0.0 | 25.0 | 100.0 | 100.0 | 100.0 | 0.0 | 40.0 | 100.0 | 41.8 |
| Llama-4-Scout-17B-16E-Instruct | 33.3 | 27.3 | 25.0 | 100.0 | 33.3 | 0.0 | 0.0 | 84.0 | 40.0 | 45.5 |
| gpt-5-nano | 60.0 | 12.7 | 85.0 | 100.0 | 20.0 | 0.0 | 60.0 | 32.0 | 20.0 | 45.5 |
| Llama-4-Maverick-17B-128E-Instruct-FP8 | 100.0 | 0.0 | 25.0 | 100.0 | 100.0 | 0.0 | 100.0 | 100.0 | 16.0 | 41.8 |
| Mistral-Small-Instruct-2409 | 100.0 | 0.0 | 25.0 | 0.0 | 100.0 | 20.0 | 0.0 | 28.0 | 76.0 | 40.0 |
| Llama-3.1-8B-Instruct | 73.3 | 1.8 | 25.0 | 0.0 | 0.0 | 0.0 | 0.0 | 28.0 | 60.0 | 9.1 |
| DeepSeek-R1 | 66.7 | 0.0 | 25.0 | 75.0 | 0.0 | 0.0 | 0.0 | 100.0 | 40.0 | 36.4 |
| Qwen2.5-7B-Instruct | 33.3 | 18.2 | 25.0 | 100.0 | 33.3 | 0.0 | 0.0 | 24.0 | 0.0 | 0.0 |
| Llama-3.2-3B-Instruct | 40.0 | 0.0 | 25.0 | 100.0 | 0.0 | 0.0 | 0.0 | 40.0 | 16.0 | 9.1 |
| phi-4 | 0.0 | 0.0 | 0.0 | 0.0 | 33.3 | 33.3 | 0.0 | 56.0 | 100.0 | 1.8 |
| gpt-4.1-nano | 33.3 | 0.0 | 25.0 | 0.0 | 0.0 | 0.0 | 13.3 | 20.0 | 0.0 | 0.0 |
| Mistral-Small-24B-Instruct-2501 | 46.7 | 0.0 | 25.0 | 25.0 | 33.3 | 0.0 | 0.0 | 0.0 | 8.0 | 0.0 |
| DeepSeek-R1-Distill-Llama-70B | 6.7 | 0.0 | 25.0 | 55.0 | 0.0 | 0.0 | 0.0 | 36.0 | 0.0 | 0.0 |
| Ministral-8B-Instruct-2410 | 33.3 | 0.0 | 40.0 | 0.0 | 0.0 | 0.0 | 0.0 | 0.0 | 0.0 | 0.0 |
| Mistral-Small-3.1-24B-Instruct-2503 | 0.0 | 0.0 | 25.0 | 100.0 | 0.0 | 0.0 | 0.0 | 64.0 | 0.0 | 0.0 |
| Mixtral-8x22B-Instruct-v0.1 | 6.7 | 0.0 | 0.0 | 0.0 | 0.0 | 0.0 | 0.0 | 0.0 | 0.0 | 0.0 |
| Llama-3.2-1B-Instruct | 0.0 | 0.0 | 10.0 | 5.0 | 0.0 | 0.0 | 0.0 | 12.0 | 0.0 | 0.0 |
| Phi-3-mini-128k-instruct | 0.0 | 0.0 | 0.0 | 0.0 | 0.0 | 0.0 | 0.0 | 0.0 | 0.0 | 0.0 |
| Phi-3.5-MoE-instruct | 0.0 | 0.0 | 0.0 | 0.0 | 0.0 | 0.0 | 0.0 | 0.0 | 0.0 | 0.0 |
| Phi-4-mini-instruct | 0.0 | 0.0 | 0.0 | 0.0 | 0.0 | 0.0 | 0.0 | 0.0 | 0.0 | 0.0 |
| Mixtral-8x7B-Instruct-v0.1 | 0.0 | 0.0 | 0.0 | 0.0 | 0.0 | 0.0 | 0.0 | 0.0 | 0.0 | 0.0 |
| Phi-3.5-mini-instruct | 0.0 | 0.0 | 0.0 | 0.0 | 0.0 | 0.0 | 0.0 | 0.0 | 0.0 | 0.0 |
| Phi-3-medium-128k-instruct | 0.0 | 0.0 | 0.0 | 0.0 | 0.0 | 0.0 | 0.0 | 0.0 | 0.0 | 0.0 |

Table 11: Model Performance on TextWorldExpress tasks.

| Models | Arithmetic | CoinCollector | CookingWorld | MapReader | PeckingOrder | SimonSays10 | SimonSays100 | SimonSays50 | SimonSaysWithMemory10 | SimonSaysWithMemory100 | SimonSaysWithMemory100Verbose | SimonSaysWithMemory10Verbose | SimonSaysWithMemory50 | SimonSaysWithMemory50Verbose | Sorting | TextWorldCommonsense |
|---|---|---|---|---|---|---|---|---|---|---|---|---|---|---|---|---|
| o3 (medium) | 100.0 | 100.0 | 76.8 | 100.0 | 100.0 | 100.0 | 100.0 | 100.0 | 100.0 | 51.4 | 51.6 | 100.0 | 94.0 | 97.2 | 100.0 | 100.0 |
| o3 (high) | 80.0 | 100.0 | 53.6 | 100.0 | 100.0 | 100.0 | 100.0 | 100.0 | 100.0 | 53.4 | 61.4 | 100.0 | 92.0 | 93.2 | 100.0 | 100.0 |
| gpt-5 (thinking) | 40.0 | 100.0 | 42.0 | 100.0 | 100.0 | 100.0 | 90.2 | 100.0 | 100.0 | 41.6 | 42.2 | 100.0 | 63.4 | 78.4 | 60.0 | 50.0 |
| o3 (low) | 100.0 | 100.0 | 45.0 | 100.0 | 100.0 | 100.0 | 100.0 | 100.0 | 100.0 | 64.4 | 87.0 | 100.0 | 100.0 | 100.0 | 60.0 | 80.0 |
| claude-3.7-sonnet (thinking) | 90.0 | 100.0 | 42.0 | 100.0 | 100.0 | 100.0 | 100.0 | 100.0 | 100.0 | 99.6 | 100.0 | 100.0 | 100.0 | 100.0 | 60.0 | 70.0 |
| claude-3.7-sonnet | 100.0 | 100.0 | 42.0 | 100.0 | 100.0 | 100.0 | 97.0 | 100.0 | 100.0 | 100.0 | 100.0 | 100.0 | 100.0 | 100.0 | 100.0 | 90.0 |
| claude-3.5-sonnet-latest | 20.0 | 100.0 | 39.2 | 100.0 | 100.0 | 100.0 | 100.0 | 100.0 | 100.0 | 100.0 | 100.0 | 100.0 | 100.0 | 100.0 | 0.0 | 50.0 |
| gpt-4.1 | 80.0 | 100.0 | 30.8 | 90.0 | 100.0 | 100.0 | 68.0 | 88.0 | 100.0 | 14.2 | 24.2 | 100.0 | 32.2 | 59.6 | 20.0 | 70.0 |
| gpt-5-mini (thinking) | 0.0 | 100.0 | 33.6 | 100.0 | 100.0 | 100.0 | 100.0 | 100.0 | 100.0 | 20.4 | 39.6 | 100.0 | 46.0 | 74.8 | 100.0 | 50.0 |
| o1 | 60.0 | 100.0 | 42.0 | 100.0 | 100.0 | 100.0 | 100.0 | 100.0 | 100.0 | 100.0 | 100.0 | 100.0 | 100.0 | 100.0 | 0.0 | 50.0 |
| gpt-4o | 0.0 | 100.0 | 39.2 | 100.0 | 100.0 | 100.0 | 100.0 | 100.0 | 100.0 | 99.0 | 99.0 | 100.0 | 100.0 | 100.0 | 0.0 | 50.0 |
| claude-3.5-haiku | 0.0 | 100.0 | 28.0 | 0.0 | 30.0 | 100.0 | 21.0 | 42.0 | 100.0 | 100.0 | 100.0 | 100.0 | 100.0 | 100.0 | 40.0 | 50.0 |
| Llama-3.1-405B-Instruct | 80.0 | 100.0 | 33.6 | 0.0 | 100.0 | 100.0 | 100.0 | 100.0 | 100.0 | 26.4 | 48.8 | 100.0 | 73.6 | 70.4 | 4.0 | 50.0 |
| gemini-2.0-flash | 30.0 | 100.0 | 33.6 | 50.0 | 60.0 | 100.0 | 100.0 | 100.0 | 100.0 | 100.0 | 100.0 | 100.0 | 86.4 | 100.0 | 12.0 | 50.0 |
| Qwen3-32B | 0.0 | 100.0 | 30.8 | 100.0 | 100.0 | 100.0 | 100.0 | 100.0 | 100.0 | 100.0 | 100.0 | 100.0 | 100.0 | 100.0 | 0.0 | 50.0 |
| Llama-3.3-70B-Instruct | 20.0 | 100.0 | 28.0 | 100.0 | 90.0 | 100.0 | 100.0 | 100.0 | 100.0 | 100.0 | 100.0 | 100.0 | 100.0 | 100.0 | 0.0 | 50.0 |
| Llama-3.1-70B-Instruct | 30.0 | 100.0 | 30.8 | 50.0 | 100.0 | 100.0 | 100.0 | 100.0 | 100.0 | 11.0 | 11.2 | 100.0 | 19.2 | 44.0 | 68.0 | 50.0 |
| Qwen2.5-72B-Instruct | 0.0 | 100.0 | 22.4 | 50.0 | 100.0 | 100.0 | 100.0 | 100.0 | 100.0 | 78.0 | 91.4 | 100.0 | 100.0 | 100.0 | 100.0 | 50.0 |
| Mistral-Large-Instruct-2407 | 20.0 | 100.0 | 42.0 | 0.0 | 30.0 | 100.0 | 100.0 | 100.0 | 100.0 | 98.2 | 99.6 | 100.0 | 100.0 | 100.0 | 0.0 | 50.0 |
| gpt-4.1-mini | 50.0 | 100.0 | 42.0 | 100.0 | 100.0 | 100.0 | 100.0 | 100.0 | 96.0 | 100.0 | 100.0 | 100.0 | 70.0 | 70.0 | 0.0 | 30.0 |
| gpt-4o-mini | 0.0 | 100.0 | 0.0 | 0.0 | 100.0 | 100.0 | 100.0 | 42.0 | 100.0 | 7.4 | 9.4 | 74.0 | 18.4 | 21.2 | 0.0 | 30.0 |
| Llama-4-Scout-17B-16E-Instruct | 0.0 | 100.0 | 42.0 | 0.0 | 60.0 | 100.0 | 21.0 | 86.8 | 100.0 | 13.0 | 5.0 | 100.0 | 26.0 | 6.0 | 0.0 | 30.0 |
| gpt-5-nano | 0.0 | 100.0 | 0.0 | 0.0 | 0.0 | 100.0 | 79.0 | 100.0 | 74.0 | 55.0 | 7.0 | 50.0 | 14.0 | 14.0 | 0.0 | 30.0 |
| Llama-4-Maverick-17B-128E-Instruct-FP8 | 0.0 | 100.0 | 42.0 | 0.0 | 100.0 | 100.0 | 100.0 | 42.0 | 100.0 | 21.0 | 1.0 | 100.0 | 2.0 | 2.0 | 36.0 | 50.0 |
| Mistral-Small-Instruct-2409 | 0.0 | 100.0 | 28.0 | 0.0 | 100.0 | 100.0 | 21.0 | 100.0 | 10.0 | 1.0 | 12.0 | 100.0 | 28.0 | 12.0 | 0.0 | 50.0 |
| Llama-3.1-8B-Instruct | 0.0 | 100.0 | 2.8 | 0.0 | 25.0 | 100.0 | 100.0 | 42.0 | 100.0 | 1.0 | 3.0 | 100.0 | 30.0 | 6.0 | 0.0 | 50.0 |
| DeepSeek-R1 | 0.0 | 100.0 | 42.0 | 50.0 | 85.0 | 100.0 | 4.0 | 100.0 | 10.0 | 15.0 | 33.0 | 100.0 | 100.0 | 100.0 | 0.0 | 50.0 |
| Qwen2.5-7B-Instruct | 0.0 | 100.0 | 0.0 | 0.0 | 0.0 | 0.0 | 0.0 | 8.0 | 100.0 | 47.6 | 100.0 | 30.0 | 100.0 | 100.0 | 0.0 | 50.0 |
| Llama-3.2-3B-Instruct | 0.0 | 80.0 | 14.0 | 0.0 | 100.0 | 100.0 | 21.0 | 42.0 | 100.0 | 100.0 | 100.0 | 100.0 | 23.2 | 80.4 | 0.0 | 0.0 |
| phi-4 | 0.0 | 100.0 | 0.0 | 0.0 | 45.0 | 100.0 | 5.0 | 10.0 | 10.0 | 1.0 | 1.0 | 40.0 | 10.0 | 8.8 | 0.0 | 0.0 |
| gpt-4.1-nano | 0.0 | 100.0 | 0.0 | 0.0 | 50.0 | 48.0 | 34.0 | 68.0 | 40.0 | 1.0 | 5.6 | 20.0 | 4.0 | 4.0 | 0.0 | 40.0 |
| Mistral-Small-24B-Instruct-2501 | 0.0 | 100.0 | 14.0 | 0.0 | 0.0 | 100.0 | 100.0 | 100.0 | 92.0 | 1.0 | 6.3 | 2.0 | 2.0 | 8.0 | 0.0 | 0.0 |
| DeepSeek-R1-Distill-Llama-70B | 0.0 | 0.0 | 28.0 | 0.0 | 0.0 | 100.0 | 17.0 | 4.0 | 10.0 | 1.0 | 4.2 | 2.0 | 2.0 | 4.0 | 0.0 | 50.0 |
| Ministral-8B-Instruct-2410 | 0.0 | 0.0 | 0.0 | 0.0 | 25.0 | 20.0 | 100.0 | 100.0 | 10.0 | 4.2 | 2.0 | 0.0 | 0.0 | 0.0 | 0.0 | 0.0 |
| Mistral-Small-3.1-24B-Instruct-2503 | 0.0 | 100.0 | 0.0 | 0.0 | 0.0 | 20.0 | 2.0 | 34.0 | 10.0 | 0.0 | 0.0 | 0.0 | 0.0 | 0.0 | 0.0 | 0.0 |
| Mixtral-8x22B-Instruct-v0.1 | 0.0 | 20.0 | 0.0 | 0.0 | 25.0 | 64.0 | 2.0 | 100.0 | 0.0 | 0.0 | 0.0 | 10.0 | 2.0 | 2.0 | 0.0 | 0.0 |
| Llama-3.2-1B-Instruct | 0.0 | 100.0 | 0.0 | 0.0 | 0.0 | 100.0 | 21.0 | 4.0 | 0.0 | 0.0 | 0.0 | 0.0 | 0.4 | 2.0 | 0.0 | 0.0 |
| Phi-3-mini-128k-instruct | 0.0 | 0.0 | 0.0 | 0.0 | 20.0 | 20.0 | 2.0 | 3.2 | 0.0 | 0.4 | 0.0 | 0.0 | 0.0 | 0.0 | 0.0 | 0.0 |
| Phi-3.5-MoE-instruct | 0.0 | 80.0 | 0.0 | 0.0 | 0.0 | 6.0 | 2.0 | 42.0 | 10.0 | 0.0 | 0.0 | 0.0 | 0.0 | 0.0 | 0.0 | 0.0 |
| Phi-4-mini-instruct | 0.0 | 0.0 | 0.0 | 0.0 | 0.0 | 0.0 | 0.8 | 4.0 | 0.0 | 0.0 | 0.0 | 0.0 | 0.0 | 0.0 | 0.0 | 0.0 |
| Mixtral-8x7B-Instruct-v0.1 | 0.0 | 0.0 | 0.0 | 0.0 | 0.0 | 20.0 | 0.0 | 3.2 | 0.0 | 0.0 | 0.0 | 0.0 | 0.0 | 0.0 | 0.0 | 0.0 |
| Phi-3.5-mini-instruct | 0.0 | 0.0 | 0.0 | 0.0 | 0.0 | 8.0 | 0.0 | 0.0 | 0.0 | 0.0 | 0.0 | 0.0 | 0.0 | 0.0 | 0.0 | 0.0 |
| Phi-3-medium-128K-instruct | 0.0 | 0.0 | 0.0 | 0.0 | 0.0 | 0.0 | 0.0 | 0.0 | 0.0 | 0.0 | 0.0 | 0.0 | 0.0 | 0.0 | 0.0 | 0.0 |

Table 12: Model Performance on ALFWORLD tasks.

| Models | LookAtObjInLightSeen | LookAtObjInLightUnseen | PickAndPlaceSimpleSeen | PickAndPlaceSimpleUnseen | PickCleanThenPlaceInRecepSeen | PickCleanThenPlaceInRecepUnseen | PickCoolThenPlaceInRecepSeen | PickCoolThenPlaceInRecepUnseen | PickHeatThenPlaceInRecepSeen | PickHeatThenPlaceInRecepUnseen | PickTwoObjAndPlaceSeen | PickTwoObjAndPlaceUnseen |
|---|---|---|---|---|---|---|---|---|---|---|---|---|
| o3 (medium) | 100.0 | 100.0 | 100.0 | 100.0 | 60.0 | 100.0 | 100.0 | 100.0 | 60.0 | 60.0 | 100.0 | 80.0 |
| o3 (high) | 60.0 | 100.0 | 100.0 | 100.0 | 20.0 | 60.0 | 80.0 | 80.0 | 100.0 | 80.0 | 100.0 | 100.0 |
| gpt-5 (thinking) | 100.0 | 100.0 | 100.0 | 100.0 | 100.0 | 100.0 | 80.0 | 100.0 | 80.0 | 60.0 | 100.0 | 100.0 |
| o3 (low) | 80.0 | 100.0 | 80.0 | 100.0 | 60.0 | 80.0 | 20.0 | 80.0 | 40.0 | 40.0 | 100.0 | 60.0 |
| claude-3.7-sonnet (thinking) | 20.0 | 100.0 | 100.0 | 100.0 | 80.0 | 100.0 | 100.0 | 100.0 | 60.0 | 80.0 | 80.0 | 80.0 |
| claude-3.7-sonnet | 0.0 | 20.0 | 100.0 | 100.0 | 100.0 | 100.0 | 100.0 | 100.0 | 80.0 | 0.0 | 100.0 | 100.0 |
| claude-3.5-sonnet-latest | 20.0 | 0.0 | 100.0 | 100.0 | 100.0 | 80.0 | 100.0 | 100.0 | 20.0 | 60.0 | 100.0 | 100.0 |
| gpt-4.1 | 80.0 | 0.0 | 100.0 | 100.0 | 20.0 | 80.0 | 80.0 | 60.0 | 80.0 | 40.0 | 100.0 | 100.0 |
| gpt-5-mini (thinking) | 100.0 | 100.0 | 40.0 | 20.0 | 60.0 | 40.0 | 0.0 | 40.0 | 40.0 | 0.0 | 0.0 | 0.0 |
| o1 | 100.0 | 100.0 | 60.0 | 100.0 | 0.0 | 60.0 | 40.0 | 0.0 | 60.0 | 40.0 | 60.0 | 80.0 |
| gpt-4o | 40.0 | 0.0 | 100.0 | 80.0 | 0.0 | 0.0 | 0.0 | 20.0 | 60.0 | 20.0 | 0.0 | 20.0 |
| claude-3.5-haiku | 100.0 | 0.0 | 100.0 | 80.0 | 100.0 | 60.0 | 0.0 | 0.0 | 0.0 | 0.0 | 100.0 | 20.0 |
| Llama-3.1-405B-Instruct | 80.0 | 0.0 | 100.0 | 40.0 | 0.0 | 0.0 | 0.0 | 0.0 | 0.0 | 0.0 | 0.0 | 0.0 |
| gemini-2.0-flash | 0.0 | 20.0 | 40.0 | 60.0 | 80.0 | 60.0 | 20.0 | 40.0 | 0.0 | 20.0 | 80.0 | 0.0 |
| Qwen3-32B | 80.0 | 0.0 | 60.0 | 0.0 | 0.0 | 0.0 | 0.0 | 0.0 | 40.0 | 0.0 | 0.0 | 0.0 |
| Llama-3.3-70B-Instruct | 40.0 | 0.0 | 100.0 | 40.0 | 0.0 | 20.0 | 0.0 | 0.0 | 0.0 | 0.0 | 0.0 | 0.0 |
| Llama-3.1-70B-Instruct | 100.0 | 0.0 | 0.0 | 0.0 | 0.0 | 0.0 | 0.0 | 0.0 | 0.0 | 0.0 | 60.0 | 0.0 |
| Qwen2.5-72B-Instruct | 20.0 | 0.0 | 0.0 | 0.0 | 0.0 | 80.0 | 0.0 | 0.0 | 0.0 | 0.0 | 20.0 | 0.0 |
| Mistral-Large-Instruct-2407 | 0.0 | 0.0 | 0.0 | 20.0 | 0.0 | 0.0 | 0.0 | 0.0 | 0.0 | 0.0 | 0.0 | 0.0 |
| gpt-4.1-mini | 0.0 | 0.0 | 0.0 | 0.0 | 0.0 | 0.0 | 0.0 | 0.0 | 0.0 | 0.0 | 0.0 | 0.0 |
| gpt-4o-mini | 0.0 | 0.0 | 0.0 | 0.0 | 0.0 | 0.0 | 0.0 | 0.0 | 0.0 | 0.0 | 0.0 | 0.0 |
| Llama-4-Scout-17B-16E-Instruct | 0.0 | 0.0 | 0.0 | 0.0 | 0.0 | 0.0 | 0.0 | 0.0 | 0.0 | 0.0 | 0.0 | 0.0 |
| gpt-5-nano | 0.0 | 0.0 | 0.0 | 0.0 | 0.0 | 0.0 | 0.0 | 0.0 | 0.0 | 0.0 | 0.0 | 0.0 |
| Llama-4-Maverick-17B-128E-Instruct-FP8 | 0.0 | 0.0 | 0.0 | 0.0 | 0.0 | 0.0 | 0.0 | 0.0 | 0.0 | 0.0 | 0.0 | 0.0 |
| Mistral-Small-Instruct-2409 | 0.0 | 0.0 | 0.0 | 0.0 | 0.0 | 0.0 | 0.0 | 0.0 | 0.0 | 0.0 | 0.0 | 0.0 |
| Llama-3.1-8B-Instruct | 0.0 | 0.0 | 0.0 | 0.0 | 0.0 | 0.0 | 0.0 | 0.0 | 0.0 | 0.0 | 0.0 | 0.0 |
| DeepSeek-R1 | 0.0 | 0.0 | 0.0 | 0.0 | 0.0 | 0.0 | 0.0 | 0.0 | 0.0 | 0.0 | 0.0 | 0.0 |
| Qwen2.5-7B-Instruct | 0.0 | 0.0 | 0.0 | 0.0 | 0.0 | 0.0 | 0.0 | 0.0 | 0.0 | 0.0 | 0.0 | 0.0 |
| Llama-3.2-3B-Instruct | 0.0 | 0.0 | 0.0 | 0.0 | 0.0 | 0.0 | 0.0 | 0.0 | 0.0 | 0.0 | 0.0 | 0.0 |
| phi-4 | 0.0 | 0.0 | 0.0 | 0.0 | 0.0 | 0.0 | 0.0 | 0.0 | 0.0 | 0.0 | 0.0 | 0.0 |
| gpt-4.1-nano | 0.0 | 0.0 | 0.0 | 0.0 | 0.0 | 0.0 | 0.0 | 0.0 | 0.0 | 0.0 | 0.0 | 0.0 |
| Mistral-Small-24B-Instruct-2501 | 0.0 | 0.0 | 0.0 | 0.0 | 0.0 | 0.0 | 0.0 | 0.0 | 0.0 | 0.0 | 0.0 | 0.0 |
| DeepSeek-R1-Distill-Llama-70B | 0.0 | 0.0 | 0.0 | 0.0 | 0.0 | 0.0 | 0.0 | 0.0 | 0.0 | 0.0 | 0.0 | 0.0 |
| Ministral-8B-Instruct-2410 | 0.0 | 0.0 | 0.0 | 0.0 | 0.0 | 0.0 | 0.0 | 0.0 | 0.0 | 0.0 | 0.0 | 0.0 |
| Mistral-Small-3.1-24B-Instruct-2503 | 0.0 | 0.0 | 0.0 | 0.0 | 0.0 | 0.0 | 0.0 | 0.0 | 0.0 | 0.0 | 0.0 | 0.0 |
| Mixtral-8x22B-Instruct-v0.1 | 0.0 | 0.0 | 0.0 | 0.0 | 0.0 | 0.0 | 0.0 | 0.0 | 0.0 | 0.0 | 0.0 | 0.0 |
| Llama-3.2-1B-Instruct | 0.0 | 0.0 | 0.0 | 0.0 | 0.0 | 0.0 | 0.0 | 0.0 | 0.0 | 0.0 | 0.0 | 0.0 |
| Phi-3-mini-128k-instruct | 0.0 | 0.0 | 0.0 | 0.0 | 0.0 | 0.0 | 0.0 | 0.0 | 0.0 | 0.0 | 0.0 | 0.0 |
| Phi-3.5-MoE-instruct | 0.0 | 0.0 | 0.0 | 0.0 | 0.0 | 0.0 | 0.0 | 0.0 | 0.0 | 0.0 | 0.0 | 0.0 |
| Phi-4-mini-instruct | 0.0 | 0.0 | 0.0 | 0.0 | 0.0 | 0.0 | 0.0 | 0.0 | 0.0 | 0.0 | 0.0 | 0.0 |
| Mixtral-8x7B-Instruct-v0.1 | 0.0 | 0.0 | 0.0 | 0.0 | 0.0 | 0.0 | 0.0 | 0.0 | 0.0 | 0.0 | 0.0 | 0.0 |
| Phi-3.5-mini-instruct | 0.0 | 0.0 | 0.0 | 0.0 | 0.0 | 0.0 | 0.0 | 0.0 | 0.0 | 0.0 | 0.0 | 0.0 |
| Phi-3-medium-128k-instruct | 0.0 | 0.0 | 0.0 | 0.0 | 0.0 | 0.0 | 0.0 | 0.0 | 0.0 | 0.0 | 0.0 | 0.0 |

Table 13: Model Performance on SCIENCEWORLD tasks. Part 1

| Models | Boil | ChangeTheStateOfMatterOf | ChemistryMix | ChemistryMixPaintSecondaryColor | ChemistryMixPaintTertiaryColor | FindAnimal | FindLivingThing | FindNonLivingThing | FindPlant | Freeze | GrowFruit | GrowPlant | IdentifyLifeStages1 | IdentifyLifeStages2 | InclinedPlaneDetermineAngle |
|---|---|---|---|---|---|---|---|---|---|---|---|---|---|---|---|
| o3 (medium) | 48.4 | 100.0 | 100.0 | 100.0 | 100.0 | 85.0 | 100.0 | 100.0 | 100.0 | 48.0 | 100.0 | 85.2 | 89.0 | 100.0 | 100.0 |
| o3 (high) | 68.0 | 100.0 | 100.0 | 100.0 | 100.0 | 88.2 | 100.0 | 100.0 | 100.0 | 49.0 | 87.4 | 92.2 | 78.0 | 100.0 | 100.0 |
| gpt-5 (thinking) | 95.6 | 100.0 | 100.0 | 100.0 | 100.0 | 88.2 | 100.0 | 100.0 | 100.0 | 53.0 | 44.4 | 77.4 | 65.4 | 65.4 | 100.0 |
| o3 (low) | 5.8 | 70.0 | 88.4 | 100.0 | 82.0 | 95.0 | 100.0 | 100.0 | 100.0 | 53.8 | 45.2 | 77.0 | 57.6 | 94.0 | 100.0 |
| claude-3.7-sonnet (thinking) | 5.8 | 43.2 | 100.0 | 100.0 | 94.0 | 85.0 | 100.0 | 100.0 | 40.0 | 45.0 | 75.0 | 48.4 | 50.4 | 54.6 | 90.0 |
| claude-3.7-sonnet | 5.0 | 89.4 | 100.0 | 100.0 | 76.0 | 66.6 | 90.0 | 100.0 | 48.4 | 50.0 | 40.4 | 100.0 | 95.0 | 64.6 | 100.0 |
| claude-3.5-sonnet-latest | 5.0 | 80.0 | 100.0 | 100.0 | 100.0 | 100.0 | 80.0 | 70.0 | 100.0 | 21.2 | 37.8 | 65.4 | 91.6 | 100.0 | 100.0 |
| gpt-4.1 | 52.4 | 43.2 | 100.0 | 100.0 | 60.0 | 45.0 | 100.0 | 81.6 | 100.0 | 48.0 | 48.8 | 62.6 | 78.0 | 45.2 | 100.0 |
| gpt-5-mini (thinking) | 21.4 | 80.8 | 75.0 | 100.0 | 52.2 | 65.0 | 96.6 | 100.0 | 100.0 | 37.4 | 72.4 | 78.2 | 71.2 | 70.6 | 100.0 |
| o1 | 2.8 | 2.6 | 33.0 | 28.0 | 82.2 | 91.6 | 90.0 | 100.0 | 70.0 | 0.0 | 29.8 | 62.4 | 35.0 | 28.4 | 82.0 |
| gpt-4o | 40.0 | 72.8 | 42.0 | 86.0 | 22.0 | 100.0 | 100.0 | 100.0 | 70.0 | 0.0 | 27.0 | 11.4 | 20.0 | 64.8 | 100.0 |
| claude-3.5-haiku | 3.2 | 2.2 | 42.0 | 86.0 | 76.6 | 58.2 | 50.0 | 100.0 | 70.0 | 17.6 | 18.4 | 34.0 | 40.4 | 46.8 | 100.0 |
| Llama-3.1-405B-Instruct | 6.6 | 80.8 | 88.4 | 60.0 | 20.0 | 45.0 | 50.0 | 85.0 | 40.0 | 40.4 | 21.0 | 11.6 | 12.0 | 12.0 | 81.0 |
| gemini-2.0-flash | 5.0 | 16.6 | 75.0 | 48.0 | 9.4 | 25.0 | 70.0 | 40.0 | 25.0 | 2.0 | 26.4 | 100.0 | 45.6 | 48.6 | 100.0 |
| Qwen3-32B | 9.2 | 1.8 | 33.0 | 62.0 | 45.4 | 25.0 | 85.0 | 100.0 | 25.0 | 0.0 | 34.6 | 13.0 | 19.4 | 37.8 | 100.0 |
| Llama-3.3-70B-Instruct | 1.6 | 2.0 | 8.0 | 74.0 | 53.4 | 50.0 | 5.0 | 100.0 | 100.0 | 0.0 | 41.2 | 10.8 | 65.0 | 35.0 | 100.0 |
| Llama-3.1-70B-Instruct | 0.0 | 0.0 | 73.2 | 100.0 | 70.8 | 25.0 | 70.0 | 100.0 | 70.0 | 0.0 | 23.4 | 9.0 | 50.0 | 20.0 | 60.0 |
| Qwen2.5-72B-Instruct | 2.4 | 16.6 | 48.2 | 90.0 | 10.0 | 36.6 | 35.0 | 85.0 | 55.0 | 0.0 | 13.2 | 10.0 | 20.0 | 20.0 | 56.0 |
| Mistral-Large-Instruct-2407 | 2.2 | 16.4 | 29.8 | 28.0 | 64.6 | 40.0 | 0.0 | 100.0 | 25.0 | 9.6 | 30.2 | 23.6 | 5.6 | 20.0 | 74.0 |
| gpt-4.1-mini | 0.8 | 0.8 | 33.0 | 20.0 | 42.0 | 40.0 | 25.0 | 63.4 | 10.0 | 0.0 | 35.4 | 6.6 | 22.0 | 47.8 | 5.0 |
| gpt-4o-mini | 0.0 | 0.0 | 33.0 | 88.0 | 9.4 | 16.8 | 21.8 | 83.0 | 0.0 | 0.0 | 20.8 | 7.0 | 4.0 | 28.4 | 5.0 |
| Llama-4-Scout-17B-16E-Instruct | 0.0 | 0.0 | 33.0 | 20.0 | 7.0 | 25.0 | 0.0 | 78.2 | 21.8 | 0.0 | 12.0 | 7.0 | 4.0 | 20.0 | 13.0 |
| gpt-5-nano | 1.8 | 0.8 | 43.2 | 72.0 | 60.0 | 23.4 | 25.0 | 8.0 | 0.0 | 0.0 | 32.0 | 7.0 | 15.4 | 30.2 | 10.0 |
| Llama-4-Maverick-17B-128E-Instruct-FP8 | 0.0 | 0.0 | 8.0 | 24.0 | 7.0 | 23.4 | 18.4 | 95.0 | 36.6 | 0.0 | 21.0 | 7.0 | 0.0 | 8.0 | 0.0 |
| Mistral-Small-Instruct-2409 | 0.0 | 0.4 | 33.0 | 20.0 | 8.8 | 25.0 | 21.8 | 48.2 | 10.0 | 0.0 | 13.4 | 7.4 | 4.0 | 9.6 | 3.0 |
| Llama-3.1-8B-Instruct | 0.4 | 0.0 | 29.8 | 22.0 | 19.6 | 20.0 | 4.8 | 53.4 | 23.4 | 0.0 | 29.2 | 7.0 | 0.0 | 14.4 | 5.0 |
| DeepSeek-R1 | 0.0 | 0.0 | 23.2 | 22.0 | 8.8 | 10.2 | 0.0 | 0.0 | 25.0 | 0.0 | 12.0 | 6.0 | 0.0 | 8.0 | 0.0 |
| Qwen2.5-7B-Instruct | 0.0 | 0.0 | 8.0 | 52.0 | 7.0 | 1.6 | 10.0 | 10.0 | 3.4 | 0.0 | 11.0 | 6.0 | 4.0 | 0.0 | 0.0 |
| Llama-3.2-3B-Instruct | 0.0 | 0.0 | 28.0 | 20.0 | 17.4 | 0.0 | 10.2 | 0.0 | 3.4 | 0.0 | 12.0 | 6.0 | 0.0 | 20.0 | 5.0 |
| phi-4 | 0.0 | 0.0 | 33.0 | 20.0 | 7.0 | 13.6 | 17.0 | 15.2 | 3.4 | 0.0 | 13.0 | 6.4 | 4.0 | 0.0 | 5.0 |
| gpt-4.1-nano | 0.0 | 0.0 | 8.0 | 22.0 | 7.0 | 0.0 | 13.6 | 17.0 | 8.4 | 0.0 | 12.0 | 6.0 | 2.4 | 14.4 | 3.0 |
| Mistral-Small-24B-Instruct-2501 | 0.0 | 0.0 | 8.0 | 20.0 | 8.2 | 5.0 | 0.0 | 10.2 | 0.0 | 0.0 | 12.0 | 6.0 | 0.8 | 14.4 | 5.0 |
| DeepSeek-R1-Distill-Llama-70B | 0.0 | 0.0 | 8.0 | 22.0 | 7.0 | 10.2 | 0.0 | 0.0 | 0.0 | 0.0 | 11.0 | 7.0 | 4.0 | 8.0 | 0.0 |
| Ministral-8B-Instruct-2410 | 0.0 | 0.0 | 11.4 | 20.0 | 7.0 | 0.0 | 0.0 | 0.0 | 0.0 | 0.0 | 11.0 | 6.0 | 0.0 | 0.0 | 0.0 |
| Mistral-Small-3.1-24B-Instruct-2503 | 0.0 | 0.0 | 8.0 | 36.0 | 7.0 | 10.0 | 0.0 | 0.0 | 15.0 | 0.0 | 11.0 | 6.0 | 0.0 | 0.0 | 0.0 |
| Mixtral-8x22B-Instruct-v0.1 | 0.0 | 0.0 | 11.4 | 20.0 | 7.0 | 0.0 | 0.0 | 0.0 | 0.0 | 0.0 | 12.0 | 6.0 | 0.0 | 8.0 | 8.0 |
| Llama-3.2-1B-Instruct | 0.0 | 0.0 | 8.0 | 20.0 | 7.0 | 0.0 | 0.0 | 0.0 | 0.0 | 0.0 | 11.0 | 6.8 | 0.0 | 6.4 | 0.0 |
| Phi-3-mini-128k-instruct | 0.0 | 0.0 | 8.0 | 20.0 | 7.0 | 0.0 | 0.0 | 0.0 | 0.0 | 0.0 | 11.0 | 6.0 | 0.0 | 0.0 | 0.0 |
| Phi-3.5-MoE-instruct | 0.0 | 0.0 | 8.0 | 20.0 | 7.0 | 0.0 | 0.0 | 0.0 | 0.0 | 0.0 | 11.0 | 6.0 | 0.0 | 0.0 | 0.0 |
| Phi-4-mini-instruct | 0.0 | 0.0 | 8.0 | 20.0 | 7.0 | 0.0 | 0.0 | 0.0 | 0.0 | 0.0 | 11.0 | 6.0 | 0.0 | 0.0 | 0.0 |
| Mixtral-8x7B-Instruct-v0.1 | 0.0 | 0.0 | 8.0 | 20.0 | 7.0 | 0.0 | 0.0 | 0.0 | 0.0 | 0.0 | 11.0 | 6.0 | 0.0 | 0.0 | 0.0 |
| Phi-3.5-mini-instruct | 0.0 | 0.0 | 8.0 | 20.0 | 7.0 | 1.6 | 0.0 | 0.0 | 0.0 | 0.0 | 11.0 | 6.0 | 0.0 | 0.0 | 0.0 |
| Phi-3-medium-128k-instruct | 0.0 | 0.0 | 8.0 | 20.0 | 7.0 | 0.0 | 0.0 | 0.0 | 0.0 | 0.0 | 11.0 | 6.0 | 0.0 | 0.0 | 0.0 |

Table 14: Model Performance on SCIENCEWORLD tasks. Part 2.

| Models | InclinedPlaneFrictionNamedSurfaces | InclinedPlaneFrictionUnnamedSurfaces | LifespanLongestLived | LifespanLongestLivedThenShortestLived | LifespanShortestLived | MeasureMeltingPointKnownSubstance | MeasureMeltingPointUnknownSubstance | Melt | MendelianGeneticsKnownPlant | MendelianGeneticsUnknownPlant | PowerComponent | PowerComponentRenewableVsNonrenewableEnergy | TestConductivity | TestConductivityOfUnknownSubstances | UseThermometer |
|---|---|---|---|---|---|---|---|---|---|---|---|---|---|---|---|
| o3 (medium) | 100.0 | 100.0 | 100.0 | 100.0 | 100.0 | 100.0 | 100.0 | 100.0 | 100.0 | 100.0 | 81.6 | 53.0 | 100.0 | 100.0 | 100.0 |
| o3 (high) | 100.0 | 100.0 | 100.0 | 100.0 | 100.0 | 100.0 | 100.0 | 100.0 | 100.0 | 100.0 | 77.0 | 53.0 | 100.0 | 100.0 | 100.0 |
| gpt-5 (thinking) | 100.0 | 100.0 | 100.0 | 100.0 | 100.0 | 100.0 | 100.0 | 100.0 | 100.0 | 100.0 | 77.0 | 53.0 | 100.0 | 100.0 | 100.0 |
| o3 (low) | 100.0 | 100.0 | 100.0 | 100.0 | 100.0 | 81.4 | 77.4 | 76.0 | 83.4 | 51.0 | 82.2 | 62.4 | 70.6 | 100.0 | 100.0 |
| claude-3.7-sonnet (thinking) | 100.0 | 100.0 | 100.0 | 100.0 | 100.0 | 76.0 | 80.6 | 62.0 | 19.0 | 14.6 | 71.6 | 33.2 | 64.4 | 64.4 | 100.0 |
| claude-3.7-sonnet | 82.0 | 100.0 | 100.0 | 100.0 | 100.0 | 73.0 | 90.0 | 81.4 | 83.4 | 50.6 | 91.4 | 71.8 | 100.0 | 37.0 | 100.0 |
| claude-3.5-sonnet-latest | 100.0 | 100.0 | 100.0 | 100.0 | 100.0 | 100.0 | 100.0 | 30.8 | 100.0 | 50.6 | 71.4 | 43.2 | 100.0 | 84.8 | 100.0 |
| gpt-4.1 | 100.0 | 100.0 | 100.0 | 100.0 | 100.0 | 45.0 | 95.4 | 43.4 | 60.4 | 100.0 | 90.8 | 43.0 | 100.0 | 57.2 | 100.0 |
| gpt-5-mini (thinking) | 100.0 | 100.0 | 100.0 | 100.0 | 100.0 | 100.0 | 100.0 | 100.0 | 100.0 | 3.0 | 69.0 | 71.8 | 100.0 | 100.0 | 100.0 |
| o1 | 100.0 | 100.0 | 100.0 | 100.0 | 100.0 | 78.4 | 72.2 | 3.0 | 100.0 | 82.2 | 58.2 | 23.6 | 91.0 | 50.6 | 84.4 |
| gpt-4o | 84.4 | 100.0 | 100.0 | 100.0 | 90.0 | 100.0 | 89.2 | 59.6 | 47.8 | 13.6 | 57.0 | 38.0 | 100.0 | 61.8 | 84.4 |
| claude-3.5-haiku | 100.0 | 100.0 | 80.0 | 100.0 | 100.0 | 6.0 | 9.0 | 4.0 | 83.2 | 100.0 | 33.0 | 28.8 | 17.0 | 58.8 | 100.0 |
| Llama-3.1-405b-Instruct | 81.0 | 100.0 | 100.0 | 100.0 | 100.0 | 81.8 | 90.6 | 5.0 | 66.0 | 28.8 | 86.2 | 11.6 | 7.0 | 39.4 | 37.6 |
| gemini-2.0-flash | 74.0 | 30.0 | 100.0 | 73.2 | 80.0 | 81.6 | 7.4 | 22.0 | 100.0 | 100.0 | 27.0 | 31.8 | 8.0 | 58.0 | 97.8 |
| Qwen3-32B | 100.0 | 100.0 | 100.0 | 53.0 | 100.0 | 62.6 | 34.6 | 4.4 | 100.0 | 65.2 | 27.0 | 5.0 | 25.0 | 10.0 | 42.8 |
| Llama-3.3-70B-Instruct | 80.0 | 100.0 | 100.0 | 33.0 | 100.0 | 0.0 | 72.6 | 2.6 | 100.0 | 11.0 | 47.0 | 38.0 | 8.6 | 12.8 | 68.2 |
| Llama-3.1-70b-Instruct | 80.0 | 60.0 | 0.0 | 0.0 | 0.0 | 100.0 | 10.0 | 2.6 | 65.2 | 31.0 | 57.0 | 38.0 | 100.0 | 64.0 | 84.4 |
| Qwen2.5-72B-Instruct | 4.0 | 2.0 | 100.0 | 96.6 | 100.0 | 28.0 | 70.4 | 2.6 | 64.8 | 64.8 | 21.0 | 5.0 | 65.8 | 64.4 | 76.4 |
| Mistral-Large-Instruct-2407 | 35.0 | 70.0 | 70.0 | 100.0 | 90.0 | 73.0 | 34.6 | 1.0 | 13.0 | 3.4 | 23.6 | 10.0 | 15.4 | 48.0 | 28.6 |
| gpt-4.1-mini | 1.0 | 18.0 | 100.0 | 100.0 | 100.0 | 73.0 | 73.0 | 2.0 | 3.4 | 3.4 | 7.0 | 5.0 | 5.0 | 81.0 | 45.8 |
| gpt-4o-mini | 0.0 | 3.0 | 100.0 | 33.0 | 50.0 | 80.2 | 30.0 | 1.0 | 1.0 | 1.0 | 57.0 | 38.0 | 81.0 | 60.0 | 75.4 |
| Llama-4-Scout-17B-16E-Instruct | 15.0 | 15.0 | 90.0 | 53.0 | 70.0 | 6.0 | 1.0 | 2.0 | 0.0 | 1.0 | 58.2 | 31.4 | 5.0 | 8.0 | 0.0 |
| gpt-5-nano | 3.0 | 5.0 | 50.0 | 33.0 | 50.0 | 80.4 | 73.2 | 1.2 | 0.0 | 1.0 | 57.0 | 38.0 | 5.0 | 10.0 | 18.2 |
| Llama-4-Maverick-17B-128E-Instruct-FP8 | 5.0 | 0.0 | 50.0 | 33.0 | 50.0 | 4.0 | 31.0 | 0.0 | 2.6 | 100.0 | 57.0 | 38.0 | 5.0 | 0.0 | 8.0 |
| Mistral-Small-Instruct-2409 | 5.0 | 43.0 | 40.0 | 0.0 | 50.0 | 24.8 | 73.2 | 0.0 | 0.0 | 11.0 | 7.0 | 5.0 | 5.0 | 0.0 | 3.0 |
| Llama-3.1-8B-Instruct | 0.0 | 0.0 | 50.0 | 100.0 | 0.0 | 0.0 | 4.0 | 0.0 | 0.0 | 0.0 | 7.0 | 5.0 | 55.0 | 10.0 | 9.2 |
| DeepSeek-R1 | 0.0 | 3.0 | 20.0 | 33.0 | 50.0 | 24.8 | 1.0 | 0.0 | 0.0 | 1.0 | 57.0 | 24.8 | 5.0 | 5.0 | 8.8 |
| Qwen2.5-7B-Instruct | 5.0 | 0.0 | 50.0 | 33.0 | 0.0 | 2.4 | 1.0 | 1.2 | 1.0 | 1.0 | 7.0 | 18.2 | 55.0 | 5.0 | 1.8 |
| Llama-3.2-3B-Instruct | 0.0 | 5.0 | 20.0 | 0.0 | 0.0 | 5.0 | 2.4 | 0.0 | 0.0 | 0.0 | 7.0 | 5.0 | 5.0 | 5.0 | 3.0 |
| phi-4 | 0.0 | 5.0 | 50.0 | 33.0 | 50.0 | 6.0 | 1.0 | 0.0 | 0.0 | 2.6 | 7.0 | 5.0 | 5.0 | 5.0 | 0.6 |
| gpt-4.1-nano | 0.0 | 4.0 | 100.0 | 59.8 | 50.0 | 6.0 | 1.8 | 2.0 | 0.0 | 1.0 | 57.0 | 5.0 | 5.0 | 4.0 | 40.0 |
| Mistral-Small-24B-Instruct-2501 | 0.0 | 5.0 | 0.0 | 33.0 | 0.0 | 6.0 | 1.0 | 0.0 | 0.0 | 1.0 | 7.0 | 5.0 | 5.0 | 5.0 | 49.0 |
| DeepSeek-R1-Distill-Llama-70B | 0.0 | 5.0 | 0.0 | 0.0 | 0.0 | 0.0 | 0.0 | 2.0 | 0.0 | 0.0 | 7.0 | 5.0 | 5.0 | 0.0 | 0.0 |
| Ministral-8B-Instruct-2410 | 0.0 | 0.0 | 0.0 | 33.0 | 50.0 | 0.0 | 1.0 | 0.0 | 0.0 | 0.0 | 57.0 | 11.0 | 5.0 | 0.0 | 0.0 |
| Mistral-Small-3.1-24B-Instruct-2503 | 0.0 | 0.0 | 100.0 | 0.0 | 0.0 | 0.0 | 1.0 | 1.2 | 0.0 | 0.0 | 7.0 | 5.0 | 5.0 | 0.0 | 0.0 |
| Mixtral-8x22B-Instruct-v0.1 | 0.0 | 0.0 | 0.0 | 0.0 | 0.0 | 0.0 | 0.0 | 0.0 | 0.0 | 0.0 | 7.0 | 5.0 | 5.0 | 5.0 | 0.0 |
| Llama-3.2-1B-Instruct | 0.0 | 0.0 | 0.0 | 0.0 | 0.0 | 0.0 | 0.0 | 0.0 | 0.0 | 0.0 | 7.0 | 5.0 | 5.0 | 5.0 | 0.0 |
| Phi-3-mini-128k-instruct | 0.0 | 0.0 | 0.0 | 0.0 | 0.0 | 0.0 | 0.0 | 0.0 | 0.0 | 0.0 | 7.0 | 5.0 | 5.0 | 0.0 | 0.0 |
| Phi-3.5-MoE-instruct | 0.0 | 0.0 | 0.0 | 0.0 | 0.0 | 0.0 | 0.0 | 0.0 | 0.0 | 0.0 | 7.0 | 5.0 | 5.0 | 0.0 | 0.0 |
| Phi-4-mini-instruct | 0.0 | 0.0 | 0.0 | 0.0 | 0.0 | 0.0 | 0.0 | 0.0 | 0.0 | 0.0 | 7.0 | 5.0 | 55.0 | 0.0 | 0.0 |
| Mixtral-8x7B-Instruct-v0.1 | 0.0 | 0.0 | 0.0 | 0.0 | 0.0 | 0.0 | 0.0 | 0.0 | 0.0 | 0.0 | 7.0 | 5.0 | 5.0 | 0.0 | 0.0 |
| Phi-3.5-mini-instruct | 0.0 | 0.0 | 0.0 | 0.0 | 0.0 | 0.0 | 0.0 | 0.0 | 0.0 | 0.0 | 7.0 | 5.0 | 5.0 | 0.0 | 0.0 |
| Phi-3-medium-128k-instruct | 0.0 | 0.0 | 0.0 | 0.0 | 0.0 | 0.0 | 0.0 | 0.0 | 0.0 | 0.0 | 7.0 | 5.0 | 5.0 | 0.0 | 0.0 |

Table 15: Model Performance on JERICHO games(tasks), part 1.

| Models | 905 | Acorncourt | Advent | Adventureland | Afflicted | Anchor | Awaken | Balances | Ballyhoo | Curses | Cuthroat | Deephome | Detective | Dragon | Enchanter | Enter | Gold | Hhgg | Huntdark | Infidel | Inhumane | Jewel | Karn | Library | Loose | Lostpig | Ludicorp |
|---|---|---|---|---|---|---|---|---|---|---|---|---|---|---|---|---|---|---|---|---|---|---|---|---|---|---|---|
| o3 (medium) | 100.0 | 100.0 | 20.6 | 11.2 | 18.9 | 1.6 | 0.0 | 23.5 | 1.0 | 0.8 | 13.6 | 7.0 | 33.3 | 4.8 | 12.3 | 72.0 | 10.8 | 3.2 | 0.0 | 1.2 | 37.8 | 0.9 | 2.4 | 33.3 | 6.0 | 25.7 | 12.0 |
| o3 (high) | 100.0 | 100.0 | 20.7 | 14.0 | 17.3 | 2.0 | 0.0 | 27.5 | 2.0 | 0.7 | 13.6 | 10.0 | 54.4 | 6.4 | 6.0 | 74.0 | 9.6 | 3.2 | 0.0 | 1.2 | 28.9 | 0.4 | 4.7 | 40.7 | 7.6 | 31.4 | 12.3 |
| gpt-5 (thinking) | 100.0 | 100.0 | 27.7 | 9.8 | 26.4 | 2.4 | 0.0 | 27.5 | 2.0 | 1.1 | 12.0 | 10.3 | 38.3 | 4.8 | 12.8 | 65.0 | 12.0 | 2.5 | 0.0 | 1.2 | 35.6 | 1.1 | 5.3 | 39.3 | 6.0 | 40.0 | 11.2 |
| o3 (low) | 80.0 | 40.0 | 23.2 | 4.2 | 19.2 | 1.2 | 0.0 | 29.4 | 2.0 | 0.5 | 16.8 | 13.8 | 36.7 | 4.0 | 13.8 | 72.0 | 6.0 | 2.5 | 0.0 | 0.8 | 42.2 | 1.8 | 2.9 | 26.7 | 3.6 | 22.9 | 10.3 |
| claude-3-7-sonnet (thinking) | 40.0 | 26.7 | 17.6 | 1.4 | 36.0 | 1.8 | 4.0 | 25.5 | 0.0 | 0.0 | 12.0 | 8.1 | 76.7 | 4.8 | 13.2 | 58.0 | 0.0 | 2.5 | 0.0 | 0.5 | 31.1 | 0.0 | 1.8 | 6.7 | 0.0 | 20.0 | 8.8 |
| claude-3-7-sonnet | 40.0 | 6.7 | 19.4 | 0.0 | 35.5 | 0.4 | 0.0 | 39.2 | 0.0 | 0.1 | 20.0 | 9.0 | 88.9 | 4.0 | 12.8 | 66.0 | 6.0 | 2.5 | 0.0 | 1.0 | 40.0 | 0.0 | 2.9 | 33.3 | 6.8 | 28.6 | 6.7 |
| claude-3-5-sonnet-latest | 0.0 | 33.3 | 12.1 | 5.6 | 15.2 | 0.4 | 0.0 | 25.9 | 0.0 | 0.3 | 8.8 | 8.8 | 67.2 | 4.0 | 5.8 | 75.0 | 9.0 | 2.5 | 0.0 | 0.2 | 37.8 | 0.9 | 3.5 | 33.3 | 0.0 | 2.9 | 8.1 |
| gpt4.1 | 40.0 | 0.0 | 10.3 | 0.0 | 4.3 | 0.4 | 0.0 | 23.5 | 2.0 | 0.2 | 5.6 | 5.9 | 65.6 | 5.6 | 9.8 | 39.0 | 1.2 | 2.5 | 0.0 | 0.5 | 4.4 | 0.4 | 0.0 | 13.3 | 4.0 | 2.9 | 8.7 |
| gpt-5-mini (thinking) | 80.0 | 20.0 | 13.9 | 7.0 | 5.1 | 0.4 | 0.0 | 19.6 | 1.0 | 0.0 | 9.6 | 9.3 | 31.1 | 7.2 | 12.5 | 58.0 | 7.8 | 2.5 | 0.0 | 1.0 | 11.1 | 0.0 | 1.8 | 11.3 | 0.8 | 25.7 | 7.9 |
| o1 | 0.0 | 66.7 | 10.3 | 1.4 | 1.1 | 0.4 | 0.0 | 23.9 | 0.0 | 0.0 | 0.0 | 6.7 | 27.2 | 4.8 | 4.2 | 22.0 | 1.2 | 2.5 | 0.0 | 1.2 | 4.4 | 0.9 | 1.8 | 34.7 | 0.8 | 14.3 | 8.3 |
| gpt-4o | 0.0 | 0.0 | 10.3 | 0.0 | 1.1 | 0.4 | 0.0 | 13.7 | 0.0 | 0.0 | 2.4 | 6.3 | 34.4 | 7.2 | 3.2 | 38.0 | 7.8 | 2.5 | 0.0 | 1.0 | 4.4 | 0.4 | 1.2 | 13.3 | 0.0 | 5.7 | 7.9 |
| claude-3-5-haiku | 0.0 | 20.0 | 11.7 | 0.0 | 4.5 | 0.8 | 0.0 | 19.6 | 0.0 | 0.0 | 5.6 | 5.7 | 25.0 | 4.0 | 4.0 | 23.0 | 0.0 | 1.0 | 0.0 | 1.2 | 0.0 | 0.4 | 0.0 | 0.0 | 0.8 | 11.4 | 0.8 |
| Llama-3.1-405B-Instruct | 0.0 | 0.0 | 10.3 | 0.0 | 0.0 | 0.4 | 0.0 | 19.6 | 0.0 | 0.0 | 0.0 | 5.6 | 50.0 | 0.8 | 4.0 | 15.0 | 1.8 | 2.5 | 0.0 | 1.2 | 0.0 | 0.0 | 0.6 | 20.0 | 0.0 | 14.3 | 2.5 |
| gemini-2.0-flash | 0.0 | 0.0 | 10.3 | 0.0 | 3.5 | 0.8 | 0.0 | 11.8 | 2.0 | 0.0 | 1.6 | 5.1 | 28.3 | 4.0 | 3.8 | 51.0 | 0.0 | 1.0 | 0.0 | 0.0 | 0.0 | 0.4 | 2.4 | 3.3 | 0.0 | 0.0 | 2.5 |
| Qwen3-32B | 0.0 | 6.7 | 10.3 | 0.0 | 0.0 | 0.4 | 0.0 | 5.9 | 0.0 | 0.0 | 0.0 | 4.5 | 31.1 | 3.2 | 0.0 | 3.0 | 1.8 | 2.5 | 0.0 | 0.0 | 0.0 | 0.0 | 0.6 | 0.0 | 4.0 | 11.4 | 2.9 |
| Llama-3.3-70B-Instruct | 0.0 | 0.0 | 10.3 | 0.0 | 0.8 | 1.2 | 0.0 | 11.8 | 0.0 | 0.0 | 0.0 | 5.3 | 36.1 | 4.8 | 0.0 | 23.0 | 0.0 | 0.0 | 0.0 | 0.5 | 0.0 | 0.0 | 0.0 | 16.7 | 0.0 | 14.3 | 3.2 |
| Llama-3.1-70B-Instruct | 0.0 | 0.0 | 10.3 | 0.0 | 0.8 | 0.0 | 0.0 | 19.6 | 0.0 | 0.0 | 5.6 | 5.6 | 28.9 | 4.8 | 0.0 | 22.0 | 0.0 | 0.0 | 0.0 | 0.0 | 0.0 | 0.0 | 0.0 | 33.3 | 0.0 | 14.3 | 2.7 |
| Qwen2.5-72B-Instruct | 0.0 | 0.0 | 10.3 | 0.0 | 0.0 | 0.0 | 0.0 | 11.8 | 0.0 | 0.0 | 0.0 | 3.5 | 11.1 | 4.0 | 0.0 | 4.0 | 0.0 | 0.0 | 0.0 | 0.2 | 0.0 | 0.0 | 0.0 | 0.0 | 0.0 | 14.3 | 0.9 |
| Mistral-Large-Instruct-2407 | 60.0 | 0.0 | 10.3 | 0.0 | 1.6 | 0.0 | 0.0 | 11.8 | 0.0 | 0.0 | 0.0 | 4.5 | 33.3 | 8.0 | 6.5 | 41.0 | 0.0 | 2.5 | 0.0 | 0.0 | 11.1 | 0.0 | 0.0 | 0.0 | 0.0 | 14.3 | 0.7 |
| gpt-4.1-mini | 0.0 | 0.0 | 10.3 | 0.0 | 0.0 | 0.0 | 0.0 | 13.7 | 0.0 | 0.2 | 0.0 | 6.2 | 41.7 | 0.0 | 0.0 | 14.0 | 0.0 | 1.0 | 0.0 | 0.0 | 0.0 | 0.0 | 0.0 | 0.0 | 0.0 | 0.0 | 1.3 |
| gpt-4o-mini | 0.0 | 0.0 | 10.3 | 0.0 | 0.0 | 0.0 | 0.0 | 9.8 | 0.0 | 0.0 | 2.4 | 3.4 | 8.3 | 0.0 | 0.8 | 2.0 | 2.4 | 0.0 | 0.0 | 0.0 | 0.0 | 0.0 | 0.0 | 0.0 | 0.0 | 0.0 | 0.7 |
| Llama-4-Scout-17B-16E-Instruct | 0.0 | 0.0 | 10.3 | 0.0 | 2.7 | 0.0 | 0.0 | 0.0 | 0.0 | 0.0 | 3.2 | 1.2 | 8.3 | 0.0 | 0.0 | 0.0 | 0.0 | 0.5 | 0.0 | 0.0 | 0.0 | 0.0 | 0.0 | 0.0 | 0.0 | 8.6 | 0.8 |
| gpt-5-nano | 0.0 | 0.0 | 10.3 | 0.0 | 0.0 | 0.0 | 0.0 | 0.0 | 0.0 | 0.0 | 0.0 | 6.3 | 12.2 | 0.0 | 0.0 | 0.0 | 0.0 | 0.5 | 0.0 | 0.0 | 0.0 | 0.0 | 0.0 | 0.0 | 0.0 | 0.0 | 1.5 |
| Llama-4-Maverick-17B-128E-Instruct-FP8 | 0.0 | 0.0 | 10.3 | 0.0 | 0.0 | 0.0 | 0.0 | 5.9 | 0.0 | 0.0 | 7.2 | 4.1 | 12.8 | 3.2 | 5.0 | 10.0 | 0.0 | 0.0 | 0.0 | 0.0 | 0.0 | 0.0 | 0.0 | 0.0 | 0.0 | 14.3 | 6.7 |
| Mistral-Small-Instruct-2409 | 0.0 | 0.0 | 10.3 | 0.0 | 0.5 | 0.0 | 0.0 | 0.0 | 0.0 | 0.0 | 0.8 | 4.0 | 8.3 | 0.0 | 0.0 | 8.3 | 0.0 | 0.0 | 0.0 | 0.0 | 0.0 | 0.0 | 0.0 | 0.0 | 0.0 | 8.6 | 0.7 |
| Llama-3.1-8B-Instruct | 0.0 | 0.0 | 10.3 | 0.0 | 0.0 | 0.0 | 0.0 | 15.7 | 0.0 | 0.0 | 4.0 | 4.3 | 15.6 | 0.0 | 0.0 | 2.0 | 0.0 | 0.0 | 0.0 | 0.0 | 0.0 | 0.0 | 0.0 | 0.0 | 0.0 | 2.9 | 1.3 |
| DeepSeek-R1 | 0.0 | 0.0 | 10.3 | 0.0 | 0.0 | 0.0 | 0.0 | 0.0 | 0.0 | 0.0 | 1.6 | 0.3 | 11.1 | 0.0 | 0.0 | 0.0 | 0.0 | 0.0 | 0.0 | 0.0 | 0.0 | 0.0 | 0.0 | 0.0 | 0.0 | 14.3 | 0.7 |
| Qwen2.5-7B-Instruct | 0.0 | 0.0 | 10.3 | 0.0 | 0.0 | 0.0 | 0.0 | 3.9 | 0.0 | 0.0 | 1.6 | 2.6 | 8.3 | 0.0 | 0.0 | 0.0 | 0.0 | 0.0 | 0.0 | 0.0 | 0.0 | 0.0 | 0.0 | 0.0 | 0.0 | 14.3 | 0.7 |
| Llama-3.2-3B-Instruct | 0.0 | 0.0 | 10.3 | 0.0 | 0.0 | 0.0 | 0.0 | 9.8 | 0.0 | 0.0 | 0.0 | 0.3 | 8.3 | 0.0 | 0.0 | 0.0 | 0.0 | 0.0 | 0.0 | 0.0 | 0.0 | 0.0 | 0.0 | 0.0 | 0.0 | 14.3 | 0.9 |
| phi-4 | 0.0 | 0.0 | 10.3 | 0.0 | 0.0 | 0.0 | 0.0 | 7.8 | 0.0 | 0.0 | 3.2 | 2.7 | 18.9 | 0.0 | 0.0 | 0.0 | 0.0 | 0.0 | 0.0 | 0.0 | 0.0 | 0.0 | 0.0 | 16.7 | 0.0 | 0.0 | 1.3 |
| gpt-4.1-nano | 0.0 | 0.0 | 10.3 | 0.0 | 0.0 | 0.0 | 0.0 | 0.0 | 0.0 | 0.0 | 0.0 | 1.7 | 13.9 | 0.0 | 0.0 | 0.0 | 0.0 | 0.0 | 0.0 | 0.0 | 0.0 | 0.0 | 0.0 | 0.0 | 0.0 | 0.0 | 0.7 |
| Mistral-Small-24B-Instruct-2501 | 0.0 | 0.0 | 10.3 | 0.0 | 0.8 | 0.0 | 0.0 | 7.8 | 0.0 | 0.0 | 1.6 | 3.2 | 5.0 | 0.0 | 0.0 | 0.0 | 0.0 | 0.0 | 0.0 | 0.0 | 0.0 | 0.0 | 0.0 | 0.0 | 0.0 | 8.6 | 0.7 |
| DeepSeek-R1-Distill-Llama-70B | 0.0 | 0.0 | 10.3 | 0.0 | 0.0 | 0.0 | 0.0 | 0.0 | 0.0 | 0.0 | 0.0 | 1.7 | 11.1 | 0.0 | 0.0 | 0.0 | 0.0 | 0.0 | 0.0 | 0.0 | 0.0 | 0.0 | 0.0 | 0.0 | 0.0 | 5.7 | 0.7 |
| Ministral-8B-Instruct-2410 | 0.0 | 0.0 | 10.3 | 0.0 | 0.0 | 0.0 | 0.0 | 0.0 | 0.0 | 0.0 | 0.0 | 0.3 | 5.6 | 0.0 | 0.0 | 0.0 | 0.0 | 0.0 | 0.0 | 0.0 | 0.0 | 0.0 | 0.0 | 0.0 | 0.0 | 0.0 | 0.7 |
| Mistral-Small-3.1-24B-Instruct-2503 | 0.0 | 0.0 | 10.3 | 0.0 | 0.0 | 0.0 | 0.0 | 9.8 | 0.0 | 0.0 | 0.0 | 0.3 | 6.7 | 0.0 | 0.0 | 0.0 | 0.0 | 0.0 | 0.0 | 0.0 | 0.0 | 0.0 | 0.0 | 0.0 | 0.0 | 0.0 | 0.7 |
| Mixtral-8x22B-Instruct-v0.1 | 0.0 | 0.0 | 10.3 | 0.0 | 0.0 | 0.0 | 0.0 | 0.0 | 0.0 | 0.0 | 0.0 | 0.3 | 3.3 | 0.0 | 0.0 | 0.0 | 0.0 | 0.0 | 0.0 | 0.0 | 0.0 | 0.0 | 0.0 | 0.0 | 0.0 | 14.3 | 0.7 |
| Llama-3.2-1B-Instruct | 0.0 | 0.0 | 10.3 | 0.0 | 0.0 | 0.0 | 0.0 | 0.0 | 0.0 | 0.0 | 0.0 | 0.3 | 2.8 | 0.0 | 0.0 | 0.0 | 0.0 | 0.0 | 0.0 | 0.0 | 0.0 | 0.0 | 0.0 | 0.0 | 0.0 | 0.0 | 0.7 |
| Phi-3-mini-128k-instruct | 0.0 | 0.0 | 10.3 | 0.0 | 0.0 | 0.0 | 0.0 | 0.0 | 0.0 | 0.0 | 0.0 | 0.3 | 2.8 | 0.0 | 0.0 | 0.0 | 0.0 | 0.0 | 0.0 | 0.0 | 0.0 | 0.0 | 0.0 | 0.0 | 0.0 | 0.0 | 0.7 |
| Phi-3.5-MoE-instruct | 0.0 | 0.0 | 10.3 | 0.0 | 0.0 | 0.0 | 0.0 | 0.0 | 0.0 | 0.0 | 0.0 | 0.3 | 2.8 | 0.0 | 0.0 | 0.0 | 0.0 | 0.0 | 0.0 | 0.0 | 0.0 | 0.0 | 0.0 | 0.0 | 0.0 | 0.0 | 0.7 |
| Phi-4-mini-instruct | 0.0 | 0.0 | 10.3 | 0.0 | 0.0 | 0.0 | 0.0 | 0.0 | 0.0 | 0.0 | 0.0 | 0.3 | 2.8 | 0.0 | 0.0 | 0.0 | 0.0 | 0.0 | 0.0 | 0.0 | 0.0 | 0.0 | 0.0 | 0.0 | 0.0 | 0.0 | 0.7 |
| Mixtral-8x7B-Instruct-v0.1 | 0.0 | 0.0 | 10.3 | 0.0 | 0.0 | 0.0 | 0.0 | 0.0 | 0.0 | 0.0 | 0.0 | 0.3 | 2.8 | 0.0 | 0.0 | 0.0 | 0.0 | 0.0 | 0.0 | 0.0 | 0.0 | 0.0 | 0.0 | 0.0 | 0.0 | 0.0 | 0.8 |
| Phi-3.5-mini-instruct | 0.0 | 0.0 | 10.3 | 0.0 | 0.0 | 0.0 | 0.0 | 0.0 | 0.0 | 0.0 | 0.0 | 0.3 | 2.8 | 0.0 | 0.0 | 0.0 | 0.0 | 0.0 | 0.0 | 0.0 | 0.0 | 0.0 | 0.0 | 0.0 | 0.0 | 0.0 | 0.7 |
| Phi-3-medium-128k-instruct | 0.0 | 0.0 | 10.3 | 0.0 | 0.0 | 0.0 | 0.0 | 0.0 | 0.0 | 0.0 | 0.0 | 0.3 | 2.8 | 0.0 | 0.0 | 0.0 | 0.0 | 0.0 | 0.0 | 0.0 | 0.0 | 0.0 | 0.0 | 0.0 | 0.0 | 0.0 | 0.7 |

Table 16: Model Performance on JERICHO games(tasks), part 2.

| Models | Lurking | Moonlit | Murdac | Night | Omniquest | Partyfoul | Pentari | Planetfall | Plundered | Reverb | Seastalker | Sherlock | Snacktime | Sorcerer | Spellbrkr | Spirit | Temple | Trinity | Tryst205 | Weapon | Wishbringer | Yomomma | Zenon | Zork1 | Zork2 | Zork3 | Zuul |
|---|---|---|---|---|---|---|---|---|---|---|---|---|---|---|---|---|---|---|---|---|---|---|---|---|---|---|---|
| o3 (medium) | 5.0 | 0.0 | 9.8 | 24.0 | 26.0 | 0.0 | 7.1 | 7.5 | 2.4 | 30.0 | 4.0 | 11.4 | 40.0 | 7.0 | 9.2 | 0.6 | 16.0 | 12.0 | 3.4 | 0.0 | 13.2 | 2.9 | 0.0 | 15.5 | 0.0 | 42.9 | 2.0 |
| o3 (high) | 5.0 | 0.0 | 9.0 | 16.0 | 14.0 | 0.0 | 7.1 | 7.5 | 3.2 | 28.8 | 8.0 | 11.6 | 40.0 | 7.0 | 10.3 | 1.1 | 14.3 | 13.4 | 4.0 | 0.0 | 15.2 | 2.9 | 0.0 | 13.3 | 3.0 | 40.0 | 5.0 |
| gpt-5 (thinking) | 5.0 | 20.0 | 12.5 | 12.0 | 10.0 | 0.0 | 15.7 | 7.5 | 2.4 | 34.0 | 11.0 | 9.4 | 60.0 | 9.2 | 10.0 | 1.3 | 21.1 | 11.0 | 3.7 | 0.0 | 13.0 | 1.7 | 0.0 | 17.8 | 0.0 | 34.3 | 6.0 |
| o3 (low) | 5.0 | 0.0 | 6.8 | 24.0 | 14.0 | 0.0 | 12.9 | 7.5 | 3.2 | 24.8 | 11.2 | 9.2 | 40.0 | 7.0 | 10.7 | 1.0 | 16.0 | 14.2 | 0.9 | 0.0 | 13.4 | 2.9 | 0.0 | 14.4 | 0.0 | 48.6 | 3.0 |
| claude-3.7-sonnet (thinking) | 5.0 | 0.0 | 11.3 | 20.0 | 10.0 | 0.0 | 7.1 | 4.5 | 4.0 | 4.0 | 18.6 | 10.6 | 36.0 | 5.3 | 11.3 | 1.4 | 14.3 | 11.8 | 0.6 | 0.0 | 11.6 | 0.0 | 0.0 | 12.2 | 0.0 | 40.0 | 5.0 |
| claude-3.7-sonnet | 5.0 | 0.0 | 10.8 | 20.0 | 10.0 | 0.0 | 7.1 | 3.8 | 2.4 | 0.0 | 20.6 | 3.6 | 32.0 | 9.2 | 11.7 | 1.4 | 14.3 | 13.8 | 0.0 | 0.0 | 13.0 | 0.0 | 0.0 | 13.5 | 0.0 | 42.9 | 5.0 |
| claude-3.5-sonnet-latest | 2.0 | 0.0 | 8.7 | 0.0 | 10.0 | 0.0 | 10.0 | 6.8 | 6.4 | 0.0 | 11.8 | 3.2 | 12.0 | 7.0 | 9.7 | 0.5 | 2.9 | 6.0 | 0.3 | 0.0 | 6.8 | 1.1 | 0.0 | 11.7 | 4.0 | 20.0 | 6.0 |
| gpt-4.1 | 5.0 | 0.0 | 5.6 | 8.0 | 16.0 | 0.0 | 4.3 | 6.0 | 1.6 | 8.8 | 7.6 | 8.8 | 24.0 | 4.2 | 6.7 | 1.0 | 0.0 | 9.2 | 0.6 | 0.0 | 13.8 | 2.3 | 0.0 | 12.5 | 1.2 | 31.4 | 1.0 |
| gpt-5-mini (thinking) | 5.0 | 0.0 | 8.7 | 0.0 | 12.0 | 0.0 | 7.1 | 2.2 | 2.4 | 6.8 | 10.4 | 7.2 | 40.0 | 7.8 | 9.0 | 1.1 | 16.6 | 6.4 | 1.7 | 0.0 | 13.0 | 0.6 | 0.0 | 7.9 | 1.0 | 42.9 | 4.0 |
| o1 | 7.0 | 0.0 | 8.6 | 0.0 | 10.0 | 0.0 | 8.6 | 3.8 | 4.0 | 14.0 | 3.8 | 0.4 | 40.0 | 7.8 | 8.0 | 1.1 | 11.4 | 9.4 | 0.9 | 0.0 | 13.8 | 1.7 | 0.0 | 12.5 | 1.0 | 42.9 | 9.6 |
| gpt-4o | 5.0 | 0.0 | 5.6 | 0.0 | 10.0 | 0.0 | 8.6 | 7.5 | 3.2 | 8.8 | 3.2 | 8.8 | 20.0 | 2.0 | 9.0 | 0.8 | 0.0 | 9.6 | 0.0 | 0.0 | 7.4 | 0.0 | 0.0 | 14.5 | 0.0 | 31.4 | 0.0 |
| claude-3.5-haiku | 5.0 | 0.0 | 5.3 | 20.0 | 10.0 | 0.0 | 5.7 | 6.8 | 2.4 | 14.0 | 10.8 | 8.8 | 8.0 | 3.2 | 4.2 | 0.8 | 0.0 | 9.0 | 0.0 | 0.0 | 11.8 | 0.0 | 0.0 | 13.1 | 0.0 | 31.4 | 0.0 |
| Llama-3.1-405B-Instruct | 5.0 | 0.0 | 5.6 | 0.0 | 10.0 | 0.0 | 7.1 | 3.8 | 1.6 | 0.0 | 8.2 | 2.6 | 24.0 | 1.2 | 6.7 | 1.3 | 14.3 | 4.0 | 0.0 | 0.0 | 6.0 | 0.0 | 0.0 | 11.1 | 0.0 | 25.7 | 13.0 |
| gemini-2.0-flash | 3.0 | 0.0 | 5.2 | 20.0 | 10.0 | 0.0 | 4.3 | 0.0 | 4.0 | 0.0 | 10.6 | 10.4 | 40.0 | 1.2 | 4.2 | 1.3 | 0.0 | 4.0 | 0.0 | 0.0 | 11.6 | 0.0 | 0.0 | 11.4 | 0.0 | 28.6 | 1.0 |
| Qwen3-32B | 2.0 | 0.0 | 4.4 | 0.0 | 8.0 | 0.0 | 5.7 | 3.8 | 4.0 | 0.0 | 8.2 | 2.4 | 0.0 | 1.7 | 6.7 | 0.0 | 14.3 | 4.8 | 0.0 | 0.0 | 7.4 | 0.0 | 0.0 | 9.4 | 0.0 | 42.9 | 4.0 |
| Llama-3.3-70B-Instruct | 2.0 | 0.0 | 5.6 | 0.0 | 10.0 | 0.0 | 2.9 | 5.3 | 1.6 | 0.0 | 17.4 | 3.8 | 4.0 | 1.2 | 4.2 | 0.0 | 16.0 | 4.4 | 0.0 | 0.0 | 11.6 | 0.6 | 0.0 | 4.3 | 0.0 | 31.4 | 3.0 |
| Llama-3.1-70B-Instruct | 5.0 | 0.0 | 5.6 | 0.0 | 10.0 | 0.0 | 4.3 | 3.8 | 1.6 | 0.0 | 21.0 | 2.2 | 16.0 | 1.2 | 4.2 | 0.0 | 0.0 | 8.4 | 0.0 | 0.0 | 6.0 | 0.0 | 0.0 | 3.1 | 0.0 | 42.9 | 13.0 |
| Qwen2.5-72B-Instruct | 5.0 | 0.0 | 4.6 | 0.0 | 10.0 | 0.0 | 2.9 | 0.8 | 0.0 | 0.0 | 9.6 | 3.6 | 24.0 | 2.0 | 7.2 | 0.8 | 0.0 | 5.0 | 0.0 | 0.0 | 1.0 | 0.6 | 0.0 | 2.3 | 0.0 | 28.6 | 0.0 |
| Mistral-Large-Instruct-2407 | 4.0 | 0.0 | 5.6 | 0.0 | 0.0 | 0.0 | 0.0 | 4.5 | 0.8 | 0.0 | 9.8 | 8.8 | 0.0 | 1.2 | 7.0 | 1.6 | 0.0 | 2.8 | 0.0 | 0.0 | 6.0 | 0.0 | 0.0 | 12.8 | 1.0 | 28.6 | 0.0 |
| gpt-4.1-mini | 5.0 | 0.0 | 5.6 | 0.0 | 10.0 | 0.0 | 0.0 | 2.2 | 0.0 | 0.0 | 3.2 | 1.6 | 0.0 | 1.7 | 3.3 | 0.3 | 0.0 | 3.0 | 0.0 | 0.0 | 6.4 | 0.6 | 0.0 | 6.9 | 2.0 | 28.6 | 1.0 |
| gpt-4o-mini | 4.0 | 0.0 | 5.6 | 0.0 | 0.0 | 0.0 | 4.3 | 3.8 | 0.0 | 0.0 | 2.8 | 1.4 | 0.0 | 1.2 | 4.2 | 0.3 | 0.0 | 7.4 | 0.0 | 0.0 | 6.0 | 0.0 | 0.0 | 3.1 | 2.0 | 5.7 | 0.0 |
| Llama-4-Scout-17B-16E-Instruct | 0.0 | 0.0 | 0.0 | 0.0 | 2.0 | 0.0 | 7.1 | 3.8 | 0.0 | 0.0 | 3.0 | 1.4 | 0.0 | 1.2 | 5.2 | 0.2 | 0.0 | 4.0 | 0.0 | 0.0 | 6.0 | 0.0 | 0.0 | 1.4 | 0.0 | 42.9 | 0.0 |
| gpt-5-nano | 0.0 | 0.0 | 1.3 | 12.0 | 10.0 | 0.0 | 0.0 | 3.0 | 0.0 | 0.0 | 3.0 | 1.2 | 0.0 | 1.2 | 4.7 | 0.0 | 0.0 | 3.8 | 0.0 | 0.0 | 6.0 | 0.0 | 0.0 | 0.6 | 0.0 | 28.6 | 2.0 |
| Llama-4-Maverick-17B-128E-Instruct-FP8 | 3.0 | 0.0 | 0.0 | 0.0 | 10.0 | 0.0 | 0.0 | 3.8 | 0.0 | 0.0 | 2.6 | 0.0 | 0.0 | 1.2 | 0.0 | 0.0 | 0.0 | 0.6 | 0.0 | 0.0 | 6.0 | 0.0 | 0.0 | 2.0 | 0.0 | 0.0 | 0.0 |
| Mistral-Small-Instruct-2409 | 0.0 | 0.0 | 3.4 | 0.0 | 10.0 | 0.0 | 0.0 | 3.0 | 0.0 | 0.0 | 2.0 | 0.0 | 0.0 | 1.2 | 3.3 | 0.0 | 8.6 | 1.0 | 0.0 | 0.0 | 6.0 | 0.0 | 0.0 | 1.4 | 0.0 | 17.1 | 0.0 |
| Llama-3.1-8B-Instruct | 0.0 | 0.0 | 0.0 | 0.0 | 0.0 | 0.0 | 0.0 | 0.8 | 0.0 | 0.0 | 3.0 | 0.0 | 0.0 | 1.2 | 4.2 | 0.0 | 0.0 | 1.0 | 0.0 | 0.0 | 6.0 | 0.0 | 0.0 | 2.0 | 0.0 | 0.0 | 0.0 |
| DeepSeek-R1 | 0.0 | 0.0 | 0.0 | 0.0 | 10.0 | 0.0 | 0.0 | 0.0 | 0.0 | 0.0 | 3.0 | 102.0 | 0.0 | 0.0 | 0.0 | 0.0 | 0.0 | 4.0 | 0.0 | 0.0 | 1.0 | 0.0 | 0.0 | 2.9 | 0.0 | 0.0 | 1.0 |
| Qwen2.5-7B-Instruct | 1.0 | 0.0 | 0.0 | 0.0 | 10.0 | 0.0 | 0.0 | 0.0 | 0.0 | 0.0 | 2.6 | 0.0 | 0.0 | 1.2 | 0.0 | 0.2 | 0.0 | 0.0 | 0.0 | 0.0 | 6.0 | 0.0 | 0.0 | 1.4 | 0.0 | 11.4 | 0.0 |
| Llama-3.2-3B-Instruct | 0.0 | 0.0 | 0.0 | 0.0 | 0.0 | 0.0 | 0.0 | 0.0 | 0.0 | 0.0 | 2.4 | 0.0 | 0.0 | 1.2 | 0.0 | 0.0 | 0.0 | 0.0 | 0.0 | 0.0 | 6.0 | 0.0 | 0.0 | 0.6 | 0.0 | 0.0 | 0.0 |
| phi-4 | 0.0 | 0.0 | 0.0 | 0.0 | 0.0 | 0.0 | 0.0 | 0.0 | 0.0 | 0.0 | 3.0 | 0.0 | 0.0 | 1.2 | 0.0 | 0.0 | 14.3 | 0.0 | 0.0 | 0.0 | 6.0 | 0.0 | 0.0 | 2.0 | 0.0 | 22.9 | 0.0 |
| gpt-4.1-nano | 1.0 | 0.0 | 0.0 | 0.0 | 10.0 | 0.0 | 0.0 | 3.0 | 0.0 | 0.0 | 2.6 | 0.0 | 0.0 | 0.0 | 0.0 | 0.0 | 0.0 | 0.0 | 0.0 | 0.0 | 6.0 | 0.0 | 0.0 | 2.9 | 0.0 | 14.3 | 0.0 |
| Mistral-Small-24B-Instruct-2501 | 0.0 | 0.0 | 0.0 | 0.0 | 10.0 | 0.0 | 0.0 | 0.0 | 0.0 | 0.0 | 2.2 | 0.0 | 0.0 | 1.2 | 0.0 | 0.0 | 5.7 | 0.0 | 0.0 | 0.0 | 4.0 | 0.0 | 0.0 | 0.0 | 0.0 | 14.3 | 0.0 |
| DeepSeek-R1-Distill-Llama-70B | 1.0 | 0.0 | 0.0 | 0.0 | 10.0 | 0.0 | 0.0 | 0.0 | 0.0 | 0.0 | 0.0 | 0.0 | 0.0 | 1.2 | 0.0 | 0.0 | 2.9 | 0.0 | 0.0 | 0.0 | 0.0 | 0.0 | 0.0 | 0.0 | 0.0 | 0.0 | 0.0 |
| Ministral-8B-Instruct-2410 | 0.0 | 0.0 | 0.0 | 0.0 | 0.0 | 0.0 | 0.0 | 0.0 | 0.0 | 0.0 | 1.0 | 0.0 | 0.0 | 0.2 | 0.0 | 0.0 | 0.0 | 0.0 | 0.0 | 0.0 | 6.0 | 0.0 | 0.0 | 0.0 | 0.0 | 0.0 | 0.0 |
| Mistral-Small-3.1-24B-Instruct-2503 | 0.0 | 0.0 | 0.0 | 0.0 | 10.0 | 0.0 | 0.0 | 0.0 | 0.0 | 0.0 | 1.2 | 0.0 | 0.0 | 0.0 | 0.0 | 0.0 | 0.0 | 0.0 | 0.0 | 0.0 | 6.0 | 0.0 | 0.0 | 0.0 | 0.0 | 0.0 | 0.0 |
| Mixtral-8x22B-Instruct-v0.1 | 0.0 | 0.0 | 0.0 | 0.0 | 0.0 | 0.0 | 0.0 | 0.0 | 0.0 | 0.0 | 1.0 | 0.0 | 0.0 | 0.0 | 0.0 | 0.3 | 0.0 | 2.2 | 0.0 | 0.0 | 0.0 | 0.0 | 0.0 | 0.0 | 0.0 | 0.0 | 0.0 |
| Llama-3.2-1B-Instruct | 0.0 | 0.0 | 0.0 | 0.0 | 2.0 | 0.0 | 0.0 | 0.0 | 0.0 | 0.0 | 0.0 | 0.0 | 0.0 | 1.2 | 0.0 | 0.0 | 0.0 | 0.0 | 0.0 | 0.0 | 1.0 | 0.0 | 0.0 | 0.0 | 0.0 | 11.4 | 0.0 |
| Phi-3-mini-128k-instruct | 0.0 | 0.0 | 0.0 | 0.0 | 0.0 | 0.0 | 0.0 | 0.0 | 0.0 | 0.0 | 0.0 | 0.0 | 0.0 | 0.2 | 0.0 | 0.0 | 0.0 | 0.0 | 0.0 | 0.0 | 1.2 | 0.0 | 0.0 | 0.0 | 0.0 | 0.0 | 0.0 |
| Phi-3.5-MoE-instruct | 0.0 | 0.0 | 0.0 | 0.0 | 8.0 | 0.0 | 0.0 | 0.0 | 0.0 | 0.0 | 0.0 | 0.0 | 0.0 | 1.2 | 0.0 | 0.0 | 0.0 | 0.0 | 0.0 | 0.0 | 1.6 | 0.0 | 0.0 | 0.0 | 0.0 | 0.0 | 0.0 |
| Phi-4-mini-instruct | 0.0 | 0.0 | 0.0 | 0.0 | 10.0 | 0.0 | 0.0 | 0.0 | 0.0 | 0.0 | 2.4 | 0.0 | 0.0 | 0.5 | 0.0 | 0.0 | 0.0 | 1.0 | 0.0 | 0.0 | 1.0 | 0.0 | 0.0 | 0.0 | 0.0 | 0.0 | 0.0 |
| Mixtral-8x7B-Instruct-v0.1 | 0.0 | 0.0 | 0.0 | 0.0 | 0.0 | 0.0 | 0.0 | 0.0 | 0.0 | 0.0 | 0.0 | 0.0 | 0.0 | 0.2 | 0.0 | 0.0 | 0.0 | 0.0 | 0.0 | 0.0 | 0.2 | 0.0 | 0.0 | 0.0 | 0.0 | 0.0 | 0.0 |
| Phi-3.5-mini-instruct | 0.0 | 0.0 | 0.0 | 0.0 | 0.0 | 0.0 | 0.0 | 0.0 | 0.0 | 0.0 | 0.0 | 0.0 | 0.0 | 0.0 | 0.0 | 0.0 | 0.0 | 0.0 | 0.0 | 0.0 | 0.0 | 0.0 | 0.0 | 0.0 | 0.0 | 0.0 | 0.0 |
| Phi-3-medium-128k-instruct | 0.0 | 0.0 | 0.0 | 0.0 | 0.0 | 0.0 | 0.0 | 0.0 | 0.0 | 0.0 | 0.0 | 0.0 | 0.0 | 0.0 | 0.0 | 0.0 | 0.0 | 0.0 | 0.0 | 0.0 | 0.0 | 0.0 | 0.0 | 0.0 | 0.0 | 0.0 | 0.0 |

Table 17: Average reasoning failures for the Claude family of models across 8 games (32 logs total).

| | Failures(↓) | | | | | TALES Score (↑) |
|---|---|---|---|---|---|---|
| Model | Spatial | Deductive | Inductive | Grounded | Total | |
| 3.7-Sonnet (Thinking) | 1.29 | 1.29 | 2.14 | 1.0 | 5.71 | 52.5 |
| 3.7-Sonnet | 0.86 | 3.57 | 5.43 | 1.0 | 10.86 | 52.1 |
| 3.5-Sonnet | 1.14 | 1.43 | 9.43 | 0.86 | 12.86 | 50.4 |
| 3.5-Haiku | 3.86 | 7.14 | 25.29 | 2.86 | 39.14 | 39.6 |

## K AVERAGE SIMON SAYS SCORE VERSUS OVERALL TALES SCORE

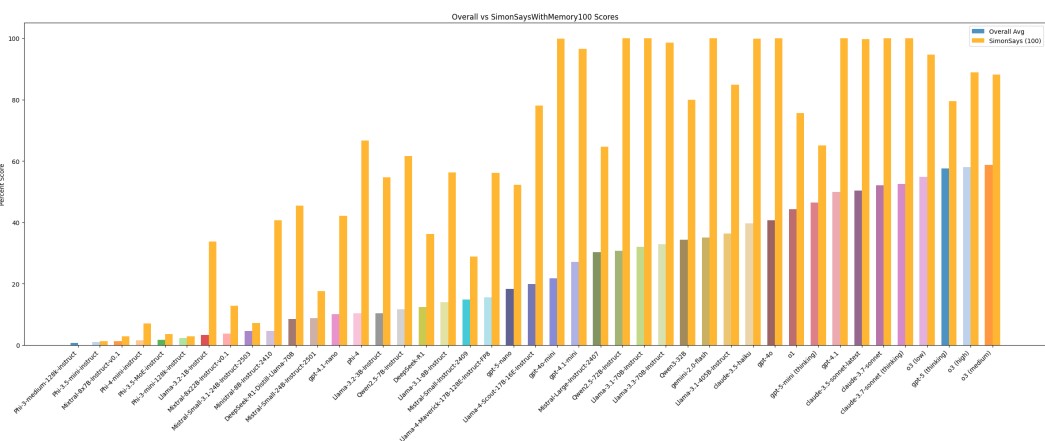

Figure 3: All TALES and average SIMON SAYS scores for each model, sorted by TALES performance. We see that an increase in performance in SIMON SAYS typically correlates with an increase in performance for TALES overall.

## L COMPUTE

All experiments were run intermittently over a course of roughly six months. Open-weight models were run on a combination of a four node cluster of 8xMI300s and one node of 8xA100s. Anthropic API experiments accrued a cost of 1,562.15 U.S. dollars. OpenAI API experiments accrued an estimated cost of 6,870.76 U.S. dollars.

## M REASONING FAILURES

In this section, we provide explicit reasoning failure numbers from a smaller subset of the top performing LLMs, similar to what is shown in Figure 1. We focus again on the Claude family of models due to having access to a range of model performances as well as both thinking and non-thinking modes for their top model. We outline the specific criteria we used to classify actions as a specific type of reasoning failure. From a selection of 32 game logs of the most difficult games in the frameworks discussed in Section 5, we exhaustively label all actions that exhibit some reasoning failure, discuss our criteria for designating a chosen action as a reasoning failure, and present the cumulative, average failures per LLM, finding that **inductive reasoning failures** where the agent fails to account for explicit or implicit feedback are by far the most common.

As shown in Table 17, **top LLMs still make reasoning errors but resolve them quickly enough to still succeed.** When explicitly labeling and enumerating reasoning errors made on a per-action basis, we see a clear trend with the worse performing LLMs also making a larger number of reasoning errors. These reasoning errors ultimately waste a large portion of the weaker LLM's allotted steps for the environments, leading to a decreased performance. The stronger LLMs, while still making reasoning errors, make few enough errors to still have sufficient time to complete the tasks.

Claude-3.5-Haiku frequently encounters inductive reasoning failures as weaker models tend to get trapped in cyclical action loops that do not result in any meaningful change in state. We consider each repeated step in this cycle as a distinct failure because each step represents an instance where the agent fails to incorporate feedback. See Appendix Q for our labeling criteria and examples.

## N    1000 STEPS OF ZORK1

While even the best performing LLMs make reasoning mistakes, we find what allows them to still find success in TALES is the ability to both avoid making an excessive number of these mistakes and the ability to self-correct. We argue that 100 steps are sufficient to evaluate the performance of current state-of-the-art LLMs because even the best overall LLM, o3, fails to approach the maximum possible score within 100 steps for ZORK1 (13.8% vs 29.1%). In this experiment, we explore whether any of the top models can achieve a score comparable to the walkthrough after 100 steps in ZORK1 while allowed to run for 1000 steps with the entire history kept in the context. If so, we examine the behaviors that enable this and determine the required number of steps. We select the overall top 3 performing models, o3 and both the thinking and non-thinking modes of Claude-3.7-Sonnet.

**Scores improve slightly, but the best LLMs are still far from the walkthrough score even with 10 times more steps.** That is with 1000 steps, the best LLMs fail to reach 29.3% of the total score. o3 manages to achieve a score of 20.9%, a performance increase of only 7.6% over its original score of 13.3% for 10 times the steps. Claude-3.7 non-thinking[5] and thinking achieve 16.9% and 15.3% respectively. The key behavior pattern we see in both thinking models is a slightly directed, random exploration of the area of the game past the bottleneck that stops other, weaker LLMs. This exploration is far less focused than the iterative search we see agents perform early in the game and in simpler environments such as AW.

We tried allowing Claude-3.7-Sonnet to think for up to 4096 tokens. However, the model never uses more than 700 tokens for its thinking, a similar value seen for the rest of the benchmark where the thinking effort is capped to 1024. This is a significantly smaller thinking effort than o3 which uses up to 5000 thinking tokens throughout its 1000 steps ZORK1 playthrough. This suggests that o3's performance is due to a willingness to leverage many more thinking tokens at any particular step. However, the highest thinking efforts do not appear to occur at any significant points during gameplay and we are unable to verify the actual contents since we do not have access to the thinking traces.

## O    RL FINE-TUNING HYPERPARAMETERS

We use Qwen3 rather than Qwen2.5 due to its inherit reasoning training. Following guidance located in the appendix of their publication, for verl-agent we use a learning rate of 1e-6 for the actor and 1e-5 for the critic. We train over a total of 150 'epochs' which verl-agent represents as one full rollout of the environment. To more closely adhere to our zero-shot evaluations, we kee the entire history in the trajectory up to 50 steps. This necessitates a far smaller batch size of 8. During evaluation, we use a batch size of 32 to sufficiently sample all evaluation environments. We use the collection of all existing game seeds and formats not evaluated on in our zero-shot experiments as a training set and the specific environments from the zero-shot experiments as the test set. We allocate a thinking budget of 512 tokens. In their ALFWORLD implementation, (Feng et al., 2025) use a terminal reward of 10. In our RL experiments, we use the native reward from the environment, as our motivation is not to contest (Feng et al., 2025)'s results but to evaluate the effects of including privileged or domain knowledge directly in the prompt.

## P    FRAMEWORK ENVIRONMENT SUBSELECTION

### P.1    TEXTWORLD

For TEXTWORLD, we use the following environments:

---

[5]Despite scoring higher than its thinking variant, zero-shot Claude-3.7-Sonnet suffers a catastrophic inductive reasoning failure by repeatedly issuing the quitting commands after step 479.

```
test/difficulty_level_1/tw-cooking-recipe1+take1+open-0nQyHWbvh6d
↪    XFPmhLKX.z8
test/difficulty_level_2/tw-cooking-recipe1+take1+cook+open-0nQyHW
↪    bvh6dXFPmhLKX.z8
test/difficulty_level_3/tw-cooking-recipe1+take1+cut+open-0nQyHWb
↪    vh6dXFPmhLKX.z8
test/difficulty_level_4/tw-cooking-recipe1+take1+open+go6-0nQyHWb
↪    vh6dXFPmhLKX.z8
test/difficulty_level_5/tw-cooking-recipe1+take1+open+go9-0nQyHWb
↪    vh6dXFPmhLKX.z8
test/difficulty_level_6/tw-cooking-recipe1+take1+open+go12-0nQyHW
↪    bvh6dXFPmhLKX.z8
test/difficulty_level_7/tw-cooking-recipe1+take1+cook+cut+open-0n
↪    QyHWbvh6dXFPmhLKX.z8
test/difficulty_level_8/tw-cooking-recipe3+take3+open+go6-0nQyHWb
↪    vh6dXFPmhLKX.z8
test/difficulty_level_9/tw-cooking-recipe3+take3+cook+cut+open+go
↪    6-0nQyHWbvh6dXFPmhLKX.z8
test/difficulty_level_10/tw-cooking-recipe3+take3+cook+cut+open+g
↪    o12-0nQyHWbvh6dXFPmhLKX.z8
```

## P.2 TEXTWORLDEXPRESS

For TEXTWORLDEXPRESS, we use the game parameters:

```
TASKS = [
    (
        "CookingWorld",
        "cookingworld",
        "numLocations=1, numIngredients=2, numDistractorItems=5,
        ↪    includeDoors=0, limitInventorySize=0",
    ),
    (
        "TextWorldCommonsense",
        "twc",
        "numLocations=1,numItemsToPutAway=1,includeDoors=0,limitI
        ↪    nventorySize=0",
    ),
    (
        "CoinCollector",
        "coin",
        "numLocations=1, numDistractorItems=5,
        ↪    limitInventorySize=0",
    ),
    ("Arithmetic", "arithmetic", ""),
    (
        "MapReader",
        "mapreader",
        "numLocations=2, maxDistanceApart=1,
        ↪    maxDistractorItemsPerLocation=2, includeDoors=0,
        ↪    limitInventorySize=0",
    ),
    ("Sorting", "sorting", ""),
    ("SimonSays10", "simonsays", "gameLength=10, numDistractors=4,
    ↪    memorization=0"),
    ("SimonSays50", "simonsays", "gameLength=50, numDistractors=4,
    ↪    memorization=0"),
    ("SimonSays100", "simonsays", "gameLength=100,
    ↪    numDistractors=4, memorization=0"),
```

```
1620        (
1621            "SimonSaysWithMemory10",
1622            "simonsays",
1623            "gameLength=10, numDistractors=4, memorization=1,
1624            ↪  verbose=0",
1625        ),
1626        (
1627            "SimonSaysWithMemory50",
1628            "simonsays",
1629            "gameLength=50, numDistractors=4, memorization=1,
1630            ↪  verbose=0",
1631        ),
1632        (
1633            "SimonSaysWithMemory100",
1634            "simonsays",
1635            "gameLength=100, numDistractors=4, memorization=1,
1636            ↪  verbose=0",
1637        ),
1638        (
1639            "SimonSaysWithMemory10Verbose",
1640            "simonsays",
1641            "gameLength=10, numDistractors=4, memorization=1,
1642            ↪  verbose=1",
1643        ),
1644        (
1645            "SimonSaysWithMemory50Verbose",
1646            "simonsays",
1647            "gameLength=50, numDistractors=4, memorization=1,
1648            ↪  verbose=1",
1649        ),
1650        (
1651            "SimonSaysWithMemory100Verbose",
1652            "simonsays",
1653            "gameLength=100, numDistractors=4, memorization=1,
1654            ↪  verbose=1",
        ),
        ("PeckingOrder", "peckingorder", ""),
    ]
```

## P.3 ALFWORLD

The 12 games for ALFWORLD. Note that these are from when the "–game-seed" is not set. Changing this value would cause the games to change.

```
valid_seen/pick_and_place_simple-Book-None-SideTable-329/trial_T2↵
↪  0190908_050633_745514
valid_seen/look_at_obj_in_light-AlarmClock-None-DeskLamp-323/tria↵
↪  l_T20190909_044715_250790
valid_seen/pick_clean_then_place_in_recep-ButterKnife-None-Counte↵
↪  rTop-8/trial_T20190909_105559_983897
valid_seen/pick_heat_then_place_in_recep-Apple-None-DiningTable-2↵
↪  6/trial_T20190907_060234_011675
valid_seen/pick_cool_then_place_in_recep-Apple-None-CounterTop-14↵
↪  /trial_T20190909_044933_815840
valid_seen/pick_two_obj_and_place-AlarmClock-None-Dresser-305/tri↵
↪  al_T20190907_165826_194855
valid_unseen/pick_and_place_simple-Mug-None-Desk-308/trial_T20190↵
↪  908_125200_737896
```

```
valid_unseen/look_at_obj_in_light-AlarmClock-None-DeskLamp-308/tr⌟
↪ ial_T20190908_222917_366542
valid_unseen/pick_clean_then_place_in_recep-Bowl-None-Cabinet-10/⌟
↪ trial_T20190909_061130_844814
valid_unseen/pick_heat_then_place_in_recep-Apple-None-Fridge-10/t⌟
↪ rial_T20190906_182259_116320
valid_unseen/pick_cool_then_place_in_recep-Bread-None-CounterTop-⌟
↪ 10/trial_T20190908_091747_866951
valid_unseen/pick_two_obj_and_place-CD-None-Safe-308/trial_T20190⌟
↪ 907_050942_897916
```

## P.4 SCIENCEWORLD

We use the first variation of the test set for each of the 30 tasks.

# Q ANNOTATED LOGS: CRITERIA AND EXAMPLES

## Q.1 LABELING CRITERIA

For spatial reasoning failures, we primarily label those actions that involve failing navigation through path finding, backtracking, or iteratively searching spaces for necessary elements for completing the assigned task. Deductive reasoning failures were primarily actions that involved interaction with game elements that cannot be reasonably linked back to the target objective or failures to solve in-game puzzles. Inductive reasoning failures were comprised mostly of actions that the agent had previously attempted in a near-identical state to negative feedback. If there was any non-trivial change in game state compared to when the action was first attempted, we did not label the action as an inductive reasoning failure if the action did not fail due to a parser issue. Grounded reasoning failures encompassed what is commonly referred to as hallucination: attempts to interact with game elements not mentioned in observations or feedback.

## Q.2 SPATIAL REASONING FAILURE EXAMPLE

```
Task Description:Your task is to melt lead. First, focus on the
↪ substance. Then, take actions that will cause it to change its
↪ state of matter...

```
This room is called the kitchen. In it, you see:
        the agent
        a substance called air
        a chair. On the chair is: nothing.
        a counter. On the counter is: a bowl (containing a red
        ↪ apple, a banana, an orange, a potato), a drawer.
        a cupboard. The cupboard door is closed.
        a freezer. The freezer door is closed.
        a fridge. The fridge door is closed.
        a glass jar (containing a substance called sodium
        ↪ chloride)
        a lighter
        a oven, which is turned off. The oven door is closed.
        a painting
        a sink, which is turned off. In the sink is: nothing.
        a substance called soap
        a stopwatch, which is deactivated.
        a stove, which is turned off. On the stove is: nothing.
        a table. On the table is: a glass cup (containing
        ↪ nothing).
```

```
        a thermometer, currently reading a temperature of 10
        ↪ degrees celsius
You also see:
        A door to the bathroom (that is open)
        A door to the hallway (that is closed)
        A door to the outside (that is closed)
>
Action: open oven

...

Action: go hallway

...

Action: go workshop

```
This room is called the workshop. In it, you see:
        the agent
        a substance called air
        a table. On the table is: a battery, a green wire, a metal
        ↪ pot (containing a substance called lead), a orange
        ↪ light bulb, which is off, a orange wire, a red wire, a
        ↪ switch, which is off, a violet light bulb, which is
        ↪ off, a yellow light bulb, which is off.
        a ultra low temperature freezer. The ultra low temperature
        ↪ freezer door is closed.
You also see:
        A door to the hallway (that is open)
>
Action: pick up metal pot

You move the metal pot to the inventory.

Action: go kitchen
```

Above we provide an example of a spatial reasoning failure from Claude-3.5-Haiku in ScienceWorld-Melt. The agent navigates to the workshop, through the hallway and thus should realize the path is kitchen to hallway to workshop. However, despite this, the agent attempts the action go kitchen.

### Q.3 DEDUCTIVE REASONING FAILURE EXAMPLE

```
Task Description:Your task is to freeze lead. First, focus on the
↪ substance. Then, take actions that will cause it to change its
↪ state of matter.

...

Action: look around

This room is called the kitchen. In it, you see:
        the agent
        a substance called air
        a chair. On the chair is: nothing.
        a counter. On the counter is: a bowl (containing a red
        ↪ apple, a banana, an orange, a potato), a drawer.
        a cupboard. The cupboard door is closed.
```

```
           a freezer. The freezer door is closed.
           a fridge. The fridge door is closed.
           a glass jar (containing a substance called sodium
        ↪  chloride)
           a lighter
           a oven, which is turned off. The oven door is closed.
           a painting
           a sink, which is turned off. In the sink is: nothing.
           a substance called soap
           a stopwatch, which is deactivated.
           a stove, which is turned off. On the stove is: nothing.
           a table. On the table is: a glass cup (containing
        ↪  nothing).
           a thermometer, currently reading a temperature of 10
        ↪  degrees celsius
You also see:
           A door to the bathroom (that is open)
           A door to the hallway (that is closed)
           A door to the outside (that is closed)

Action: open freezer
```

Above we provide an example of a spatial reasoning failure from Claude-3.7-Sonnet (zero-shot) in ScienceWorldFreeze. While looking for lead, the agent checks a kitchen freezer, an extremely unlikely place to find lead.

## Q.4 INDUCTIVE REASONING FAILURE EXAMPLE

```
...

If you pray enough, your prayers may be answered.

Action: pray

If you pray enough, your prayers may be answered.

Action: pray

If you pray enough, your prayers may be answered.

Action: pray

If you pray enough, your prayers may be answered.

Action: pray

...
```

Above we provide an example of an inductive reasoning failure from Claude-3.5-Haiku in ZORK1. While the game environment returns a signal that praying might result in some change, Claude-3.5-Haiku continues to do this for 36 steps, far past when most reasonable players would have moved on.

## Q.5 GROUNDED REASONING FAILURE EXAMPLE

```
This room is called the workshop. In it, you see:
           the agent
```

```
                a substance called air
                a table. On the table is: a battery, a green wire, a
            ↪   orange light bulb, which is off, a orange wire, a red
            ↪   wire, a switch, which is off, a violet light bulb,
            ↪   which is off, a yellow light bulb, which is off.
                a ultra low temperature freezer. The ultra low temperature
            ↪   freezer door is closed.
You also see:
                A door to the hallway (that is open)

...

Action: go to hallway

You move to the hallway.

...

Action: go to hallway

You move to the hallway.

Action: go to kitchen

You move to the kitchen.

Action: use lighter on lead

The lighter heats up the lead a small amount.

Action: take red wire
```

Above we provide an example of an inductive reasoning failure from Claude-3.7-Sonnet (Thinking) in ScienceWorldBoil. We see the agent attempt to interact with an item in its context despite having moved to another location where the item is no longer present.

## R DATA CONTAMINATION EVIDENCE FROM THINKING TRACES: EXAMPLES

```
The blue glow of my sword suggests there's a danger nearby, likely
↪   something from Zork lore (like a grue). Let me explore the
↪   available paths carefully. I have the lantern on, which should
↪   protect me from grues in dark areas. Let me try going north
↪   first.
```

An example of a thinking trace from the Claude-3.7-Sonnet thinking mode that leads us to strongly suspect data contamination. Most notably, the thinking traces references a *grue*, a creature from ZORK1 mythology that had not at all been previously mentioned in the game's history.

## S HYPERPARAMETERS

The following is the command used to run experiments across 5 seeds. All arguments not included are default values.

```
for i in {{1..5}}; do
```

```
  python benchmark.py --agent agents/llm.py zero-shot --env
  ↪    jericho scienceworld textworld textworld_express alfworld
  ↪   --llm $LLM       --context 100 --nb-steps 100 --conversation
  ↪   --wandb --seed "20241106$((i))" &
  sleep 60
done
```

