# OpenReview forum: "TALES: Text Adventure Learning Environment Suite"
_ICLR.cc/2026/Conference — Submitted to ICLR 2026_

### Official Review · Reviewer_93MW · 2025-10-27

**Soundness:** 2
**Presentation:** 3
**Contribution:** 2
**Rating:** 4
**Confidence:** 3

**Summary:**

This paper introduces TALES, a unified benchmark spanning five text-adventure frameworks, with minimal scaffolding to probe raw, compositional reasoning. The authors evaluate 42 models in a zero-shot setting and find that while systems perform well on synthetic environments, they struggle markedly on human-written interactive fiction like JERICHO.

**Strengths:**

1. The definition of the four reasoning skills feels well-grounded and convincing, and the descriptions and setups of the tasks are clear and reasonable.
2. This paper contributes to evaluation stability and reproducibility, which are often overlooked but crucial for benchmarks.
3. The benchmark covers models very comprehensively, giving a broad and useful picture of how current models perform in different types of reasoning.
4. The paper reads smoothly overall

**Weaknesses:**

1. The benchmark imposes a fixed 100-step cap for all environments with different complexity. A dynamic step setting may be more meaningful. And Table 5 shows extremely low scores of all the LLMs on Jericho. This raises concerns about whether the reported performance reflects actual reasoning limitations or just under-allocated interaction budgets.
2. TEXTWORLD tasks seem trivial for modern LLMs, which all reach near 100%. It's unclear whether this task provides discriminative metric.
3. Lack of justification on the benchmark’s validity through, for example, ablative studies, consistency analyses. Clear empirical evidence of construct validity would strengthen the benchmark.
4. The final average score seems to be computed uniformly across environments. However, Table 8 show variance on different tasks. And the score in different task can be quite high or quite low. The paper does not discuss weighting strategies like whether certain environments dominate the final score distribution.
5. Although the paper mentions “strong evidence of data contamination” in human-written games, it's not clear how the reported results can be interpreted as measurements of reasoning rather than memorization.
6. Since the reward sparsity varies across the tasks. It is unclear whether observed performance differences reflect reasoning or reward shaping artifacts.
6. Reporting results in Table 6 to two decimal places is unnecessary.

**Questions:**

1. Although it’s necessary to evaluate the LLMs’ raw capabilities, I still wonder what’s their performance under the same rule instruction setting.
2. How does TALES ensure construct validity that the measured scores truly reflect the four proposed reasoning skills rather than other factors?
3. Has the benchmark been validated against human baselines or expert heuristics to confirm the intended difficulty hierarchy?
4. Is the results sensitive to small perturbations like the change in prompt?
5. More analysis on the four proposed skills in the experiments is needed

---

> ### Author Response · Authors · 2025-11-19
> **Rebuttal for Reviewer 93MW's review**
>
> We thank reviewer 93MW for their examination of our work and the subsequent feedback.
>
> Below are our responses to the outlined weaknesses and questions. We do apologize as we had difficulty understanding some points and have asked for more explanation where relevant:
>
> Weaknesses:
>
> - The benchmark imposes a fixed 100-step cap for all environments with different complexity. A dynamic step setting may be more meaningful. And Table 5 shows extremely low scores of all the LLMs on Jericho. This raises concerns about whether the reported performance reflects actual reasoning limitations or just under-allocated interaction budgets.
>   - The way we ran these benchmarks, we can continue evaluation beyond 100 steps, however in ablation experiments, (section M in the appendix) we saw very minor improvements up to 1000 steps.
> - TEXTWORLD tasks seem trivial for modern LLMs, which all reach near 100%. It's unclear whether this task provides discriminative metric.
>   - Please see section 'Difficulty/Saturation of the Benchmark' in the General Rebuttal as this is a concern shared with other reviewers
>
> - Lack of justification on the benchmark’s validity through, for example, ablative studies, consistency analyses. Clear empirical evidence of construct validity would strengthen the benchmark.
>   - We apologize as this was one point we had difficulty understanding. Would reviewer 93MW be able to provide some specific examples in terms of what ablations or consistency analyses they would have liked to see along with motivation?
>
> - The final average score seems to be computed uniformly across environments. However, Table 8 show variance on different tasks. And the score in different task can be quite high or quite low. The paper does not discuss weighting strategies like whether certain environments dominate the final score distribution.
>   - Specifically, we weigh the score across the tasks rather than the frameworks. We do provide the raw scores in the appendix, however we felt that a weighting was unnecessary due to the explicit progressive difficulty of the benchmark.
>
> - Although the paper mentions “strong evidence of data contamination” in human-written games, it's not clear how the reported results can be interpreted as measurements of reasoning rather than memorization.
>   - We would like to emphasize two points in response to this:
>     Some tasks, especially those in Jericho, have both stochasticity in the environment as well as multiple possible paths of completion, so the optimal solution is likely to differ from whatever transcripts could be found online.
>
>     For the synthetic games, these were likely not seen during training. We posit that solving these games in the allocated step budget would indicate some amount of reasoning.
>
> - Since the reward sparsity varies across the tasks. It is unclear whether observed performance differences reflect reasoning or reward shaping artifacts.
>   - In addition to reward, we could also report game completion rate
>
> - Reporting results in Table 6 to two decimal places is unnecessary.
>   - We felt it necessary to include decimal places when reporting all scores as they are averaged over 5 seeds for consistency: we felt a performance where the model could consistently achieve a score of, for example, 10 each time should be properly differentiated from a model that achieves scores of 12, 8, 9, 10, and 10
>
> - Although it’s necessary to evaluate the LLMs’ raw capabilities, I still wonder what’s their performance under the same rule instruction setting.
>   - We apologize and would ask for more clarification on this point. Specifically, what rule instruction setting is the reviewer referring to?
>
> - How does TALES ensure construct validity that the measured scores truly reflect the four proposed reasoning skills rather than other factors?
>   - We address this in the general rebuttal under section ‘Validity of Qualitative Analysis Results’ as this is a concern shared with other reviewers
>
> - Has the benchmark been validated against human baselines or expert heuristics to confirm the intended difficulty hierarchy?
>   - Could the reviewer expand on what they mean by ‘expert heuristics’?
>
> - Is the results sensitive to small perturbations like the change in prompt?
>   - Specifically, we opt to provide a baseline by using the most minimal prompt possible and discuss how certain changes to information included in the prompt can provide an unfair advantage and misrepresent a model’s capabilities in the included frameworks.
>
> - More analysis on the four proposed skills in the experiments is needed
>   - Would the reviewer be able to provide suggestions on what such experiments would be needed? We would like to emphasize again that the proposed skills are meant to supplement the TALES score and expand on what failure in TALES means in a qualitative manner.

---

> > ### Comment · Reviewer_93MW · 2025-11-27
> > **Clarification of the questions**
> >
> > 1. By construct validity, I meant some experiments to prove that TALES scores truly measure the reasoning abilities rather than other factors. Here is an example: in ScienceWorld, prior knowledge appears to strongly influence the final score, so is there any ablation study that can minimize the effect of prior knowledge, isolate pure reasoning ability?
> > 2. By ”the same rule instruction setting”, I’m referring to providing all the models with a standardized rules and constraints, so that we can identify whether the models are failing due to lack of rule-inference ability or reasoning abilities.
> > 3. By ”expert heuristics”, I’m suggesting that this benchmark include the score of simple, heuristic strategies. This can serve as a lightweight baselines to identify the difficulty of each environment.
> > 4. For the experiments on the four reasoning abilities, the paper mentions the four core abilities(deductive reasoning, inductive reasoning, spatial reasoning and grounded reasoning), but has few experiments to verify which reasoning ability contributes to the result. For example,  to mask the effect of spatial reasoning, we can provide the agent the spatial map of the current position and only ask it to perform other reasoning.

---

### Official Review · Reviewer_XTMp · 2025-10-29

**Soundness:** 3
**Presentation:** 3
**Contribution:** 2
**Rating:** 4
**Confidence:** 5

**Summary:**

The authors have introduced the Text Adventure Learning Environment Suite (TALES), a collection of text-adventure games designed to rigorously evaluate the reasoning abilities of LLMs. TALES presents a challenge in reasoning for the current state-of-the-art AI, revealing that even the most advanced models struggle with complex reasoning problem.

Key Contributions of the TALES paper:

A Unified Evaluation Framework: TALES provides a standardized benchmark, allowing for consistent and comparable evaluation across different models. The suite includes games from various frameworks like Jericho, TextWorld, TextWorldExpress, ALFWorld,
ScienceWorld, offering a broad spectrum of challenges.

Benchmarking Leading LLMs: The authors tested a wide range of both open- and closed-weight LLMs on the TALES benchmark.
Analysis of Model Failures: They also conduct a qualitative analysis of the top-performing models identified common failure points.

**Strengths:**

Both of the strengths and the weaknesses of the paper are quite evident.

1. The text-adventure games provide a good test bed to assess LLM's reasoning ability, which are verifiable (whether succeed in completing the tasks) and challenging enough with long-horizon tasks.
2. The paper provides a unified suite for the existing benchmarks like TextWorld, TextWorldExpress, ALFWorld, ScienceWorld, Jericho, which would make it more convenient for a systematic evaluation in various text adventure based games.
3. The paper includes a large amount of experiments across these games and mainstreaming open-source and close-source LLMs, and provides detailed results and solid setup for evaluation.

**Weaknesses:**

The provided experiments are robust and the provided unifying of the existing text-adventure benchmarks would be beneficial from a engineering perspective. However, from the novelty view, this paper doesn't include new design of gaming benchmark. This work does not include new games, nor new design of evaluation.

Meanwhile, leveraging interactive environments to assess various reasoning abilities is not new. And it is not surprising to see LLMs would fail in gaming where complex reasoning is needed. [1][2][3]

[1] Jinhao Duan, Renming Zhang, James Diffenderfer, Bhavya Kailkhura, Lichao Sun, Elias Stengel-Eskin, Mohit Bansal, Tianlong Chen, and Kaidi Xu. 2024b. Gtbench: Uncovering the strategic reasoning capabilities of llms via game-theoretic evaluations. In NeurIPS.

[2] Jen-tse Huang, Eric John Li, Man Ho Lam, Tian Liang, Wenxuan Wang, Youliang Yuan, Wenxiang Jiao, Xing Wang, Zhaopeng Tu, and Michael R Lyu. 2024. How far are we on the decision-making of llms? evaluating llms’ gaming ability in multi-agent environments. arXiv preprint arXiv:2403.11807.

[3] Wenye Lin, Jonathan Roberts, Yunhan Yang, Samuel Albanie, Zongqing Lu, and Kai Han. 2025. GAMEBoT: Transparent Assessment of LLM Reasoning in Games. In Proceedings of the 63rd Annual Meeting of the Association for Computational Linguistics (Volume 1: Long Papers), pages 7656–7682, Vienna, Austria. Association for Computational Linguistics.

**Questions:**

Except convenience, what are the other advantages of TALES compared to respectively evaluating LLMs on each of these benchmarks: TextWorld, TextWorldExpress, ALFWorld, ScienceWorld, Jericho and combine the results?

---

> ### Author Response · Authors · 2025-11-19
> **Rebuttal for Reviewer XTMp's review**
>
> We thank reviewer XTMp for their examination of our work and the subsequent feedback.
>
> Below are our responses to the outlined weaknesses and questions:
>
> Weaknesses:
>
> - The provided experiments are robust and the provided unifying of the existing text-adventure benchmarks would be beneficial from a engineering perspective. However, from the novelty view, this paper doesn't include new design of gaming benchmark. This work does not include new games, nor new design of evaluation. Meanwhile, leveraging interactive environments to assess various reasoning abilities is not new. And it is not surprising to see LLMs would fail in gaming where complex reasoning is needed. [1][2][3]
>   - We address this in section ‘Motivation of Curation and Questions of Novelty’ of the general rebuttal as this is a concern shared with other reviewers
>
> Questions:
> - Except convenience, what are the other advantages of TALES compared to respectively evaluating LLMs on each of these benchmarks: TextWorld, TextWorldExpress, ALFWorld, ScienceWorld, Jericho and combine the results?
>   - Atop our arguments in ‘Motivation of Curation and Questions of Novelty’ from the general rebuttal, we would also like to emphasize that as part of our RL experiments, we will define explicit training and test sets (the seeds and configurations used to obtain the existing scores will be used as the test set) to allow for the community to leverage TALES as a measuring stick when building on models of a lower weight class than the edge of the frontier.

---

### Official Review · Reviewer_4NwH · 2025-11-06

**Soundness:** 3
**Presentation:** 2
**Contribution:** 3
**Rating:** 6
**Confidence:** 5

**Summary:**

This paper presents a unified benchmark suite integrating five existing text-adventure learning environments. The proposed benchmark is used to assess the zero-shot capabilities of LLMs. The evaluation measures the maximum score attainable within a fixed number of turns. Additionally, the analysis identifies common failure modes related to spatial, deductive, inductive, and grounded reasoning. The findings indicate that LLM-based agents can struggle even with a simplified "Simon says"-style game—a task considerably simpler than solving complex virtual puzzles. The results also suggest that these agents remain far from achieving optimal performance in games designed for human players.

**Strengths:**

1. The topic of this work is both compelling and valuable; the integration of diverse reasoning skills into a unified framework offers significant practical utility.

2. The experiment section demonstrates thorough engineering effort, encompassing a wide range of models, and clearly highlights a critical limitation in their long-horizon reasoning capabilities.

3. The design of the SIMON SAYS task offers a natural and well-grounded method for evaluating a model's ability to follow instructions.

4. With multiple concrete examples and detailed explanations provided in the appendix, this paper delivers a benchmark that will greatly benefit the research community and facilitate future studies.

**Weaknesses:**

1. The four reasoning skills proposed by the authors raise concerns regarding comprehensiveness, and there is a lack of discussion on the interrelationships among these skills. The classification comes across as more of an intuitive listing rather than a systematically constructed framework.
2. The discussion of "TO THINK OR NOT TO THINK" could be more thorough; it would benefit from a comparative analysis between the reasoning modes in this benchmark and those commonly found in pre-training data for LLMs.

**Questions:**

1. I want to know whether It would be valuable if the authors could provide results and analysis from simple fine-tuning of open-source models on this benchmark, which could further encourage the research community to adopt it for training and optimization purposes.

---

> ### Author Response · Authors · 2025-11-19
> **Rebuttal for Reviewer 4NwH's review**
>
> We thank reviewer 4NwH for their examination of our work and the subsequent feedback.
>
> Below are our responses to the outlined weaknesses and questions:
>
> Weaknesses:
>
> - The four reasoning skills proposed by the authors raise concerns regarding comprehensiveness, and there is a lack of discussion on the interrelationships among these skills. The classification comes across as more of an intuitive listing rather than a systematically constructed framework.
>   - We would direct the reviewer to section ‘Validity of Qualitative Analysis Results’  in the general rebuttal as this is a shared concern among other reviewers.
>
> - The discussion of "TO THINK OR NOT TO THINK" could be more thorough; it would benefit from a comparative analysis between the reasoning modes in this benchmark and those commonly found in pre-training data for LLMs.
>   - We are having difficulty understanding the reviewer’s intent when describing “reasoning modes commonly found in pre-training data for LLMs’. Would they be able to expand on this point?
>
> Questions:
> - I want to know whether It would be valuable if the authors could provide results and analysis from simple fine-tuning of open-source models on this benchmark, which could further encourage the research community to adopt it for training and optimization purposes.
>   - Per the reviewer’s suggestions, and to address other reviewers’ concerns about the level of contribution of this work, we will include this. Please see section ‘Motivation of Curation and Questions of Novelty’ for more details.

---

### Official Review · Reviewer_rY5d · 2025-11-07

**Soundness:** 2
**Presentation:** 2
**Contribution:** 1
**Rating:** 2
**Confidence:** 3

**Summary:**

Note: This is a review by an emergency reviewer.


This paper presents the TALES benchmark for evaluating LLMs on text-adventure game environments that require 4 core reasoning abilities: deductive, inductive, spatial, and grounded reasoning. The authors argue that the compositional reasoning capabilities needed to perform well on TALES are critical for real-world applications of LLM-based agents. The benchmark is created from 5 existing frameworks, each comprised of multiple games, and introduces an initial instruction-following task (Simon Says).  The authors present the scores for 10 strong LLMs on TALES, finding low scores on the games from one framework but very high scores on the others. Further analysis evaluates the reasoning traces of some of the evaluated models, identifies reasoning failures, and compares thinking vs non-thinking models. Finally, prompting strategies are compared for a weak open-source model.

**Strengths:**

**Concept**: the idea of integrating existing text-based adventure games into a reasoning-focused benchmark as an evaluation of reasoning for real-world applications is interesting.

**Evaluation breadth**: the authors evaluate a wide range of frontier LLMs on TALES with results for additional weaker models also reported in the appendix (42 LLMs in total). This gives a comprehensive overview of current capabilities.

**Evaluation approach**: the decision to use a standardised, lightweight evaluation approach that does not include domain knowledge strengthens the evaluation.

**Limitations**: the authors acknowledge and discuss several limitations with their approach.

**Visualizations**: the anonymized github link includes numerous visualizations of per-game performance, along with a measure of score spread across different runs.

**Weaknesses:**

**Difficulty**: for me, a major limitation of this benchmark lies in its difficulty. Specifically, frontier models score very highly on 4 of the 5 frameworks (88-100% for o3 medium) – the games in these frameworks are either approaching saturation or are saturated, and evaluating on them offers limited insights. Therefore, I see the core value lies in the results of the games from the Jericho framework, which do prove challenging for current frontier models (though they are acknowledged by the authors as potentially suffering from data contamination). Given this, the significance of the contribution of TALES seems limited. Down-selecting just the most challenging subset of games from the 4 high-scoring frameworks could increase the overall difficulty, but given the lack of headroom, this would filter out most games.

**Curation**: the description of TALES lacks reasoning for why the 5 frameworks were selected. Statistics on the reasoning types covered by each game would be useful here, as would a clearer overview of the types of games included in each. How do these games relate to real-world applications, and what specific real-world applications do you think attaining strong performance on TALES unlocks?

**Analysis**: the analysis (Sec. 5) would be strengthened by adding example reasoning traces with failures (from the Appendix). This would help contextualise the qualitative insights.

**Clarity**: several aspects of the paper could be made clearer:

-	The authors compare the results of synthetic games vs games “designed for human enjoyment” but do not specify which games/frameworks correspond to which.

-	The TALES score calculation is not defined in the paper.

-	In Tab. 1, the units for walkthrough length are not stated

-	More examples of the games from the different frameworks (e.g., Fig. 1) would help provide context.


**Minor errors/typos**: the following lists some of the minor errors, inconsistencies and typos in the paper:

-	Line 156: “Figure 1 illustrates a simple task in a text-adventure game where multiple reasoning skills are required at each step…”. Most steps in the figure don’t require multiple reasoning skills

-	Line 40 missing word in “need apply”

-	Lines 49, 401 missing space between text and citation e.g., “task(Paglieri…)”

-	Line 93 missing word in “a collection games”

-	Line 214: “the player receive” -> receives

-	Line 239: remove comma after 54

-	Line 257: missing word: “coming from human expert”

-	Line 288: missing word: “but find”

-	Line 351: missing word “when a subgoal was and”

-	Line 358: missing word “failures still when multiple…”

-	Line 406: missing word: “are same as main results”

-	Line 434: missing word: “reduce the space possible commands”

-	Line 446: missing full stop/period.

-	Table 1 is not referenced in the paper.

-	Line 243: “For example, 9:05 follows the morning of an ordinary office worker where ANCHORHEAD is a Lovecraftian Horror Story” – where should be while?

-	Introduction: “games” and “tasks” are used interchangeably to refer to the suite of 122 games.

-	citep / citet usage is inconsistent

-	TALES is presented as both a collection of frameworks (Line 179) and also described as a framework itself (Line 99).

**Questions:**

How are the TALES scores calculated? It’s not the mean of the average scores per framework but appears to be a weighted average based on the number of games in each framework.

Have you explored evaluating a human baseline on TALES or the Jericho games in particular?

---

> ### Author Response · Authors · 2025-11-19
> **Rebuttal for Reviewer rY5d's review:**
>
> We thank reviewer rY5d for their examination of our work and the subsequent feedback. We would further like to explicitly thank them for their time, effort, and service as an emergency reviewer in ensuring our submission was given proper attention.
>
> Below are our responses to the outlined weaknesses and questions:
>
> Weaknesses:
> - Difficulty: for me, a major limitation of this benchmark lies in its difficulty. Specifically, frontier models score very highly on 4 of the 5 frameworks (88-100% for o3 medium) – the games in these frameworks are either approaching saturation or are saturated, and evaluating on them offers limited insights. Therefore, I see the core value lies in the results of the games from the Jericho framework, which do prove challenging for current frontier models (though they are acknowledged by the authors as potentially suffering from data contamination). Given this, the significance of the contribution of TALES seems limited. Down-selecting just the most challenging subset of games from the 4 high-scoring frameworks could increase the overall difficulty, but given the lack of headroom, this would filter out most games.
>   - On top of our responses in the general rebuttal, we would like to add the following:
> The synthetic environments (textworld, twx, alf, etc) were specifically designed for training AI agents, so we were not surprised that the frontier LLMs climbed the scoreboard here very quickly. However, we believe this acts as a good ‘measure stick’ for progress of smaller models as many fail to replicate anywhere near the success of their larger counterparts.
>
>
> - Curation: the description of TALES lacks reasoning for why the 5 frameworks were selected. Statistics on the reasoning types covered by each game would be useful here, as would a clearer overview of the types of games included in each. How do these games relate to real-world applications, and what specific real-world applications do you think attaining strong performance on TALES unlocks?
>   - The five frameworks specifically were selected as we felt each has a unique attribute that contributes to TALES as a whole:
>
>     Textworld evaluates performance on the ‘same’ task across a range of difficulties (Minimal acts to complete the task, distractors present in the environment, etc).
>
>     TextworldExpress evaluates performance across a range of different tasks, for example, ranging from a similar task to Textworld to SimonSays to simple mathematics.
>
>     ALFWorld evaluates a model’s ability to deal with strict requirements on command specification along with limited feedback from the environment.
>
>     Scienceworld tests similar qualities as Textworld and TextworldExpress but across a far larger range of tasks with a much higher baseline difficulty.
>
>     Jericho tests all of the above with each attribute represented in one or more of the tasks(games) present in its framework.
>
>     We argue that strong performance in TALES is indicative of the ability to consistently perform the outlined reasoning skills in composition, continuously over long horizons. We discuss this in the general rebuttal under section ‘Validity of Qualitative Analysis Results’. Expanding on this, we believe these reasoning skills are critical for LLMs when acting as the driver for agentic applications. With a coding agent as an example, ‘grounded reasoning’ may be critical for an agent to keep track of what changes it's already made in an extended context. ‘Inductive reasoning’ would be essential as if an agent’s code does not work out of the box, it will need to iterate and adjust according to environment feedback (or lack thereof).
> - Analysis: the analysis (Sec. 5) would be strengthened by adding example reasoning traces with failures (from the Appendix). This would help contextualise the qualitative insights.
>   - We will make adjustments to do this in our fully revised draft with the additional page limit.
> - Clarity/ Minor errors/typos:
>   - We sincerely thank the reviewer for their attention to detail in this aspect: We will ensure all of these concerns are addressed in our fully revised draft.
>
> Questions:
> - How are the TALES scores calculated? It’s not the mean of the average scores per framework but appears to be a weighted average based on the number of games in each framework.
>   - That is correct, the score is calculated by averaging the score per task across all tasks. We will make sure to clarify this.
> - Have you explored evaluating a human baseline on TALES or the Jericho games in particular?
>   - While we do know that humans are able to beat all Jericho games given enough time, we believe a human baseline largely to be outside the scope of this work. However, this would definitely be an interesting direction to explore in future work.

---

### Author Response · Authors · 2025-11-19
**General Rebuttal: Common Trends and What We Are Working On**

We thank all reviewers for their feedback and exploration of our work. Below, we address general strengths and concerns shared by reviews. Concerns specific to a single reviewer are addressed in the respective comments.

**Common Strengths**:
- Motivation and Concept behind our benchmark: Reviewers rY5d, 4NwH, and XTMp all acknowledge the value of text-adventure games in evaluating the long-horizon reasoning capabilities of LLMs
- Breadth of zero-shot experiments: All reviewers praise the comprehensiveness of our zero-shot evaluation on many existing open and closed weight models.
- Unified and standardization: Reviewers rY5d, 4NwH, and 93MW report positively on the evaluation standardization contributed by TALES

**Common Concerns**:
- **Difficulty/Saturation of the Benchmark**: Reviewers rY5d, 4NwH, 93MW all express concern over the difficulty of the benchmark due to the performance of frontier LLMs
  - Response:
    We argue that the saturation of the easier frameworks in our benchmark by frontier LLMs does not make it less valuable. We offer all the frameworks in TALES as a progression system with easier environments (textworld) as entry points to gauge the capability of a model before moving on to harder and more comprehensive environments (jericho).

    We believe this is needed due to the sheer computational requirement for some environments. The most complex environments can take well over 100 steps to complete, which can be an insurmountable barrier to entry in terms of compute cost. Easier environments, even those ‘solved’ by the most powerful modern LLMs, allow for the cultivation of abilities in smaller models not yet ready to challenge the most complex tasks within TALES. We discuss this further in ‘RL finetuning results’.

    We would also like to emphasize that while some frontier LLMs have managed to achieve the complete score on ‘easier’ frameworks, these are ALL closed-source models. We would argue for the importance of providing evaluation metrics for models beyond those at the closed frontier as well.

- **Validity of Qualitative Analysis Results**: While reviewer rY5d notes the qualitative analyses might be further improved by including the explicit reasoning failures otherwise relegated to the appendix, reviewers 4NwH and 93MW believe the analysis provided is not sufficient.
  - Response:
    We would emphasize that the reasoning skills and subsequent analysis are meant to supplement the scalar insights provided by the raw scores achieved by the models, primarily derived from explicit analysis of when and why frontier models failed in TALES and what those failure modes were. The motivation behind our introduction of these reasoning skills is to distill what weaknesses in model capabilities are exposed by their failures in TALES beyond a generic ‘failure to play text-adventure games’.

- **Motivation of Curation and Questions of Novelty**: Reviewers rY5d and XTMp express concerns over the contribution of TALES beyond what could be gained by evaluating on the combination of the individual frameworks alone and question the motivation behind the inclusion of all frameworks.
  - Response: Compared to many of the other text-based games in the works cited by reviewer XTMp, text-adventure games are unique in that the observations are directly provided as a natural language where many of the referenced text games instead leverage ascii transformations of visual information.

    On top of providing results across 100 steps for many frontier models, which may otherwise be prohibitively costly for individual researchers to run, we further provide the SimonSays game mode for TextworldExpress. Additionally, TALES is kept modular such that additional text-games can be later added.

    We couldn’t stress enough the need for a standardized method of evaluation in these text-adventure games, as we argue in section 5.4. We will elaborate on this in a separate comment due to space limitations.

    While we would like to emphasize the primary contribution of TALES being the unified evaluation framework, at reviewer 4NwH’s suggestion we will provide RL-finetuning results and insights across each synthetic environment (Textworld, TextworldExpress, ALFWorld, and ScienceWorld). For this, we will specifically use the verl-agent framework, which claims state-of-the-art RL fine-tuning results on ALFWorld, to further emphasize why there is a need to standardize the adaption module in these environments.

    We will also examine the potential effects of memorization through an extended RL training where the model is both trained and evaluated solely on Zork1.

**What we are working on**
To conclude, we would like to list out our promises and deliverables.

We will:
- Upload a revised draft with all grammatical and clarical issues mentioned by reviewers addressed.
- Include a suite of baseline RL Experiments

---

> ### Author Response · Authors · 2025-11-19
> **A Note to All Reviewers: The need for a standardized baseline of evaluation in these frameworks.**
>
> We would like to specifically draw all reviewers’ attention to the following points that may have not been sufficiently emphasized in our original submission:
>
> There is a need for standardization in the evaluation of LLM agent performance in these frameworks as different works make different choices about what environment information is exposed to the agent. These choices significantly impact the difficulty and comparability of the evaluated approaches. For example, while we provide only the observation and feedback to the agent, other recent work (see https://openreview.net/pdf?id=UOzxviKVFO, also submitted to ICLR 2026) relies on ground-truth information from the environment such as admissible actions to achieve their results.
>
> Ground truth environment information, such as admissible actions, was initially exposed in the frameworks in TALES to facilitate training of non-LLM based RL agents with no natural language priors (see our related works section on non-LLM based RL work). For LLMs, capable of producing natural language actions with no additional fine-tuning, this information effectively acts more like verification provided by the environment. Viewing this extra, ground-truth information from the environment as a form of verification raises two significant concerns.
>
> First, the exposure of such information to the model significantly diminishes the challenge of these environments. For example, in the case of admissible actions, we can use an analogy to illustrate the degree to which this changes the task. The inclusion of admissible actions at each step can be seen as turning a ‘free response’ question into a ‘multiple choice’ question, the latter of which is significantly easier unless carefully crafted to explicitly include misleading options. Reviewer 93MW even raises concerns over the impact of the prompt on performance, in which these admissible actions are often embedded. This is why we advocate for a minimal prompt approach.
>
> Second, this can make it difficult to interpret the true impact of the contributions authors who evaluate on these environments. While this may be less of an issue were this information explicitly outlined as part of the authors’ methods, these details often do not make it into the main body of the work. Similar to reviewer rY5d’s concerns over the difficulty in TALES, this can lead to significant issues where methods that do not leverage, or do not need to leverage, this additional information from the environment are seen as lower performing than methods that do.
>
> We have already observed instances of this occurring in reviews of recent work where methods that do leverage this additional information are compared against methods that do not without accounting for these differences in evaluation setting: https://openreview.net/forum?id=K6T0o875zF&noteId=6uYt9jHxXl
>
> Beyond admissible actions, this further includes other domain knowledge about the environment the LLM would need to otherwise discover and reason about on its own, including the specific function of certain actions, inventory management, and even entire trajectories from adjacent tasks.
>
> In submitting TALES to the broader community, we hope to address this by providing a minimal, standardized approach to act as a baseline such that any additional information extracted from the environment will be documented and given sufficient acknowledgement when authors describe their approach to the frameworks within TALES. We see this as an important downstream impact of our work in facilitating explicit communication between researchers in terms of the minor but impactful details about approaches that tend to get lost in the cracks.
>
> We only briefly touched on this in section 5.4 to keep the focus of our paper on TALES itself and DO emphasize that prior work using these environments provide significant and meaningful contributions to the scientific community.

---

### Author Response · Authors · 2025-11-29

Hello AC/s,

If we had posted this a few days ago, it would have been addressed to our reviewers. Instead, we will direct this comment to the new AC who will be overlooking our submission.

We would like to explicitly thank you for your service due to the additional workload imposed on all ACs as a result of the incident regarding reviewer anonymity.

In this comment, we hope to concisely represent the strengths of our submission and common themes of reviewer concerns to the best of our ability as well as the steps we have taken to address them in our revised draft. We would direct you to our comments and responses below for a more extensive, point-by-point response to the concerns raised by reviewers.

We hope this comment will be useful to you in your examination of our submission.

Strengths:
- Reviewers acknowledge the value of Text-Adventure Games for evaluating long-horizon reasoning
- Breadth of zero-shot experiments to serve as a baseline (42 models, 122 tasks, 100 max steps, 5 seeds)
- Argument and groundwork for standardization of evaluation settings in text-adventure games

Weaknesses/Concerns:

- Difficulty of Benchmark due to the best models achieving 100% on easier frameworks
   - Response: This is only applicable to closed-source frontier models, these frameworks are still valuable for developing models in lower weight classes

- Validity of reasoning qualitative analysis
  - The reasoning analysis is meant primarily to supplement the results from the overall TALES score in emphasizing that failure modes in text-adventure games can result from a failure in reasoning capabilities desired for other applications.

- Motivation of Curation and Questions of Novelty
  - We emphasize the primary contribution of TALES as the unified framework, the breadth of zero-shot baseline results, and the standardization of evaluation in text-adventure games. We further supplement this with a range of RL experiments, detailed in our rebuttal revision that investigate the effects of the effects of privileged or domain knowledge often directly included in the prompt, a practice we specifically avoid and argue against.

We have posted our revised draft with all additions, error corrections, and new sections marked in green for visual clarity. Prior to the end of the discussion period, we will remove this coloring.

In our revised draft, we have included a new section, Section 6, that discusses insights from the RL experiments we provided both at Reviewer 4NwH's request, and to mitigate concerns of the level of contribution of our work.

We are still completing some experiments, however these will likely only appear in the appendix unless you indicate a strong preference otherwise. For the most part, the main body of our submission is complete.

We wish you a smooth reviewing process with no more complications, and once more thank you for your additional service due to the unexpected complications that occurred during this ICLR review cycle.

---

> ### Author Response · Authors · 2025-12-03
> **A Brief Update**
>
> Hello,
>
> As the discussion period comes to a close, we would like leave some brief remarks:
>
> - To aid the new AC in identifying new content, we have decided to leave the rebuttal-revision highlights. This will be removed in the camera ready, should we be accepted.
> - While some of our additional experiments have concluded, we feel we would not be able to sufficiently polish the added content before the end of the discussion period. As this content would be included in the appendix regardless, we have elected to simply exclude it for now.
>
> Once more, we thank the ACs of ICLR for their service and look forward to their evaluation of our work.

---

### Meta-Review · Area_Chair_cWLk · 2026-01-07

**Summary:**

This paper introduces TALES, a benchmark suite that unifies five existing text-adventure game environments to evaluate long-horizon reasoning in language-model-based agents.
Several concerns raised included whether the contribution is novel beyond unifying existing benchmarks, whether the benchmark remains challenging for current frontier models, and whether TALES scores reliably isolate reasoning ability from prior knowledge, environment quirks, or interaction budget effects. Additional concerns were raised about the clarity of benchmark design choices, the definition and interpretation of the TALES score, the absence of human or heuristic baselines, and the limited empirical validation of the proposed reasoning skill taxonomy.
The rebuttal clarified several design decisions, expanded discussion of benchmark difficulty and scope, and committed to additional analyses and experiments. These responses addressed some concerns.

**Reviewer Concerns:**

**Concerns addressed by the rebuttal:**

- The authors clarified that the main contribution is a standardized evaluation protocol across multiple existing environments, rather than the introduction of new games.

- The selection of the five frameworks was justified by explaining how each contribution distinct with the others.

- The use of minimal prompts and avoidance of privileged environment information were clarified.

- The authors explained that easier synthetic environments are intended as entry points and progression benchmarks, particularly for smaller or open-weight models, while Jericho provides headroom for frontier models.

**Concerns that remain outstanding:**

- Some reviewers continue to view the contribution as primarily a benchmarking consolidation, with limited methodological novelty.

- Requests for stronger evidence that TALES scores measure the proposed reasoning skills, rather than other factors, remain only partially addressed.

- The proposed reasoning skill taxonomy remains concerns and is not empirically validated.

- Potential data contamination is still questionable.

- Missing discussion of some related works.

**Reviewer Scores:**

Reviewer rY5d: Initially 2. Despite clarifications, concerns about novelty and benchmark difficulty still remain. Score may not change.

Reviewer 4NwH: Initially 6. Core concerns were addressed or acknowledged; the score would likely remain unchanged.

Reviewer XTMp: Initially 4. Clarifications about motivation and standardization may lead to a small increase.

Reviewer 93MW: Initially 4. Concerns about construct validity and other experimental setting details remain. The score would likely remain unchanged.

---

### Decision · Program_Chairs · 2026-01-26

Reject